# Supramolecular dynamics-enhanced synergistic antifouling mechanisms for enhanced membrane antifouling and permeability

Mingrui He, Yulun He, Dongwei Lu [ORCID] ✉, Mengfei Wang, Junjie Yang, Tong Wu & Jun Ma [ORCID] ✉

This study develops water treatment membranes using an innovative surface modifier comprising threaded supramolecular assemblies formed by hydrophilic cyclodextrin (CD) and low-surface-energy polydimethylsiloxane (PDMS). These supramolecular constructs establish dynamic hydrophilic and low-surface-energy heterogeneous microdomains that enhance synergistic resistance-release antifouling mechanisms. The modified membranes demonstrate better antifouling performance compared to conventional systems, particularly addressing the critical limitation of traditional membranes under low tangential flow conditions. The Brownian motion of the CDs sustains microdomain activity to prevent foulant accumulation in static environments, while tangential flow amplifies dynamic interactions to accelerate foulant detachment. The threading configuration of CDs along PDMS chains prevents water channel blockage caused by PDMS aggregation and facilitates water transport through the dynamic mobility of CDs. When separating bovine serum albumin solutions under an initial flux of 550 $L \cdot m^{-2} \cdot h^{-1}$ with 60 rpm stirring, the membrane exhibits merely 14.2% flux decline, highlighting its exceptional antifouling performance and permeability.

A fundamental strategy for enhancing the antifouling performance of membranes is improving their hydrophilicity. Hydrophilic membrane surfaces preferentially bind water molecules, leading to the formation of hydration shells that prevent foulants from adhering to the surface. This process is known as the fouling resistance mechanism[1,2]. However, hydrophilic modification strategies only have a limited effect on foulants that have already adhered to and deformed on the membrane surface, leading to gradual foulant accumulation and reduced membrane lifespan[3]. To overcome this problem, synergistic resistance-release antifouling mechanisms, based on hydrophilic and low-surface-energy (LSE) heterogeneous microdomains, have been adopted[4]. The hydrophilic microdomains retain the fouling resistance mechanism, while the LSE microdomains introduce fouling release mechanism by reducing the interaction forces between foulants and the surface, thereby promoting the detachment of adhered foulants under environmental disturbances[5,6]. Synergistic antifouling mechanisms have been widely adopted in marine antifouling coatings due to their broad applicability against diverse foulants and long-term stable antifouling performance[7,8]. However, the distinctive characteristics of membrane processes pose two major challenges to the application of the synergistic antifouling mechanisms in this field.

The first challenge is that, unlike marine antifouling coatings, the fouling release mechanisms of water treatment membranes require more intense tangential flow[9]. This is because the transmembrane

State Key Laboratory of Urban-Rural Water Resource & Environment, School of Environment, Harbin Institute of Technology, Harbin, People's Republic of China. ✉e-mail: lvdongwei126@126.com; majun@hit.edu.cn

water transport processes tightly press foulants against the membrane surface[10,11]. In contrast, the fouling resistance mechanism is less dependent on tangential flow[12,13]. Therefore, under low tangential water flow, membranes based on synergistic antifouling mechanisms are more prone to fouling compared to those relying solely on the fouling resistance mechanism[4,14]. The required intensity of tangential flow is related to the rate of transmembrane water transport. Under a conventional membrane flux of about 100 L·m⁻²·h⁻¹, the tangential flow velocity needed to activate the fouling release mechanism is more than three times higher than that needed to mitigate concentration polarization[15–17]. This implies the need for high-speed stirring in dead-end filtration mode and necessitates a relatively low water recovery rate in crossflow filtration mode. In practical membrane operations, weak tangential flow regions, commonly observed in the central zones of dead-end modules and around spacers in crossflow configurations, pose a significant barrier to the scalable application of synergistic antifouling mechanisms.

The second challenge is the hydrophobic nature of LSE microdomains, which significantly increases the impedance of the membranes to water permeation[18–20]. In most reported studies, hydrophilic-LSE microdomains are self-assembled from polymeric materials[21,22]. In aqueous environments, LSE microdomains are prone to agglomeration and adhere to the membrane surface, where they block water channels under transmembrane pressure, leading to a considerable reduction in membrane permeability[19]. While synergistic antifouling mechanisms can improve the membrane antifouling performance compared to the fouling resistance mechanism alone, there is a significant trade-off in terms of reduced spatial utilization and higher operational energy consumption. To address this bottleneck, researchers have explored strategies for optimizing the distribution of heterogeneous microdomains using the swelling properties of hydrophilic polymers in water. For instance, LSE molecules were grafted onto the ends of hydrophilic polymers, and the ratio of LSE to hydrophilic molecular segments were finely controlled[6,23,24]. However, these methods are complex and costly, and they only modestly improve membrane flux.

In this study, the use of supramolecules, rather than polymers, to construct antifouling layers in water treatment membranes is proposed to address the two major challenges of fouling resistance-release synergistic mechanisms. Specifically, a supramolecular complex consisting of polydimethylsiloxane (PDMS) and cyclodextrin (CD) is employed, as illustrated in Fig. 1. PDMS is a linear molecule and CD is a ring-shaped molecule. Threading CDs onto the PDMS chain forms a threaded and ring-like structure. These supramolecular products are classified as poly(pseudo)rotaxanes (PPRs) or polyrotaxanes (PRs) depending on whether the CDs can freely detach from the PDMS chain (i.e., when CD/PDMS PPRs are grafted onto the membrane surface via the PDMS end groups, they are transformed into CD/PDMS PRs). The hydroxyl-rich outer wall of CDs forms hydrophilic microdomains. Meanwhile, PDMS has a surface energy of about 20 mJ·m⁻², leading to the formation of LSE microdomains[25]. The CDs can slide along the PDMS chains, rotate around the chain segments, and vibrate perpendicularly to them, forming dynamic hydrophilic microdomains that coexist alongside the LSE microdomains. On one hand, the relative motion within the heterogeneous microdomains causes the contact between the foulants and the membrane surface to be in a highly unstable state, making it easier for the foulants to be released. On the other hand, this relative motion helps prevent LSE microdomain agglomeration and facilitates water molecule transfer, leading to lower water permeation impedance. Additionally, owing to the lack of covalent bonding constraints and the weak intermolecular interactions between CDs and PDMS, the inherent Brownian motion of CDs enables the microdomains to maintain dynamic mobility even under stagnant conditions[26,27]. Tangential flow can further enhance the activity of the dynamic microdomains, resulting in improved antifouling

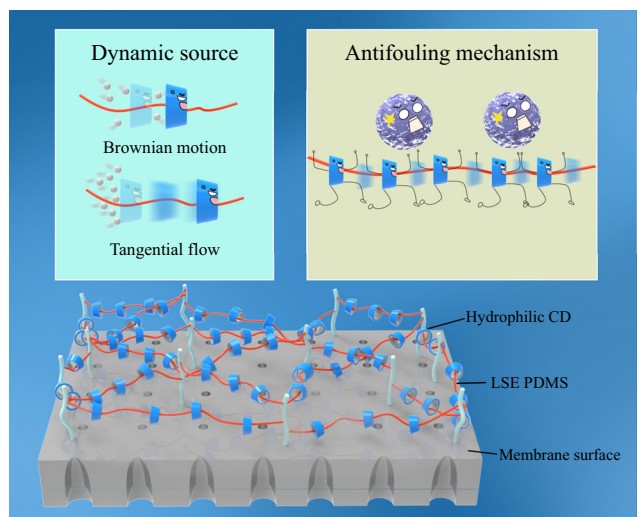

**Fig. 1 | Supramolecular structure, dynamic mechanism, and antifouling mode of CD/PDMS PRs.** Illustration of the structure and surface loading of the antifouling modifier CD/PDMS PRs (bottom), the dynamic source of CD mobility (top left, including intrinsic Brownian motion and tangential flow enhancement), and the supramolecular dynamics-enhanced antifouling mechanism (top right).

performance of the membrane. Specifically, this study investigated the influence of supramolecular dynamics on membrane permeability and synergistic antifouling performance by tuning the type of CD and the molar ratio of CD to PDMS in CD/PDMS PRs. Different types of CDs possess similar chemical compositions but vary in cavity size, which affects the spatial hindrance during their movement. This allows the elimination of compositional influences related to the hydrophilic–LSE microdomain ratio, enabling an isolated assessment of microdomain dynamics. By adjusting the CD/PDMS molar ratio, both the number of mobile CDs and their available movement space can be simultaneously regulated, facilitating the identification of supramolecular structures with the highest dynamic mobility.

## Results
### Preparation and optimization of CD/PDMS PPRs
To modulate the dynamic behavior of CD/PDMS PPRs via cavity-size-controlled interactions, the synthesis was attempted using three commonly used CDs—α-CD, β-CD, and γ-CD—with inner cavity diameters of 4.7–5.3 Å, 6.0–6.5 Å, and 7.5–8.3 Å, respectively. To date, the synthesis of the three CD/PDMS PPRs has only been explored in aqueous solutions[28–30]. According to the results, α-CD/PDMS PPRs could not be synthesized. The synthesis of γ-CD/PDMS PPRs was straightforward, with a yield (calculated based on CD) exceeding 90%. When γ-CD fully covered PDMS, the molar ratio of CDs to PDMS units ($r_{CD:PDMS}$) was 0.67, which is the theoretical maximum value for $r_{CD:PDMS}$[28,31]. For β-CD/PDMS PPRs, synthesis was possible, but the achievable $r_{CD:PDMS}$ significantly decreased when the molecular weight of PDMS exceeded 1000 Da. This is because the formation of CD/PDMS PPRs essentially involves the displacement of solvent molecules from the CD cavities by PDMS molecules. This process is reversible and influenced by steric hindrance, which acts as a resistance, and solvent polarity, which serves as the driving force. The cavity diameter of α-CD is small, causing excessive steric hindrance for the entry of PDMS, which makes synthesis difficult. γ-CD, with a large and hydrophobic cavity, facilitates the entry of PDMS. β-CD has an intermediate cavity size, allowing PDMS to enter, but the steric hindrance is more significant. As the number of β-CD molecules that interact with PDMS increases, the steric hindrance becomes even more pronounced. Steric hindrance is fixed for certain CD, so the solvent conditions were optimized to attempt the synthesis of α-CD/

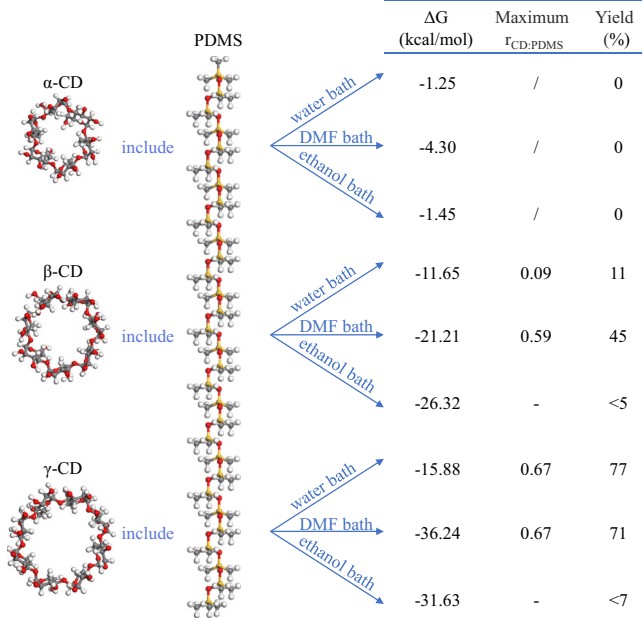

| | | ΔG (kcal/mol) | Maximum $r_{CD:PDMS}$ | Yield (%) |
|---|---|---|---|---|
| α-CD | water bath | -1.25 | / | 0 |
| | DMF bath | -4.30 | / | 0 |
| | ethanol bath | -1.45 | / | 0 |
| β-CD | water bath | -11.65 | 0.09 | 11 |
| | DMF bath | -21.21 | 0.59 | 45 |
| | ethanol bath | -26.32 | - | <5 |
| γ-CD | water bath | -15.88 | 0.67 | 77 |
| | DMF bath | -36.24 | 0.67 | 71 |
| | ethanol bath | -31.63 | - | <7 |

**Fig. 2 | Optimization of CD/PDMS PPR synthesis in different solvents.** Gibbs free energy changes during the formation of α-CD/PDMS, β-CD/PDMS, and γ-CD/PDMS PPRs, along with the maximum $r_{CD:PDMS}$ and reaction yields in water, DMF, and ethanol baths. The ΔG values were obtained via molecular simulations, while the maximum $r_{CD:PDMS}$ and yields were determined through experimental validation. "/" indicates missing data; "−" indicates data excluded due to poor repeatability.

PDMS PPRs and to improve the maximum $r_{CD:PDMS}$ and yield of β-CD/PDMS PPRs. Molecular simulations were performed to calculate the CD/PDMS PPR formation tendency in solvents with different polarities, as shown in Fig. 2 (see Supplementary Figs. 1–3, Supplementary Table 1 and Supplementary data for the simulation process). The selected solvents were water, DMF, and ethanol, which have polarities of 10.2, 6.4, and 4.3, respectively. The Gibbs free energy changes (ΔG) before and after the formation of CD/PDMS PPR were used to evaluate the formation tendency of CD/PDMS PPR. A large negative ΔG indicates that the formation of CD/PDMS PPR is thermodynamically favorable. The ΔG of α-CD/PDMS PPR in all three solvents were close to zero, indicating that changing the solvent would not promote the synthesis of α-CD/PDMS PPRs. No α-CD/PDMS PPRs were detected in verification experiments, confirming these simulation results. The formation of β-CD/PDMS and γ-CD/PDMS PPRs in DMF and ethanol environments showed more negative ΔG values than the formation in water, suggesting more favorable synthesis conditions. Experimental results (see the "Preparation of CD/PDMS PPRs" section for details) confirmed that using DMF as a solvent instead of water enabled the controlled synthesis of β-CD/PDMS PPRs with 5000 Da PDMS. The maximum $r_{CD:PDMS}$ achieved was 0.59, with a yield of 45%, which meets the study's requirements for controlling the dynamic properties of β-CD/PDMS PPRs by adjusting the $r_{CD:PDMS}$. However, replacing water with ethanol as the solvent resulted in lower actual yields for both β-CD/PDMS and γ-CD/PDMS PPRs. This discrepancy may be attributed to the poor dispersion of β-CD and γ-CD in ethanol, which hindered effective contact between the CDs and PDMS, a factor not accounted for in the simulations. Ultimately, β-CD/PDMS PPRs with $r_{CD:PDMS}$ values of 0, 0.13, 0.27, 0.40, and 0.53 were prepared using DMF as the solvent, while γ-CD/PDMS PPRs with $r_{CD:PDMS}$ values of 0, 0.13, 0.27, 0.40, 0.53, and 0.67 were synthesized using water in this study. The corresponding NMR spectra are reported in Supplementary Figs. 4 and 5, and the detailed preparation parameters and yields are listed in Supplementary Table 2.

## Dynamic behavior of CDs in CD/PDMS PPRs

Molecular simulations were conducted to investigate the supramolecular dynamics of β-CD/PDMS PPRs and γ-CD/PDMS PPRs. Theoretically, the sliding, rotational, and vibrational motions of CDs along the PDMS chains are all influenced by steric hindrance between the CDs and PDMS, exhibiting a negative correlation. As a representative parameter, the Gibbs free energy barrier associated with CD sliding along the PDMS chain was used to evaluate the impact of steric hindrance. A lower sliding energy barrier reflects lower steric hindrance, thereby enabling a higher potential for supramolecular mobility. To quantify this, simulations were performed by sliding the CDs back and forth between two adjacent PDMS units, with energy variations evaluated at 42 equidistant points (Fig. 3). Due to convergence and repeatability issues, the CD movement was simplified to a forced displacement of the geometric center rather than simulating the full Brownian motion. Dynamic diagrams are provided in Supplementary Movies 1 and 2. The simulation results showed that γ-CD exhibited a significantly lower sliding energy barrier compared to β-CD, suggesting that γ-CD/PDMS PPRs possess higher dynamic activity than β-CD/PDMS PPRs. This is primarily attributed to the larger cavity size of γ-CD, which reduces steric constraints. Another possible factor is the difference in molecular rigidity between the two CDs. The terminal hydroxyl groups of β-CD are nearly coplanar, allowing the formation of a circular hydrogen-bonding network that enhances structural rigidity. This rigidity reduces conformational flexibility and increases molecular collisions during movement. In contrast, the terminal hydroxyls of γ-CD are not coplanar, granting it greater flexibility and enabling more dynamic behavior.

The sliding motion of the CDs on PDMS was further determined using quasi-elastic light scattering (QELS) (Fig. 4). Molecular Brownian motion scattered the incident light, causing the total intensity and frequency of the scattered light to fluctuate and shift. By measuring the time-dependent decay of the light intensity function, molecular diffusion state information can be analyzed using the Stokes-Einstein relationship. A previous study on α-CD/polyethylene glycol PPRs proposed that the three peaks in the QELS spectrum correspond to self-diffusion, sliding-diffusion, and cooperative diffusion (from low to high diffusion coefficient) in under-saturated PPRs[32]. However, in fully saturated PPRs, the supramolecular host becomes too crowded to allow movement, so the sliding-diffusion peak disappears. The analysis of CD/PDMS PPRs in this study aligns with this trend. Further investigation was conducted on the influence of CD type and $r_{CD:PDMS}$ on sliding-diffusion. Compared to β-CD/PDMS PPRs, γ-CD/PDMS PPRs had broader sliding-diffusion peak areas and larger diffusion coefficients, indicating more pronounced dynamic behavior. This was consistent with the molecular simulation. As $r_{CD:PDMS}$ increased for both CD/PDMS PPRs, the sliding-diffusion peaks shifted toward lower diffusion coefficients, while the peak areas initially increased and then decreased. The sliding-diffusion peaks of β-CD/PDMS PPRs were located at $r_{CD:PDMS} = 0.40$, while those of γ-CD/PDMS PPRs were located at $r_{CD:PDMS} = 0.27$. This phenomenon results from the trade-off between the CD quantity and individual CD activity. As the density of CDs on the PDMS chain increases, the sliding behavior of individual CDs becomes increasingly hindered by the neighboring CDs. The lower $r_{CD:PDMS}$ corresponding to the maximum activity in γ-CD/PDMS PPRs compared to β-CD/PDMS PPRs can be attributed to the fact that γ-CD, due to its lower sliding resistance, requires more space to fully realize its dynamic potential.

## Surface chemical composition and morphology of CD/PDMS@Ms

To minimize the interference of the base membrane (BM) surface on the activity of the CD/PDMS PPRs, a suspended, chain-like configuration was employed to anchor the CD/PDMS PPRs onto the membrane surface (Fig. 5a). First, glycidyl methacrylate (GMA) was grafted onto

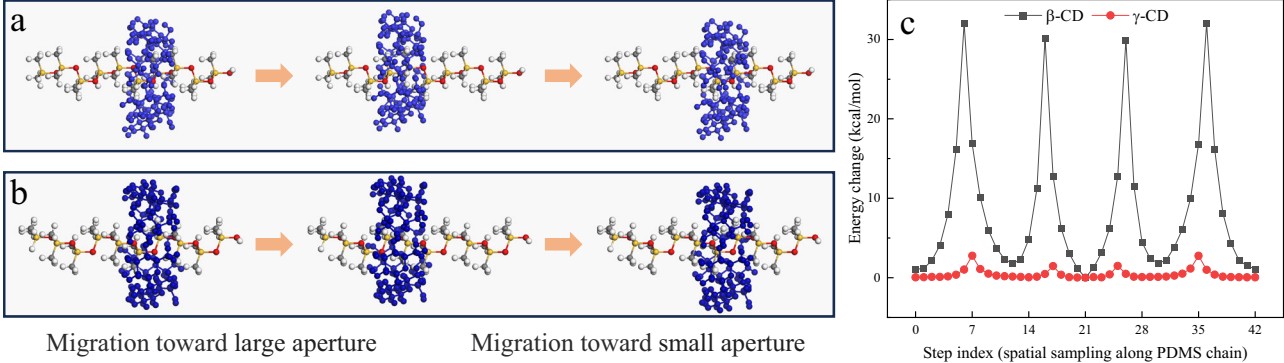

Migration toward large aperture          Migration toward small aperture

**Fig. 3 | Molecular dynamics simulation of CD sliding motion along PDMS.**
Simulated migration of **a** β-CD and **b** γ-CD along a PDMS chain, moving toward the larger aperture (steps 0–21) and then reversing (steps 21–42). CD atoms are marked in blue. **c** Gibbs free energy barriers encountered during migration. Each step index corresponds to a displacement of 1/10 of the projected z-axis length of a PDMS monomer unit.

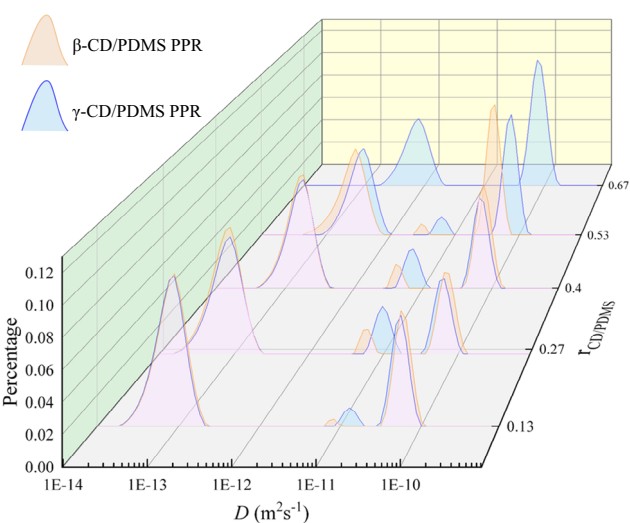

**Fig. 4 | Sliding-diffusion of CD/PDMS PPRs in DMF.** Diffusion coefficient $D$ ($m^2 \cdot s^{-1}$) distributions of β-CD/PDMS (blue) and γ-CD/PDMS (orange) PPRs. The overlapping area of the two distributions appears pink due to color blending.

the surface of a polyethersulfone (PES) BM via photo-induced radical polymerization to form linear poly(glycidyl methacrylate) (PGMA) with side epoxy groups. Next, the amino groups at the ends of the PDMS chains in the CD/PDMS PPRs underwent an addition reaction with the epoxy groups of PGMA, anchoring the CD/PDMS PPRs to the membrane. Finally, to prevent the CD leaching, the large molecule fluorescein-4-isothiocyanate (FITC) was employed to cap any unreacted terminal amino groups that were bonded to the membrane surface. The resulting membranes were denoted CD/PDMS@M-$r_{CD:PDMS}$ (e.g., PDMS@M and γ-CD/PDMS@M-0.67). Further details are provided in Supplementary Table 3.

The grafting of CD/PDMS PPRs on the membrane surface occurs in single-end and double-end grafting, its ability to more effectively prevent excessive contact between CD/PDMS PPRs and the BM. The amount of single-end grafted CD/PDMS PPRs on the membrane surface is directly proportional to the amount of the cap agent FITC, which emits green fluorescence under UV light. Therefore, fluorescence microscopy was employed for semi-quantitative analysis of the ratio of single-end to double-end grafting at different reaction duration, as shown in Fig. 5b, with corresponding fluorescence images provided in Supplementary Fig. 6. The fluorescence intensity on the membrane surface initially increased and then decreased with the

extension of reaction duration, indicating that most CD/PDMS PPRs initially underwent single-end grafting, and gradually transitioned to double-end grafting as the reaction progressed. After 4 h of reaction, the amount of single-end grafted CD/PDMS PPRs was approximately 3.4 times greater than that after 8 h of reaction, suggesting that the amount of double-end grafted CD/PDMS PPRs was at least 2.4 times greater than that of single-end grafted ones.

The surface morphologies of BM and CD/PDMS@Ms are shown in Fig. 5c–e and Supplementary Fig. 7. Both PDMS and CD/PDMS PRs covered the membrane surface without significantly obstructing the membrane pores. Comparing PDMS@M and β-CD/PDMS@M-0.13, it can be observed that CDs surround PDMS, significantly reducing its aggregation tendency. The membrane cross-sectional morphology shows that the CD/PDMS PR grafted layer does not penetrate deeply into the membrane pores, and its thickness is minimal (Supplementary Fig. 8). The molecular coverage of the modified membrane surfaces was quantitatively calculated based on their elemental composition. The XPS full spectra of the CD/PDMS@M surfaces showed oxygen, carbon, sulfur, and silicon peaks, although the nitrogen content was below the detection limit. The full spectrum of γ-CD/PDMS@M-0.27 is shown in Fig. 5f as a representative example. Disregarding the nitrogen contribution from the PDMS terminal amino groups and the iso-thiocyanate fluorophore, the mass-based surface coverage of CD, PDMS, PGMA, and PES molecules was approximately estimated from the elemental proportions of oxygen, carbon, sulfur, and silicon, along with the assumed positive correlation between PES and PGMA content (Fig. 5g, h, the calculation process is detailed in Supplementary Tables 4–7). The extent of membrane surface modification was approximately 40 wt%–55 wt%, with a slight increase with increasing $r_{CD:PDMS}$ value of CD/PDMS PRRs. This was attributed to the higher CD content on each PDMS chain. The calculated $r_{CD:PDMS}$ values of β-CD/PDMS@M surfaces were 0.11, 0.28, 0.38, and 0.48, while those of γ-CD/PDMS@M surfaces were 0.12, 0.24, 0.35, 0.49, and 0.67. Considering measurement error, it can be inferred that the grafting and capping processes on the membrane surface did not cause significant CD detachment from PDMS chains.

## Surface affinity and permeation flux of CD/PDMS@Ms

The impact of CD type and $r_{CD:PDMS}$ on the macroscopic hydrophilicity and surface energy of CD/PDMS@Ms was investigated using contact angle measurements (Fig. 6a–c, surface energy calculations provided in Supplementary Tables 8 and 9). The advancing contact angles of water were greater than the receding contact angles for both CD/PDMS@Ms, consistent with the trend of conventional heterogeneous membranes, where LSE microdomains promote water droplet detachment[15,19]. With increasing $r_{CD:PDMS}$, the advancing and receding

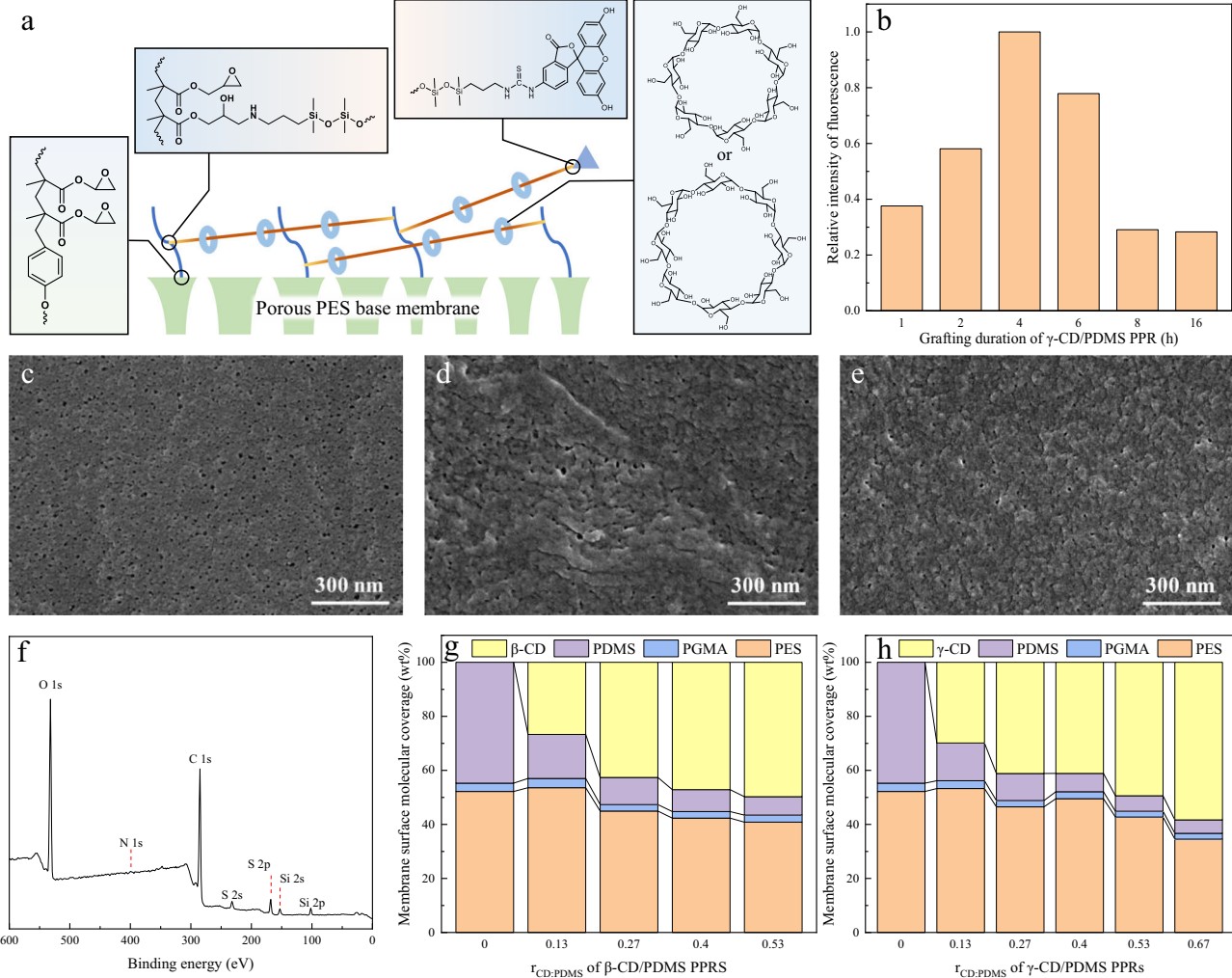

**Fig. 5 | Surface composition and morphology of CD/PDMS@Ms. a** Schematic illustration of CD/PDMS PR linkage modes on membrane surface. **b** Relative fluorescence intensities of γ-CD/PDMS@M surfaces after grafting PPRs for different durations and capping with FITC for 2 h. SEM images showing surfaces of **c** BM, **d** PDMS@M, and **e** β-CD/PDMS@M-0.13. **f** XPS full spectrum of γ-CD/PDMS@M-0.27. Surface molecular coverage of **g** β-CD/PDMS@Ms and **h** γ-CD/PDMS@Ms.

contact angles decreased for both CD/PDMS@Ms, while their surface energies increased. This effect is attributed to the hydrophilic nature of the CD outer walls. Comparing both CD/PDMS@Ms demonstrated that γ-CD/PDMS@M was more hydrophilic and had a lower surface energy than β-CD/PDMS@M. However, this difference decreased as the $r_{CD:PDMS}$ value approached 0 and 0.67. This trend suggests that, in addition to γ-CD having one more glucose unit than β-CD, the greater mobility of γ-CD also contributes to its enhanced hydrophilicity.

The initial water fluxes of CD/PDMS@Ms exhibited different trends at 5 °C, 20 °C, 35 °C, and 50 °C (Fig. 6d–g). At 5 °C, the initial water fluxes of CD/PDMS@Ms steadily increased with increasing $r_{CD:PDMS}$. At 20 °C, the fluxes significantly increased as $r_{CD:PDMS}$ increased up to 0.4, followed by a slight decrease. The trend at 35 °C was similar to that observed at 20 °C, but the turning point for γ-CD/ PDMS@M occurred at $r_{CD:PDMS}$ = 0.27. At a higher temperature of 50 °C, the turning point became more pronounced. The basic trends in water flux changes aligned with variations in membrane hydrophilicity. However, at higher temperatures, an additional factor contributed to enhanced flux, particularly at $r_{CD:PDMS}$ = 0.27 and 0.40. This reason was considered as the activity of CD/PDMS PRs. On one hand, the QELS spectra revealed that CD/PDMS PRs exhibited peak activity at $r_{CD:PDMS}$ = 0.27 and 0.40, corresponding to the flux turning point. On the other hand, grafting CD/PDMS PRs did not significantly alter membrane surface pore size (Supplementary

Fig. 9) or pore count (as seen in SEM images), excluding these factors from influencing flux. A further experiment was conducted to confirm the influence of CD/PDMS PR activity on flux. In aqueous environments, sodium dodecyl sulfate (SDS) can adsorb onto the PDMS surface and penetrate CD cavities via hydrophobic interactions, temporarily impeding CD motion along PDMS chains[33,34]. The flux variations were monitored with SDS solutions as feed (Fig. 6h). Results showed that only SDS concentrations as high as 1.00 wt% effectively suppressed membrane flux, and the suppression was temporary, for flux gradually returned to initial levels upon switching back to pure water. At 35 °C, when fed with 1.00 wt% SDS solution, the temporary steady-state fluxes of CD/PDMS@Ms and BM membranes are presented in Fig. 6i. Compared to Fig. 6f, the BM flux showed minimal change, while the flux of CD/PDMS@Ms decreased significantly at $r_{CD:PDMS}$ = 0.13, 0.27, and 0.40, with fluxes for γ-CD/ PDMS@M and β-CD/PDMS@M converging. This phenomenon inversely supports the role of CD/PDMS PRs in enhancing flux. This unusual phenomenon was likely due to the abundant hydrogen bonds on the outer surface of the CDs. These hydrogen bonds can bind water molecules and facilitate their rapid transfer through the sliding, rotating, and vibrating movements of CDs along the PDMS chains. At 20 °C, γ-CD/PDMS@M showed a maximum flux of approximately 550 L·m⁻²·h⁻¹, surpassing that of most reported heterogeneous membranes (Supplementary Table 10).

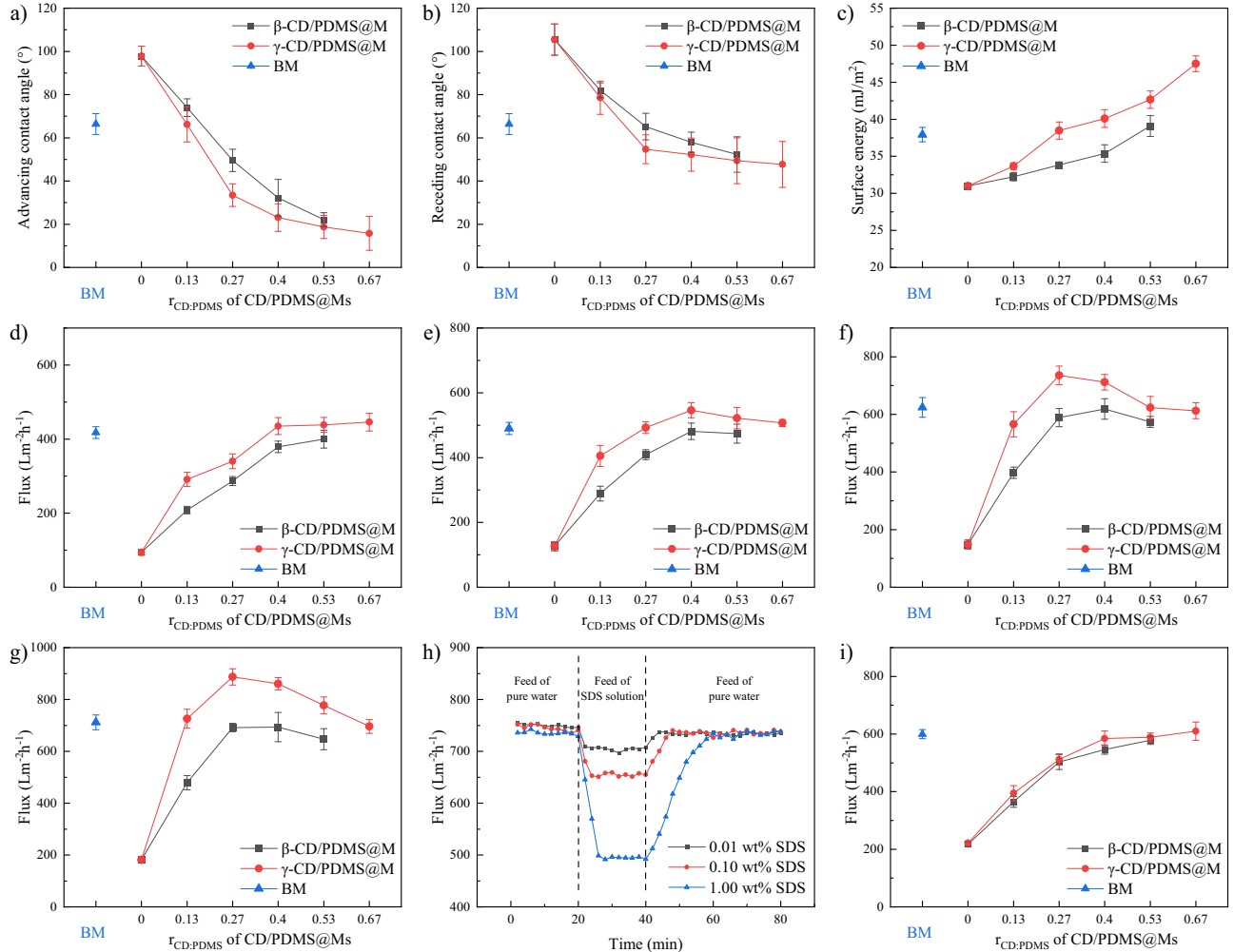

**Fig. 6 | Surface affinity and permeation flux of CD/PDMS@Ms. a** Advancing contact angles, **b** receding contact angles, and **c** surface energies of CD/PDMS@Ms. Initial water flux of CD/PDMS@Ms at **d** 5 °C, **e** 20 °C, **f** 35 °C, and **g** 50 °C. **h** Flux variation of γ-CD/PDMS@M-0.27 with the feed of SDS solution. **i** Flux of CD/PDMS@Ms with 1.00 wt% SDS feed at 35 °C. All experiments were conducted under a transmembrane pressure of 1.0 bar. Data are presented as mean ± SD ($n = 5$ for Fig. 6a–c; $n = 3$ for Fig. 6d–g and i).

## Antifouling performance of CD/PDMS@Ms

The antifouling performance of CD/PDMS@Ms was evaluated using dead-end stirred ultrafiltration experiments with conventional four steps: water feed filtration, foulant filtration, surface cleaning, and another water feed filtration. The flux decline rate (FDR) and flux recovery ratio (FRR) were used to quantify the antifouling capability of each membrane. FDR represents the ratio of the flux during foulant filtration to the initial pure water flux. A lower FDR value indicates that the actual operational flux of a membrane is close to the initial water flux of the membrane, implying higher operational efficiency. FRR represents the ratio of the pure water flux after fouling and cleaning to the initial pure water flux. A higher FRR suggests that the membrane flux remains stable after repeated use, indicating a longer membrane lifespan. As shown in Fig. 7a, both CD/PDMS@Ms showed antifouling performance that initially increased and then decreased with increasing $r_{CD:PDMS}$. For β-CD/PDMS@M, the optimal antifouling performance was achieved at $r_{CD:PDMS} = 0.4$, with a minimum FDR of 25.2% and a maximum FRR of 99.5%. For γ-CD/PDMS@M, the optimal antifouling performance was achieved at $r_{CD:PDMS} = 0.27$, with a minimum FDR of 14.2% and a maximum FRR of 99.7%. The antifouling performance of CD/PDMS@Ms showed a positive correlation with the supramolecular mobility of the loaded CD/PDMS PPRs, indicating that, as expected, the dynamic hydrophilic–LSE microdomains enhance the synergistic antifouling mechanisms by promoting the foulant detachment. Moreover, a proper ratio of hydrophilic to LSE microdomains facilitated this process by balancing the resistance and release mechanisms. Additionally, γ-CD/PDMS@M demonstrated significantly better antifouling performance than β-CD/PDMS@M, in agreement with their dynamic behavior differences. Both CD/PDMS@Ms exhibited bovine serum albumin (BSA) rejection rates of close to 99.3%, and these rejection rates were unaffected by the $r_{CD:PDMS}$ value (see Supplementary Fig. 10). This indicates that supramolecular antifouling modification does not compromise the rejection performance of the modified membranes.

The influence of operational conditions (feed pH, stirring speed, and temperature) on the antifouling performance of CD/PDMS@Ms is exhibited in Fig. 7b–d. Both CD/PDMS@Ms showed higher antifouling performance at pH values of 5, 9, or 11 compared to pH values of 3 and 7. At pH 5, the BSA isoelectric point mainly contributed to the good antifouling performance. At this pH, BSA maintains its intrinsic structure and exhibits a lower fouling propensity. At pH 9 and 11, the polarization effect of the hydroxyl groups on the outer CD wall became the key factor influencing membrane antifouling performance. The increased electron density around the oxygen atoms enhanced their ability to act as hydrogen bond acceptors, leading to intensified interactions with water molecules. Consequently, the aqueous CD

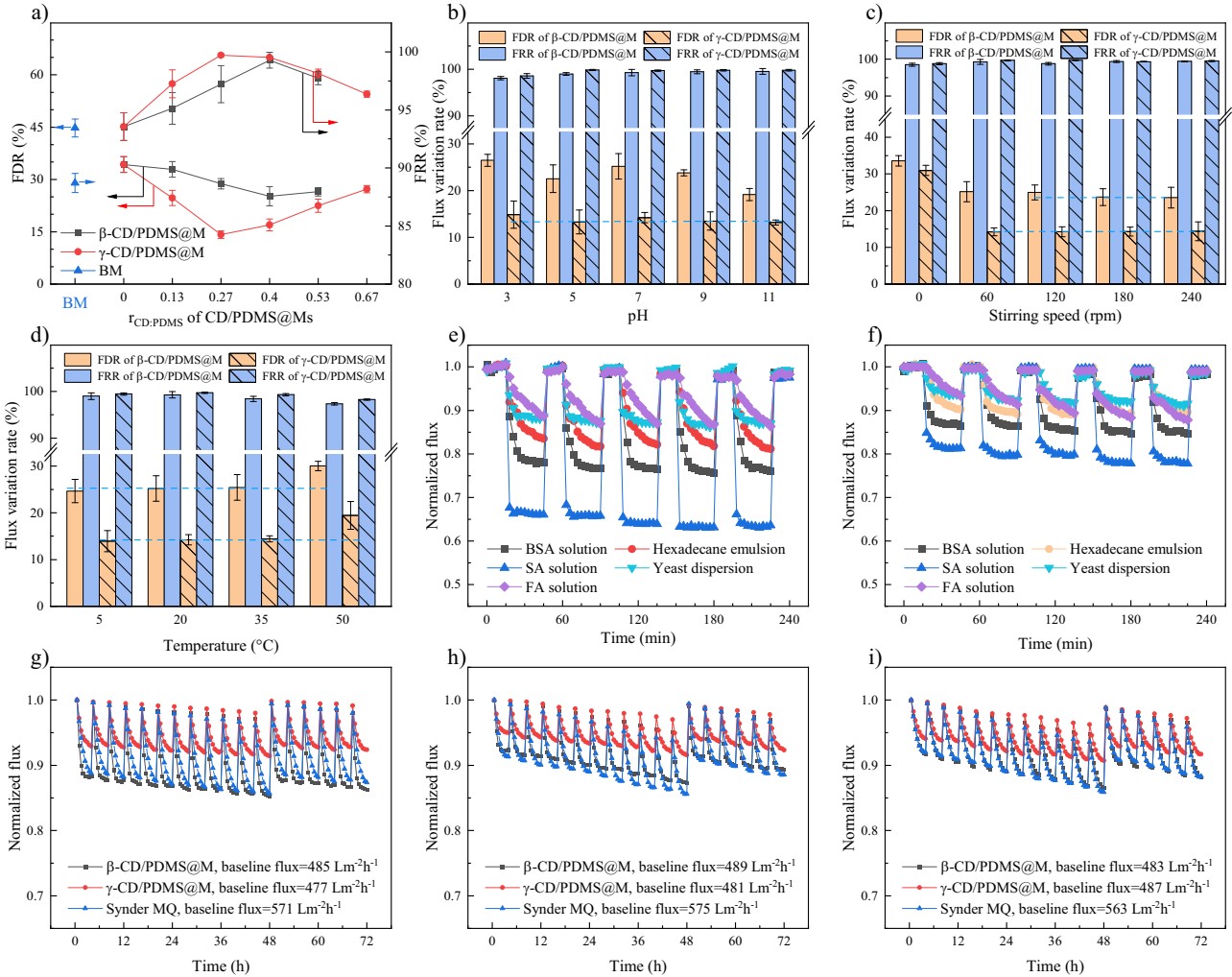

**Fig. 7 | Antifouling performance of CD/PDMS@Ms under different conditions.** **a** Influence of $r_{CD:PDMS}$, **b** feed pH, **c** stirring rate, and **d** temperature on antifouling performance of CD/PDMS@Ms in dead-end ultrafiltration experiment. Variations in membrane flux of **e** β-CD/PDMS@M and **f** γ-CD/PDMS@M over five filtration cycles with multiple foulants in dead-end ultrafiltration, the baseline flux corresponds to the pure water flux. Variations in membrane flux of CD/PDMS@Ms and Synder Filtration MQ membrane with feeds of (**g**) composite foulant solution, **h** coagulation-settled municipal wastewater, and (**i**) Lurgi gasification wastewater in

72-h cross-flow ultrafiltration, the baseline flux refers to the stabilized membrane flux after 30 min of operation. Unless otherwise stated, all experiments were conducted using BSA as the model foulant, with $r_{CD:PDMS}$ of 0.40 (β-CD/PDMS@M) and 0.27 (γ-CD/PDMS@M), at approximately pH 7, 20 °C, and 1.0 bar. Dead-end tests used 60 rpm stirring while cross-flow tests employed a tangential velocity of approximately 0.12 m/s when the membrane flux was 500 L·m⁻²·h⁻¹. Data are presented as mean ± SD ($n = 3$ for Fig. 7a–d).

activity was enhanced, which improved the membrane antifouling performance. The antifouling performance of both CD/PDMS@Ms improved with increasing stirring speed. Above 60 rpm, γ-CD/PDMS@M showed the highest antifouling performance, with an FDR of 14.2% and an FRR of 99.7%. Meanwhile, the antifouling performance of β-CD/PDMS@M remained stable at stirring speeds exceeding 120 rpm, with an FDR range of 22.9–23.7% and an FRR range of 99.5–99.7%. The difference in the optimal stirring rates for β-CD/PDMS@M and γ-CD/PDMS@M can be attributed to the lower motion resistance of γ-CD on the PDMS chains. This characteristic effectively overcomes the drawbacks of the high tangential flow typically required by conventional synergistic antifouling mechanisms. Thus, the dynamic characteristics of γ-CD/PDMS@M were enhanced even at lower stirring speeds. As the temperature increased from 5 °C to 35 °C, the antifouling performance of both CD/PDMS@Ms generally remained stable. However, when the temperature was further increased to 50 °C, the antifouling performance of both membranes declined. Across the entire temperature range, γ-CD/PDMS@M exhibited stronger antifouling performance than β-CD/PDMS@M. In conventional membrane processes,

antifouling performance tends to decrease with increasing temperature, which can be attributed to two factors. First, the thermal motion of foulant molecules increases with increasing temperature. Therefore, the movement and deformation of foulant molecules is accelerated, leading to the easy adherence of foulants to the membrane surface. Second, as membrane flux increases, the rate at which foulants accumulate on the membrane surface also increases. However, in this study, the antifouling performance of both membranes remained stable at temperatures below 35 °C. This stability can be attributed to the influence of temperature on the activity of the CDs. In other words, higher temperatures enhance the antifouling performance, which effectively counteracts the increased fouling tendency also observed at higher temperatures.

To investigate the broad applicability and multi-use stability of CD/PDMS@Ms, continuous five-cycle filtration experiments were conducted using BSA, emulsified hexadecane, sodium alginate (SA), yeast, or fulvic acid (FA) as model foulants (Fig. 7e, f). The results showed that γ-CD/PDMS@M consistently exhibited significantly lower FDR compared to β-CD/PDMS@M and nearly 100% FRR, indicating the

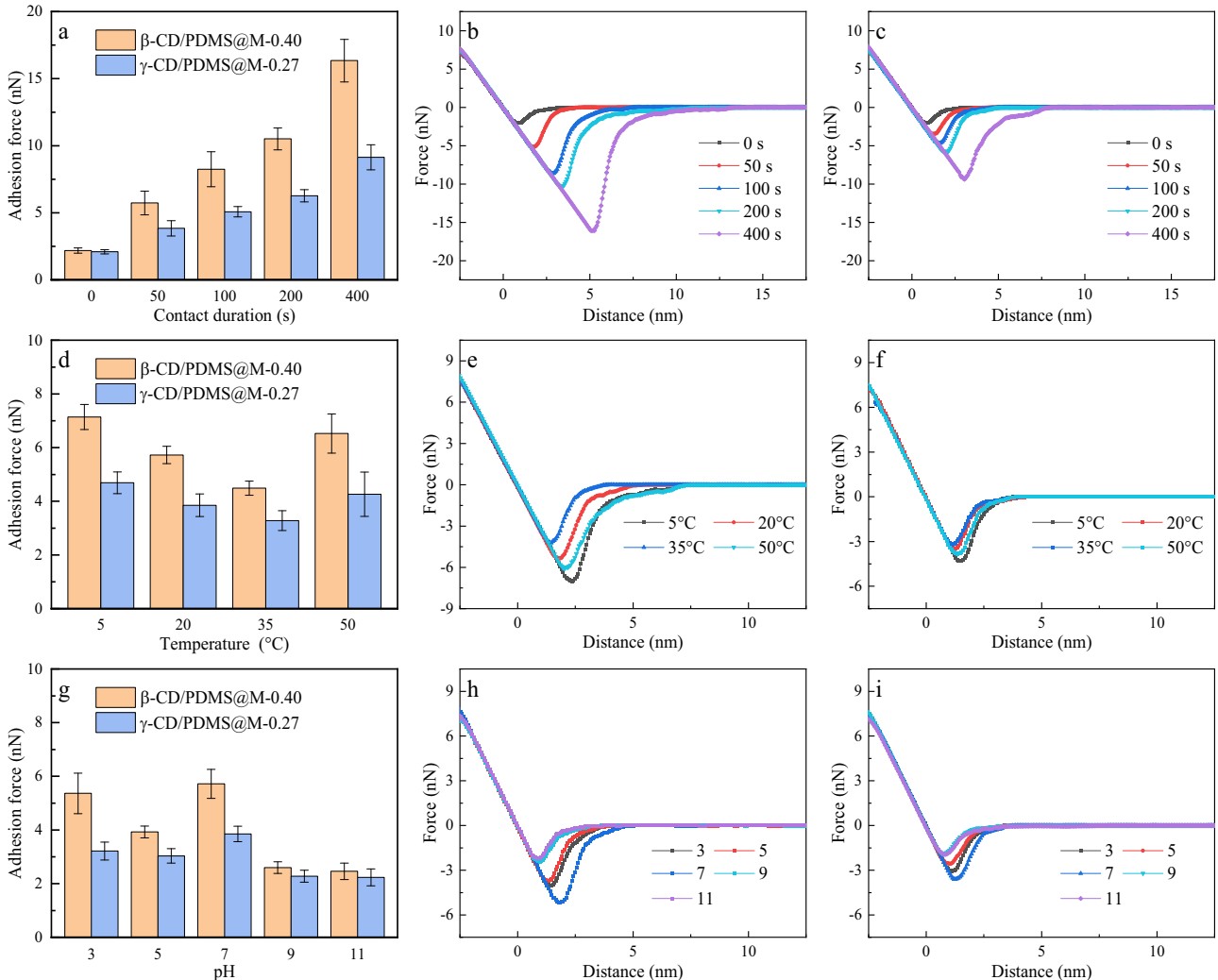

**Fig. 8 | Nanoscale analysis of BSA adsorption behavior on CD/PDMS@Ms by AFM. a** Adhesion force of BSA on CD/PDMS@Ms at different contact durations, **d** temperatures, and **g** pH values, with representative force curves shown for β-CD/ PDMS@M-0.40 (**b**, **e**, **h**) and γ-CD/PDMS@M-0.27 (**c**, **f**, **i**). Data are presented as mean ± SD ($n = 7$ for Fig. 8a, d and g).

strong generalizability of the dynamics-enhanced synergistic anti-fouling mechanisms. The high FDR caused by SA warrants attention. However, given the nearly 100% FRR after cleaning, it is speculated that this is due to the formation of a cake layer composed of gel-like structures over the membrane surface, an issue that cannot be resolved solely by enhancing the dynamic properties of the membrane surface.

To further evaluate the antifouling performance of the membrane under practical application conditions, a 72-h cross-flow ultrafiltration experiment was conducted. MQ (PES 50,000 Da) membrane from Synder Filtration, renowned for its good antifouling capability, was used as the control group, cut from an unused spiral-wound ultra-filtration membrane module. The feed solutions included one simu-lated composite foulant solution and three types of real water samples. The former was an equal-volume mixture of the five pollutant solu-tions/emulsions used in the dead-end five-cycle ultrafiltration experiment[35,36], while the latter consisted of coagulation-settled municipal wastewater, Lurgi gasification wastewater and untreated Songhua River water (Fig. 7g–i and Supplementary Fig.11). The experiment included periodic cleaning every 4 h using deionized water with both backwashing and forward flushing, while the first cleaning after the 48th hour was replaced with chemical cleaning. The results showed that γ-CD/PDMS@M maintained remarkable antifouling

performance when exposed to both composite and real foulants. The FDR after 4 h of continuous operation was only 3–7%, and FRR after deionized water cleaning was nearly 100%. The membrane flux at the 12th cycle (before chemical cleaning) remained at over 97% of its initial value, and chemical cleaning restored the flux to its original level. The antifouling performance of β-CD/PDMS@M was slightly inferior to that of γ-CD/PDMS@M, with a flux decay rate of 6–12% after 4 h of continuous operation and a membrane flux of over 94% of its initial value at the 12th cycle. The antifouling performance of the Synder Filtration MQ membrane was comparable to that of β-CD/PDMS@M, with the latter showing a somewhat higher flux recovery rate under hydraulic cleaning conditions. Compared to membranes in other stu-dies, including heterogeneous and hydrophilic membranes, the dynamics-enhanced synergistic antifouling mechanisms demon-strated significant advantages in maintaining high flux and effectively resisting and releasing composite foulants (Supplementary Tables 11 and 12).

### Dynamics-enhanced antifouling mechanisms

The influence of contact duration, temperature, and pH on the inter-action forces between BSA and CD/PDMS@Ms was assessed by atomic force microscopy (AFM) using a specialized probe carrying BSA-modified SiO₂ microspheres (see Supplementary Fig. 12 for the

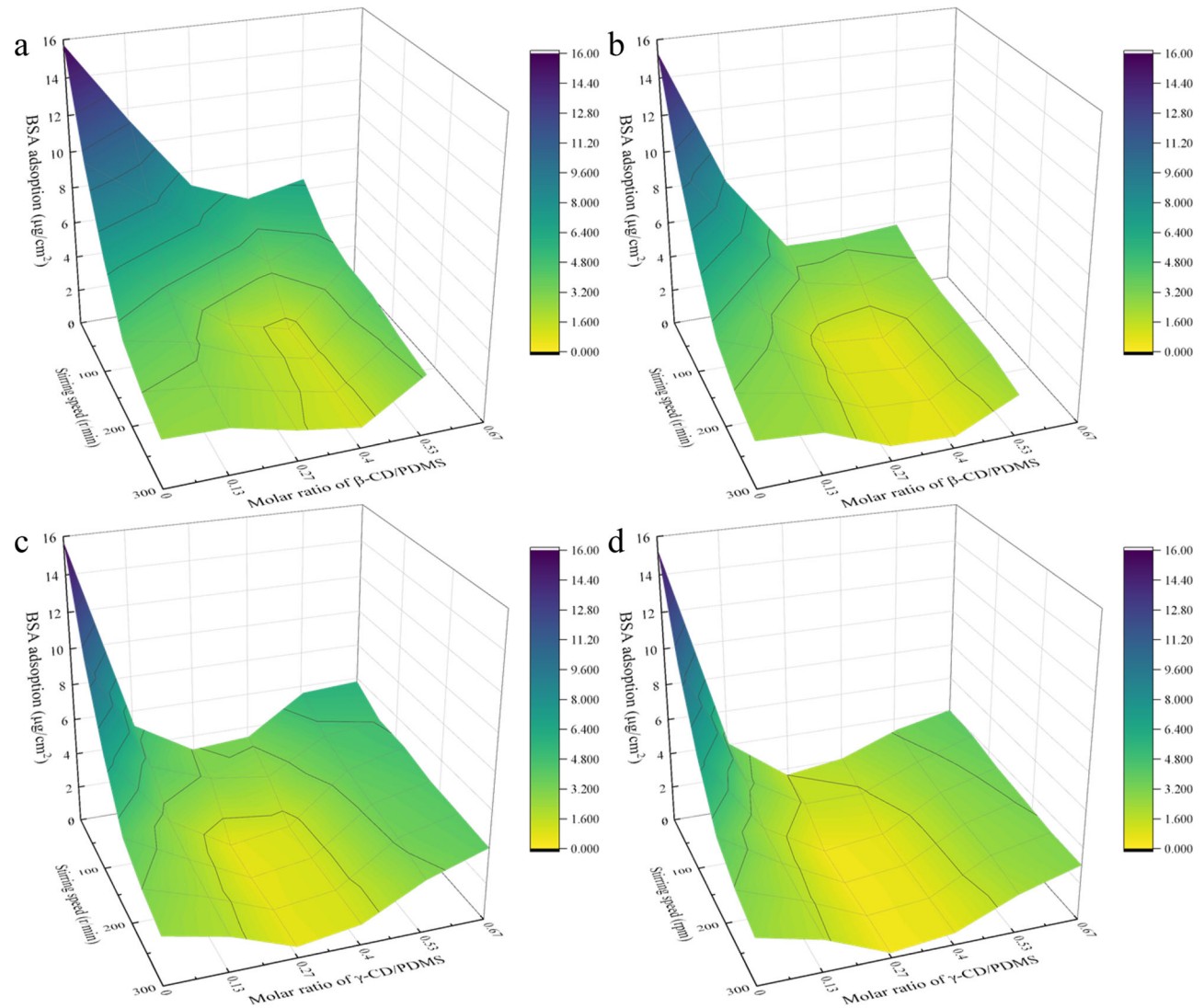

**Fig. 9 | Macroscopic analysis of BSA adsorption on CD/PDMS@Ms.** Quantification of BSA adsorption on β-CD/PDMS@M at **a** pH 7 and **b** 11, and on γ-CD/PDMS@M at pH **c** 7 and **d** 11.

microstructure). The adhesion force of BSA on both CD/PDMS@Ms generally increased with increasing contact time within the range of 0–400 s (Fig. 8a–c). However, the magnitude of this adhesion force increase on γ-CD/PDMS@M was only about half of that observed on β-CD/PDMS@M. Therefore, in the absence of tangential flow, the favorable supramolecular modifier dynamics provide resistance to foulant adsorption solely through Brownian motion. The BSA adhesion force on CD/PDMS@Ms decreased as the temperature increased from 5 °C to 35 °C but increased when the temperature was further increased from 35 °C to 50 °C (Fig. 8d–f). This is due to the higher temperatures enhancing the water activity. Both the dynamic properties of supramolecules and the fouling propensity of foulants increase with increasing water activity. Below 35 °C, the supramolecule dynamic properties dominate, while above 35 °C, the fouling propensity of the foulants becomes the dominant factor. This trend slightly differs from the effect of temperature on the FRR of the membranes because the influence of increased membrane flux on fouling propensity is excluded. The BSA adhesion forces observed under different pH conditions are depicted in Fig. 8g–i. At the isoelectric point and under alkaline conditions, CD/PDMS@Ms exhibited reduced adhesion forces. This trend is consistent with the experimental results on antifouling performance.

To further explore the antifouling mechanisms, BSA adsorption experiments on CD/PDMS@Ms were conducted under different pH values and stirring speeds (Fig. 9). Under the same pH, γ-CD/PDMS@M showed lower BSA adsorption than β-CD/PDMS@M. The lowest BSA adsorption of β-CD/PDMS@M was achieved at $r_{CD:PDMS} = 0.4$, while for γ-CD/PDMS@M, the minimum BSA adsorption occurred at $r_{CD:PDMS} = 0.27$. Under alkaline conditions, both CD/PDMS@Ms exhibited reduced adsorption. Notably, at $r_{CD:PDMS} = 0.27$ or 0.40, γ-CD/PDMS@M maintained low BSA adsorption even under minimal or no stirring. This effect was particularly noticeable at pH 11. These findings demonstrate that the supramolecular dynamics-enhanced synergistic antifouling mechanisms of the modified membranes operate effectively without the need for high tangential flow, unlike conventional ones.

## Discussion

This study reports supramolecular dynamics-enhanced synergistic antifouling mechanisms to achieve enhanced membrane antifouling performance while maintaining good membrane permeability. Hydrophilic CDs and LSE PDMS were utilized to prepare antifouling modifiers consisting of threaded, ring-like supramolecules. The CDs and PDMS exhibit relative motion, including sliding, rotation, and

vibration. Thus, dynamic hydrophilic-LSE heterogeneous microdomains are formed on the membrane surface. These dynamic microdomains maintain good fouling release efficiency even at very low or zero tangential flow rates while also improving membrane flux. This approach overcomes the two major limitations of traditional heterogeneous microdomains: their reliance on high tangential flow and low initial water flux. The prepared CD/PDMS@Ms exhibit enhanced overall performance compared to previously reported heterogeneous membranes. Moreover, the use of β-CD and γ-CD was compared and different $r_{CD:PDMS}$ were investigated to reveal the positive correlation between microdomain dynamics and enhanced antifouling/flux improvement. The effects of temperature and pH on membrane performance were also analyzed, with a temperature of 35 °C and alkaline conditions shown to further enhance performance. These findings offer a strategy for designing water treatment membranes with synergistic antifouling mechanisms.

## Methods

The materials and reagents used in this study and common characterization methods for the membrane materials are described in detail in the supplementary methods.

### Preparation of CD/PDMS PPRs

In the targeted β-CD/PDMS PPRs, the desired $r_{CD:PDMS}$ values of 0.13, 0.27, 0.40, and 0.53 were achieved by oversupplying β-CD by 0.15, 0.19, 0.22, and 0.25 times the theoretical amounts required for these respective ratios. The specific preparation steps of β-CD/PDMS PPRs involved adding PDMS (263.3, 122.5, 80.6, and 58.4 mg) to a near-saturated β-CD solution in DMF (10 mL, 59.0 mg/mL). The mixture was first subjected to ultrasonic agitation for 15 min, then slow shaken at room temperature for 7 days and last allowed to stand for 2 days. After standing, the mixture was subjected to rotary evaporation to remove 90 wt% of DMF, followed by the addition of an equal mass of water (equivalent to the removed DMF) to induce precipitation of β-CD/PDMS PPRs. The resulting precipitates were sequentially washed with cyclohexane and water, and then dried under vacuum.

In the targeted γ-CD/PDMS PPRs, the desired $r_{CD:PDMS}$ values of 0.13, 0.27, 0.40, 0.53, and 0.67 were achieved by oversupplying γ-CD by approximately 0.10, 0.14, 0.17, 0.19, and 0.20 times the theoretical amounts required for these respective ratios. The specific preparation process for γ-CD/PDMS PPRs involved adding PDMS (751.7, 349.2, 229.7, 170.4, and 133.7 mg) to a near-saturated γ-CD solution in water (10 mL, 184.2 mg/mL). The mixture was first subjected to ultrasonic agitation for 15 min, followed by slow shaking at room temperature for one day, and finally allowed to stand overnight. This resulted in a spontaneous gel-like precipitation of the γ-CD/PDMS PPRs. The resulting precipitates were sequentially washed with cyclohexane and water, and then dried under vacuum.

### Loading of CD/PDMS PPRs on membrane surface

The BM was fabricated using a combination of non-solvent induced phase separation combined with surface segregation techniques, aiming to achieve ultrafiltration-level pore sizes and high porosity. A casting solution containing 14 wt% PES, 7 wt% PVP, 7 wt% Pluronic F127, and 72 wt% DMF was stirred at 70 °C for 4 h, then left to stand for an additional 4 h to release any trapped bubbles. The solution was cooled to 25 °C and cast onto a glass pane, forming a 250 μm thick liquid film. The film was subsequently immersed in a coagulation bath of 80 wt% water and 20 wt% DMF for 5 min, allowing the solidification of the BM. After thorough rinsing, the membrane was stored in deionized water for at least 72 h before use.

The process of loading CD/PDMS PPRs onto the BM surface consisted of three main steps, as depicted in Supplementary Fig. 13. The first step was the grafting of GMA onto the membrane surface to form linear PGMA. The BM was coated with a 2.00 wt% GMA aqueous

solution, devoid of oxygen, at a rate of 100 μl/cm². It was then subjected to UV light (surface intensity 5.0 mW/cm²) for 10 min under nitrogen protection to initiate radical polymerization. Following this, the membrane was washed with deionized water to remove unreacted GMA. The second step was the reaction between CD/PDMS PPRs and GMA. The PGMA-grafted membrane was fixed at the bottom of a beaker, and a clear CD/PDMS PPR dispersion (2 g/L PPRs, approximately 50 wt% DMF and 50 wt% water) was poured in, with 10 mL of dispersion corresponding to each square centimeter of membrane surface area. Notably, the PPR dispersion was prepared by first dissolving PPRs in DMF before diluting with water, ensuring effective dispersion. When only PDMS is grafted, cyclohexane is used as the solvent. The beaker was left standing at room temperature for 12 h. During this step, the PPRs or PDMS were chemically linked to PGMA via an addition reaction between the epoxy groups on the side chains of PGMA and the terminal amines of PDMS. The third step was the capping of CD/PDMS PPRs. The membrane was immersed in a 2.5 mL/cm² FITC solution (5 mg/mL) for 3 h to cap residual aminos. After the capping process, the CD within the PPRs would not leach out. Before use and characterization, the membrane underwent a 24-h deionized water soaking period, with water changes every 8 h.

### Molecular simulation

The investigation of CD/PDMS PPR formation propensity and CD activity was conducted using the Forcite module in Materials Studio software. The simulation boxes were constructed using the COMPASS force field at 298 K, and the initial structure was energy-optimized before molecular dynamics simulations. The simulations were carried out in two stages: An NPT ensemble was applied for 1000 ps to achieve density equilibrium, followed by an NVT ensemble for an additional 1000 ps to reach motion equilibrium.

For the study of CD/PDMS PPR formation, the simulation box included 200 solvent molecules (water, DMF, or ethanol), one CD molecule, and a 14-mer PDMS chain. The ΔG was calculated before and after the formation of the CD/PDMS PPR to assess whether the formation process was spontaneous. The solvent molecules were considered in this simulation to represent the realistic system conditions.

To evaluate the migration behavior of CD along the PDMS chain, a separate simulation was conducted without solvent molecules to improve computational efficiency and ensure reproducibility. Since the dominant interactions in this system are van der Waals forces and steric hindrance, the key characteristics of CD migration along the PDMS chain are expected to be retained even in the absence of solvent. The simulation involved a single CD molecule and a 10-mer PDMS chain. The PDMS chain was configured in a helical conformation aligned along the z-axis of a 3D coordinate system, and it was fixed in place with its geometric center constrained on the z-axis. The CD molecule was translated stepwise along the PDMS chain, with each step set to one-tenth of the projected z-axis length of a single PDMS monomer unit. The migration consisted of two stages: in the first stage, the CD moved toward the direction of its larger aperture for a total distance equivalent to two PDMS monomer units along the z-axis projection; in the second stage, the CD reversed direction and moved toward its smaller aperture for the same total distance. A time step of 1 fs was used, and structural optimization was performed after each step to ensure system relaxation. The ΔG of the system was recorded before and after each step, resulting in 21 data points in each direction. By analyzing the ΔG profile throughout the CD migration process, the resistance encountered by the CD can be inferred.

### Characterization of CD's activity on PPRs by QELS

QELS experiments were carried out using a Nano ZS90 instrument from Malvern Instruments, UK, at a fixed angle of 90°. The instrument utilized a 22 mW He–Ne laser (wavelength 632.8 nm) for the incident beam. β-CD/PDMS and γ-CD/PDMS PPRs with different $r_{CD:PDMS}$ were

dissolved in a 2.0 wt% DMF solution. Time-intensity correlation functions were recorded for 60 s at 30 °C, followed by CONTIN analysis to determine the diffusion coefficient, $D(m^2s^{-1})$.

## Dead-end ultrafiltration experiment

To evaluate the permeability flux, rejection, and antifouling performance of the membranes, ultrafiltration experiments were conducted with a dead-end stirred cell filtration system driven by pressurized nitrogen gas. The experiments were carried out at a controlled temperature of 20 °C. The filtration cell (Amicon Stirred Cell 8200, Millipore Co.) had an inner diameter of 63.5 mm and a volume capacity of 200 mL. The detailed operation consisted of four consecutive steps. The first step was the evaluation of pure water flux, in which the flux was continuously measured with the feed solution of pure water at a transmembrane pressure of 1.0 bar. After keeping constant for 15 min, the pure water flux ($J_{w1}$, $Lm^{-2}h^{-1}$) was calculated by the Eq. (1):

$$J = \frac{V}{At} \qquad (1)$$

where $V$ is the volume of permeated water (L), $A$ is the membrane area (m²), $t$ is the operation time (h). The second step was the evaluation of flux under fouling, in which the flux ($J_p$, $Lm^{-2}h^{-1}$) was measured with the feed solution of BSA (1 g/L, hydrochloric acid (HCl) and sodium hydroxide (NaOH) solution were used to regulate the pH to approximately 7, sodium chloride (NaCl) was added to maintain the ionic strength I at approximately 0.01) at a transmembrane pressure of 1.0 bar and a stirring speed of 60 rpm for 30 min. The rejection ($R$) was calculated by the Eq. (2):

$$R = \left(1 - \frac{C_p}{C_f}\right) \times 100\% \qquad (2)$$

where $C_p$ and $C_f$ are the BSA concentration in permeate and feed solutions, respectively, which were measured by a Thermo Scientific Evolution 300 UV/Vis spectrophotometer (USA). The third step was the cleaning of membrane, in which the membrane was rinsed with shear-flow water on the membrane surface at a stirring speed of 480 rpm for 15 min. The temperature of the cleaning water was also maintained at 20 °C to ensure consistency throughout the experiment. The last step was the evaluation of the flux recovery, in which the flux ($J_{w2}$, $Lm^{-2}h^{-1}$) was measured with the feed solution of pure water at a transmembrane pressure of 1.0 bar for 15 min. In order to evaluate the antifouling performance quantitatively, the FDR and FRR were defined as below[37]:

$$FDR = \frac{J_{w1} - J_P}{J_{w1}} \times 100\% \qquad (3)$$

$$FRR = \frac{J_{w2}}{J_{w1}} \times 100\% \qquad (4)$$

In studying the effects of environmental conditions on membrane processes, HCl, NaOH, and NaCl were used to regulate the pH of BSA feed solution to approximately 3, 5, 7, 9, 11, and the ionic strength of BSA solution I to approximately 0.01. The stirring speed and temperature of cell filtration system was regulated by the intelligent constant temperature digital display magnetic stirring sleeve. The stirring speed values included 0, 60, 120, 180, and 240 rpm while the temperature values included 5 °C, 20 °C, 35 °C, and 50 °C.

In the five-cycle ultrafiltration experiments, the fourth step of each cycle served as the initial step for the subsequent cycle. Each membrane was evaluated through a minimum of three independent trials, with the median value being adopted for analysis. Model

foulants included hexadecane, yeast suspension, BSA, SA, and FA. All model foulant solutions were prepared at 0.1 wt% concentration. The cetane-water emulsion contained an additional 0.01 wt% SDS as an emulsifying agent. The BSA solution was pH-adjusted to approximately 7 using HCl and NaOH solution, with ionic strength maintained at I ≈ 0.01 through NaCl addition. The FA solution was treated with a minimal volume of NaOH solution to achieve pH at approximately 7.5, thereby enhancing its solubility. No pH adjustments were made to the hexadecane emulsion, yeast suspension, or SA solution, which exhibited natural pH values of 7.4, 5.9, and 7.7, respectively. All the solutions and emulsions were treated with ultrasonic (350 W/500 mL) for 20 min before use.

## Cross-flow ultrafiltration experiment

To further evaluate the antifouling performance of the membrane under practical application conditions, a cross-flow ultrafiltration experiment was conducted. The membrane module was square-shaped, with an effective membrane size of 15 cm × 15 cm. The driving pressure was provided by a variable frequency centrifugal pump, maintained at a constant 1.0 bar using an electronic pressure gauge and a PLC module. The temperature was controlled at 20 °C using a digital thermostatic jacket. The water recovery rate was maintained at about 92%. Under pure water conditions, the tangential flow velocity on the membrane surface was approximately 0.12 m/s when the transmembrane flux was around 500 L·m⁻²·h⁻¹.

The feed solutions included one simulated composite foulant solution and three types of real water samples. The composite foulant solution was an equal-volume mixture of the five pollutant solutions/emulsions used in the dead-end five-cycle ultrafiltration experiment, with a total concentration of approximately 0.1 wt%. After mixing, the solution was ultrasonicated for 60 min (350 W/500 mL) before use to ensure uniformity. The first real water sample was municipal wastewater treated by coagulation and sedimentation, with 39.7 mg/L suspended solids, 6.72 NTU turbidity, 117.5 mg/L COD (dichromate method), 43.9 mg/L TOC, and a pH of 7.6. The second real water sample was Lurgi gasification wastewater treated by coagulation and sedimentation, with 59.9 mg/L suspended solids, 3130 mg/L COD (dichromate method), 1397 mg/L TOC, and a pH of 8.1. The third real water sample was untreated Songhua River water, with 55.1 mg/L suspended solids, 5.72 NTU turbidity, 26.2 mg/L COD (dichromate method), 10.25 mg/L TOC, and a pH of 7.3.

Each cross-flow experiment was conducted for 72 h, with cleaning performed every 4 h using deionized water. The cleaning procedure included one backwashing cycle at 1.5 bar for 3 min, followed by one forward non-transmembrane rinsing cycle at 1.0 bar for 2 min (cleaning durations were not included in the 72-h experiment time). After 48 h, the first cleaning was replaced with chemical cleaning. This involved backwashing with a 1000 mg/L NaOH solution and a 2000 mg/L sodium hypochlorite solution, followed by backwashing with a 500 mg/L oxalic acid solution and a 1000 mg/L HCl solution. The cleaning pressure was maintained at 1.5 bar for 4 min in each case, followed by one forward non-transmembrane rinsing cycle at 1.0 bar for 2 min.

## Nanoscale analysis of BSA adsorption

AFM was employed to measure the adhesive force of BSA on membrane surfaces using the Bruker Dimension Icon system (Germany). Custom colloidal probes were fabricated by attaching silica microspheres to Bruker NP-O tipless cantilevers (Germany), supplied by Wenzhou Lingpian Technology Co., Ltd. (China). The probes were then immersed in a BSA solution to generate BSA-modified SiO₂ microspheres. Interaction forces between BSA and the membrane surface were measured in Force Profile Mode under various contact times, temperatures, and pH conditions. For quality control, seven measurement points were selected on each membrane sample within a flat,

circular region: six evenly spaced points along the circumference and one at the center. The adhesive force was calculated as the peak force difference between the approach and retraction curves.

## Macroscopic analysis of BSA adsorption

BSA adsorption was measured using a custom cylindrical container with a membrane holder at the bottom and sidewalls coated with an LSE anti-adsorption layer. Only the upper surface of the membrane (effective diameter: 10 cm; area: 78.5 cm²) was exposed to the solution. A magnetic stir bar (length: 9.6 cm; height: 0.5 cm) was placed 0.2 cm above the membrane surface. The container was initially filled with 300 mL of a 30 mg/L BSA solution, with the pH adjusted to approximately 7 or 11 using HCl or NaOH solution, and the ionic strength set to about 0.01 M by adding NaCl. Before starting the formal adsorption experiment, the solution was stirred at varying speeds (0–300 rpm) for 10 min to pre-wet the container. During the 30-min adsorption test, the membrane was installed and a fresh 200 mL BSA solution (30 mg/L) was added under identical pH and ionic conditions. After the experiment, the residual BSA concentration was determined using a Thermo Scientific Evolution 300 UV/Vis spectrophotometer (USA), and the difference between the initial and final BSA concentrations was used to calculate the total amount of BSA adsorbed on the membrane surface.

## Data availability

All data supporting the findings of this study are available within the article, the Supplementary Information file, the Supplementary Movie files, and the Supplementary Data file. Additional raw data and details of the analysis procedures are available from the corresponding authors upon request. Source data are provided with this paper.

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

## Acknowledgements

The authors gratefully acknowledge financial support from National Key Research and Development Program of China (Grant No. 2024YFC3214700; D.L.), National Natural Science Foundation of China (Grant No. 52100083; M.H.), China Postdoctoral Science Foundation (Grant No. 2023T160169; M.H.) and State Key Laboratory of Urban Water Resource and Environment (Harbin Institute of Technology) (Grant No. 2024TS23; M.H.).

## Author contributions

M.H., D.L. conceived the research idea and designed the study. M.H., Y.H., M.W., J.Y., and T.W. performed the experiments and collected the data. J.M. supervised the project and provided valuable insights into data interpretation. All authors contributed to data analysis, discussion, and manuscript preparation.

## Competing interests

The authors declare no competing interests.
