## [Transparent Peer Review file · Nature Communications]

Supramolecular Dynamics-Enhanced Synergistic Antifouling Mechanisms for Enhanced Membrane Antifouling and Permeability

Corresponding Author: Professor Dongwei Lu

Version 0:

Reviewer comments:

Reviewer #1

(Remarks to the Author)

[Note from Editor Please see attached file]

Reviewer #2

(Remarks to the Author)

This manuscript describes the coating of membranes with a layer of a PDMS-cyclodextrin pseudo-poly-rotaxane (PPR) to decrease fouling. The authors claim the reported fouling resistance arises from the high mobility of PDMS.

The paper reports the formation of PPRs in detail with good data, though these PPRs have been prepared before in other contexts as shown in some of the later citations in the paper. They then graft the membrane surface with epoxide groups and attach the PPR to the membrane surface from a suspension. The paper includes some membrane fouling data as well as protein adsorption and AFM experiments. While the fouling resistance is good, it is not really exceptional compared with other materials and coatings in the literature, many prepared using more scalable methods and tested under more stringent conditions. As such, I do not believe this manuscript offers sufficient novelty and significance for publication in this journal. It may be a good fit in a somewhat more specialty journal. Beyond this, it also has some significant limitations related with the description of methodology and potential issues with membrane testing methodologies in demonstrating their claims (see below). More detailed comments are listed below.

- A major problem is that the methodology in the main manuscript and SI does not include some critical pieces of information that would be needed for the work to be reproduced, and at times for the quality of the methodology to be evaluated. For example, there is no information on what the PDMS molecule is (molar mass, end groups, manufacturer), how the concentrations and conditions for PPS formation and coating were selected, or key details of how the PPS was deposited on the membrane (e.g. swatch size used, immersion geometry, etc). The dispersion from which this was done is not characterized. The base membrane preparation is described, but not well-characterized. The description of filtration experiments is somewhat cursory. Fouling compositions are not documented. These are significant issues regarding publication.

- Figure 1 is not really informative; it is unclear what anything really represents. The cartoons make mechanistic claims that are not backed up by data.

- The surface SEMs imply rough, uneven coatings. It is unclear if there is full coverage of the PES surface or not, though that seems unlikely.

- Cross-sectional SEM images are required to show the coating thickness and structure, and to determine if there is deposition inside membrane pores.

- How were the compositions in Figure 5 g-h calculated? If XPS survey elemental compositions were used, were they calibrated with correct materials? Or were high-resolution scans (e.g. C1s) used and if so, how was quantification performed?

- The data shows very little PDMS deposited on the surfaces versus how much was used for the PPR. It seems the coatings

are almost all CD. How is this explained?

- The permeances of several coated membranes appear to be higher than the value for the uncoated base membrane. How is this possible?
- There is no measurement or report of any membrane selectivity measure (e.g. rejection, MWCO). This is important to contextualize the reported permeances, and also to determine if coating changes this capability.
- There are no fouling data reported for the base membrane or the epoxy-only coated membrane. This is important to show a baseline.
- How stable is the coating? I noted the fouling experiments were done at only 60 rpm stirring, max 240 rpm, which is very slow. Did the coating delaminate at higher stir speeds?
- The fouling experiments reported are extremely short, only 10 minutes of exposure to the foulant between washes. This is not realistic. Importantly, the AFM data on Figure 8 indicate longer contact results in stronger adhesion to the foulant - so longer exposures can lead to major changes to the reported results.
- As mentioned earlier, several other researchers have reported similar or better fouling resistance performance, often using amphiphilic copolymers. I do not know why these are not included in the comparison table in the SI, though it appears cited manuscripts are mostly from a smaller subset of groups/institutions. Some other papers that have reported comparable or higher FRR values, often under more challenging fouling conditions include:

<https://www.sciencedirect.com/science/article/abs/pii/S0376738807002669>

<https://www.sciencedirect.com/science/article/abs/pii/S0376738818303727?via=ihub>

<https://pubs.acs.org/doi/abs/10.1021/acsami.0c03075>

<https://www.sciencedirect.com/science/article/abs/pii/S0376738817314485>

<https://pubs.acs.org/doi/abs/10.1021/acsami.7b04884>

Reviewer #3

(Remarks to the Author)

The “resistance-release synergistic antifouling mechanism” based on hydrophilic–low surface energy heterogeneous structure is an excellent strategy for antifouling applications. However, its implementation on membrane surfaces faces two major bottlenecks: low initial water flux and reliance of antifouling performance on intense tangential flow. This study introduces a heterogeneous modification of the membrane surface using pseudo-polyrotaxane (PPR) composed of hydrophilic cyclodextrin (CD) and low-surface-energy polydimethylsiloxane (PDMS). Leveraging the dynamic nature of PPRs, the study effectively addresses the two key challenges of the synergistic antifouling mechanism, demonstrating significant innovation and practical value. It is recommended for publication in Nature Communications. The following suggestions are provided to refine and optimize the manuscript:

1. The figure mentions “against fine particulate foulants” and “against coarse particulate foulants,” but corresponding descriptions are missing in the text. Either revise the figure labels or supplement the text with the relevant descriptions.
2. PPRs exhibit three types of dynamics: sliding, rotating, and vibrating. This study focuses solely on sliding dynamics through molecular simulations and quasi-elastic light scattering. While the results show that the sliding dynamics of β -CD/PDMS PPRs are weaker than those of γ -CD/PDMS PPRs, does this necessarily imply that the overall dynamics of β -CD/PDMS PPRs are also weaker? Or is sliding dynamics the only critical factor influencing membrane performance? Please clarify.
3. How was fluorescence intensity used to determine the degree of single-ended and double-ended grafting of PPRs on the membrane surface? The current description is confusing. Please provide a detailed explanation in the text.
4. In the section Separation performance of CD/PDMS@Ms, systematically analyze the differences in antifouling performance between the two membranes for various pollutants, and discuss the underlying mechanisms.
5. In the section Loading of CD/PDMS PPRs on membrane surface, it is stated that the solvent for the PPR is approximately 50 wt% DMF and 50 wt% water. However, in an extreme case where the PPR contains no CD (i.e., the modifier consists solely of PDMS), DMF and water are unable to dissolve PDMS. Please address this discrepancy.
6. In the same section, some experimental steps only provide the concentration and volume of the modification solution but omit the membrane area submerged in the solution. Please specify that the solution was used in significant excess or provide the amount of solution used per unit membrane area.
7. In this study, the dead-end stirred cell filtration system is used to evaluate membrane performance. A long-term filtration experiment using cross flow equipment should proceed to assess its potential in industry application.

Version 1:

Reviewer comments:

Reviewer #1

(Remarks to the Author)

The authors have satisfactorily addressed most of my comments on the original manuscript. The following are some minor questions/suggestions for their reference.

1. I was not able to find the methodology for grafting CD/PDMS to membrane surface, either in the manuscript or in the

supplementary materials. Please add it.

2. I still feel that the description on the methodology of Molecular (Dynamics) Simulation is not sufficient. Please add more details.

3. Please make clear how the system energies reported in Supplementary Table 1 were obtained.

Reviewer #2

(Remarks to the Author)

I would like to thank the authors for their responses. The long-term fouling studies add to the study. The responses regarding some of the details for XPS, experimental details for coating, and other requests, which were echoed by other reviewers, are appreciated.

However, I still have some concerns about this paper and its true novelty and impact. Please see below for further comments:

- I still do not believe that this manuscript shows any data or justification regarding the mechanisms of fouling resistance they claim quite strongly throughout the manuscript. The image in Figure 1 is still unclear in what it means to me - what is meant by "dynamic source"? There is no clear data showing the tangential flow is really changing the movement of CDs. The anti-fouling mechanism with a running CD (I think) pushing off a foulant is also unreasonable. At a basic level, a macromolecular foulant like BSA is much bigger than CD (by an order of magnitude); there is no pushing off involved in fouling resistance. There are, indeed, some studies on how the high molecular mobility of PDMS can contribute to its self-cleaning capabilities, but it is not this type of mechanism.

- As the authors note, heterogeneous surfaces have been studied as fouling-resistant materials. At a minimum, CD is indeed quite hydrophilic (more than the base membrane or PDMS). I noticed that the authors created a separate table in the SI with the references I provided and claimed these are "hydrophilic" homogeneous materials. I suggest the authors read these papers more carefully as these papers, from multiple groups, all include copolymers of various architectures that combine hydrophilic and hydrophobic/LS units. For instance, reference 21 combines the hydrophilic MPC and SBMA with highly hydrophobic/LS trifluoroethyl methacrylate (TFEMA) groups. Reference 23 combines zwitterions (hydrophobic) with PES units. Reference 24 combines PEO and other charged groups with TFEMA. As such, these references should not be deemed outside of what should be compared with the manuscript described here. They are clear prior demonstration of the ability of multiple heterogeneous surfaces that combine hydrophilic and hydrophobic groups to resist fouling, with FRR values comparable to or better than reported in this document. It is important to acknowledge this, in my personal opinion.

- As I read carefully, the reported CD/PDMS ratios are extremely high, particularly given how much larger a CD molecule is in comparison with a PDMS repeat unit. From figures S6 and S7, you can see that when one CD molecule is placed on a PDMS chain, it covers 3-4 repeat units within its core. In this light, CD:PDMS molar ratios between 0.27-0.67 do not look reasonable, and definitely do not appear/act like a clothesline as far as I can imagine. Even if the units were narrower, a CD:PDMS ratio of 0.67 would be more than every other repeat unit. This really needs to be addressed in the document, as it is in contrast with the proposed mechanisms.

- I am still unclear about the XPS data, and the mismatch between CD:PDMS ratios vs the "molecular coverage" ratios from XPS. The authors claim, in their response, that "For example, in the case of γ -CD/PDMS@M-0.40, the membrane was tested twice after preparation. The molecular coverage ratio of CD to PDMS was 6.98 in the first test and 6.68 in the second test, with an average of 6.83. This corresponds to an rCD:PDMS of 0.41.". I do not understand how this works, honestly.

- About the membrane formation process and flux changes: The authors immerse the membrane into a DMF/water mixture for coating. The base membrane material is soluble in DMF and will likely swell heavily in this mixture. Therefore, it is crucial to include a base membrane immersed in DMF/water with no polymer in it as a control at least for its effect on water flux, as it can cause increased flux. The same is needed for the epoxy-modified membrane, which will make it more hydrophilic. Indeed, the fact that the PDMS-immersed membrane has a higher permeance (with nothing to attach the PDMS to the membrane) than the base implies there is a combination of these effects going on.

- I think the x axis in Figure 7 g-i is meant to be in hours, not minutes.

- I appreciate the cross-flow data. However, once again, this needs to be benchmarked with a commercial membrane. Given the regular backwashing and chemical cleaning, coupled with the relatively low COD of these feeds, I am not positive about the advantage the coating offers without the benchmarking.

- In their responses, the authors state the novelty of their work differently in different sections. The PPRs reported are not novel on their own. The authors state "The main innovations of this study lie in the optimization of PPR preparation and the synergistic antifouling mechanism." The optimization of a material in terms of component ratios is of course valuable, but I am not sure it fits in this journal. In other sections, they also note "This study aims to highlight the potential of the supramolecular dynamics of CD/PDMS PRs in enhancing membrane flux and antifouling capabilities, rather than focusing on the modification effects of hydrophilic-LSE materials—an aspect that has been thoroughly analyzed in our prior research." The data reported, however, does not really show clear insight into these mechanisms as is...

To be clear, I am not aware of any membrane materials that combine PDMS, a very high mobility chain material, with

hydrophobic groups, so there is still some novelty. I do believe this manuscript has a home in the literature, and this is an interesting membrane material. However, given how few membrane papers this journal publishes, I am not sure it hits that bar of novelty specifically based on how the authors state their intended contribution and impact.

Reviewer #3

(Remarks to the Author)

The authors have solved all my concerns, and I think it can be considered to be accepted.

Version 2:

Reviewer comments:

Reviewer #1

(Remarks to the Author)

The authors have addressed all my questions.

Reviewer #2

(Remarks to the Author)

I appreciate the responses of the authors. I still have a few comments.

- I appreciated the explanations the authors provided in the response document. However, very few of these seem to be reflected to the main text. Some of the phenomena are not described clearly - especially given the interdisciplinary nature of the paper, where some aspects and terminology will be obvious to researchers in one area but not others. An example is the discussion of the maximum CD:PDMS ratios, which the authors allude to in the text as follows:

"While the 126 synthesis of β -CD/PDMS PPRs was possible, the maximum molar ratio of CDs to 127 PDMS units (rCD:PDMS) and the synthesis yield (calculated based on CD) declined when 128 the molecular weight of PDMS exceeded 1000 Da. In contrast, the synthesis of γ -129 CD/PDMS PPRs showed limited sensitivity to PDMS chain length (maximum rCD:PDMS 130 = 0.67, maximum yield = 90%)^{28, 31}."

This statement does not clarify what limits the CD:PDMS ratio; it sounds a lot more like an experimental limitation than a CD coverage effect. I recommend clarifying this in the text better also.

- A similar issue is with the XPS section. It seems the surface coverage is based on a mass fraction; this is not clearly stated anywhere - it must be specified. The word "molecular coverage" with an undefined percentage implies a mole percentage; it needs to be stated either way. The CD:PDMS ratio is a mole ratio.

- In every figure where flux is reported, the pressure difference used in the experiment must be noted in the caption. I think 1 bar, was used, but not this is not clear anywhere in the main text as the experimental section in the main text does not discuss filtration experiments; it is only in the SI section.

- I appreciate the addition of the base membrane, but there is some issue going on there. The FilmTec membrane reported is a brackish water RO membrane, not a UF membrane. A quick search led me to:

https://www.home-water-purifiers-and-filters.com/filmtec-TW30-2514.php?srltid=AfmBOop-aNvB7S5YZt0Df24tm2eLjI1JVwl28SaErRlens_mdIK3uEOd

I back-calculated its permeance to be 3.2 LMH/bar (so its flux at 1 bar would be 3.2 LMH) based on these data sheets.

Granted testing conditions will affect this value - but still, the reported flux seems quite unreasonable. Maybe the membrane acquired defects during the construction and deconstruction of the module; maybe something else happened - but the data seems problematic. The authors also note this is a UF membrane, so they should select a UF membrane. Perhaps the base membrane could be an option.

- Every figure with Normalized flux should include the base flux all values were normalized with for each membrane clearly in the caption.

Version 3:

Reviewer comments:

Reviewer #2

(Remarks to the Author)

Thank you for your edits. They are very helpful. I think the manuscript is now publishable.

RESPONSE TO REVIEWERS' COMMENTS

Reviewer #1:

The manuscript reports a novel approach utilizing a surface modifier composed of hydrophilic cyclodextrin (CD) rings threaded on hydrophobic polydimethylsiloxane (PDMS) to enhance the antifouling properties of polyethersulfone (PES) ultrafiltration membranes. While the concept appears innovative and the reported membrane performance improvements are notable, I have major concerns in the ambiguity of the presentations of the manuscript, which often make it almost impossible to understand. It is more about the logic flow of ideas and cohesiveness of the presentations rather than a language issue. The following are some examples.

A: Thank you for your guidance and support. In the updated manuscript, we have thoroughly revised the presentation for better clarity and added supporting and extended experiments to address these concerns.

Q1.1: The authors state that simulation on the CD/PDMS ratios were "partially confirmed by verification experiments". Questions: 1) What was the simulation method? And 2) what was the methodology of the verification experiments?

A1.1: The precursor research indicates that when the molecular weight of PDMS is large, the synthesis of β -CD/PDMS PPRs is challenging, as evidenced by low yields and smaller $r_{CD:PDMS}$ values. This study aims to improve the synthesis of β -CD/PDMS PPRs by modifying the solvent. Simulation calculations were used for solvent screening, and the selected solvents were experimentally validated. The simulation methodology is detailed in the first section of the supplementary information and has been referenced in the updated manuscript. The experimental method for preparing β -CD/PDMS PPRs is outlined in the updated manuscript, with the relevant descriptions reorganized.

R1.1: **Manuscript-Line 146-157.** Experimental results (see the Preparation of CD/PDMS PPRs section for details) confirmed that using DMF as a solvent instead of water increased the maximum $r_{CD:PDMS}$ and the yield of β -CD/PDMS PPRs to 0.59 and 51%, respectively, meeting the requirements of this study to control the dynamic properties of β -CD/PDMS PPRs by adjusting $r_{CD:PDMS}$. However, replacing water with

ethanol as the solvent resulted in lower actual yields for both β -CD/PDMS and γ -CD/PDMS PPRs. This discrepancy may be attributed to the poor dispersion of β -CD and γ -CD in ethanol, which hindered effective contact between the CDs and PDMS, a factor not accounted for in the simulations. Ultimately, β -CD/PDMS PPRs with $r_{\text{CD:PDMS}}$ values of 0, 0.13, 0.27, 0.40, 0.53, and 0.67 were prepared using DMF as the solvent, while γ -CD/PDMS PPRs with $r_{\text{CD:PDMS}}$ values of 0, 0.13, 0.27, 0.40, and 0.53 were synthesized using water in this study.

Manuscript-Line 504-508. The materials and reagents used in this study, the preparation method of the base membrane, molecular simulation parameters, characterization methods for the membrane materials, and evaluation methods for the overall membrane performance are described in detail in the section Materials and Methods of Supplementary Information.

Supplementary Information-Line 35-53. The investigation of CD/PDMS PPR formation propensity and CD activity was performed using the Forcite module in Materials Studio software. Simulation boxes were created using the COMPASS force field at 298K, and their structural energy was optimized prior to running molecular dynamics simulations. The simulations were carried out in two stages: first, an NPT ensemble was applied for 1000 ps to achieve density equilibrium, followed by an NVT ensemble for an additional 1000 ps to attain motion equilibrium.

For the study of CD/PDMS PPR formation, the simulation box contained 200 solvent molecules (water, DMF, or ethanol), 1 CD molecule, and a 14-mer PDMS chain. To assess CD activity, solvent molecules were excluded to improve repeatability and computational efficiency, with the focus placed on 1 CD molecule and a 10-mer PDMS chain. To minimize uncontrollable variables and ensure comparability of CD migration behaviors during the simulation, the following conditions were applied: 1) The PDMS chain was configured in a helical extension along the z-axis of the 3D coordinate system and was kept stationary throughout the CD migration, with its geometric center fixed on the z-axis. 2) The movement of CD along the PDMS chain was divided into two stages at each step, with the geometric center of the CD first moved along the z-axis by a fixed step length, followed by structural optimization while keeping the z-coordinate

of the CD's geometric center fixed. Initially, CDs were directed to move along the PDMS chain towards the direction corresponding to the larger aperture of the CD. The step length was set to 2/21 of the z-axis length of a PDMS monomer. The change in the total system energy was recorded at 21 step points (steps 1 to 21) across two PDMS monomers. At the 21st step point, the direction of CD movement was reversed towards the smaller aperture of the CD, and the corresponding Gibbs free energy changes (ΔG) for the next 21 step points (steps 22 to 42) were documented. While the z-axis coordinate of the 42nd step point aligns with the first monomer's position, the geometric structures of the system may differ due to the CD's movement.

Q1.2: It seems that the molecular simulation (Section 1.3, Supplementary Information) was used to determine the CD/PDMS ratios, a critical parameter in the study, in some places, e.g., Lines 168-170. However, this seems to be unlikely since the simulation was about calculating the energy changes in association with the "Brownian motions" of CDs along their PDMS thread. Whereas the experimental procedures of CD/PDMS threading (Lines 474–495) suggest that the probability of CDs to align satisfactorily well with one end of a PDMS molecule in the homogenized mixture should dictate the CD/PDMS ratio.

A1.2: The Gibbs free energy in Figure 2 is the only data derived from simulations; the rest are experimental results. The descriptions that may have caused confusion have been revised in the updated manuscript.

R1.2: Manuscript-Line 164-169.

		PDMS		ΔG (kcal/mol)	Maximum $r_{CD:PDMS}$	Yield (%)
 α -CD include		  	water bath	-1.25	/	0
			DMF bath	-4.30	/	0
			ethanol bath	-1.45	/	0
 β -CD include		  	water bath	-11.65	0.09	11
			DMF bath	-21.21	0.59	45
			ethanol bath	-26.32	-	<5
 γ -CD include		  	water bath	-15.88	0.67	77
			DMF bath	-36.24	0.67	71
			ethanol bath	-31.63	-	<7

Fig. 2 Formation of α -CD/PDMS, β -CD/PDMS, and γ -CD/PDMS PPRs in water, DMF, and ethanol baths. The ΔG values were obtained via molecular simulation, the maximum $r_{CD:PDMS}$ and yield were acquired from experimental validation. "/" indicates the absence of data, and "-" indicates the exclusion of data due to poor repeatability.

Q1.3: If you used molecular simulation for the Brownian motions, please make it clear in relevant parts in both the manuscript and the supplementary information.

A1.3: In this study, when simulating the resistance of CD sliding along PDMS, the inclusion of solvents made it difficult for the simulation results to converge or for significant patterns to emerge. Therefore, the full Brownian motion was not simulated; instead, forced displacement of the geometric center was used. The resistance of CD sliding along PDMS was mainly evaluated by analyzing the energy changes before and

after the movement. To avoid any misunderstanding, relevant explanations have been clarified in the updated manuscript.

R1.3: Manuscript-Line 177-179. Due to convergence and repeatability issues, the CD movement was simplified to a forced displacement of the geometric center rather than simulating the full Brownian motion.

Q1.4: How to interpret the significance of the diffusion coefficients (Figure 4) and how exactly were they determined?

A1.4: Using the QELS method, the percentage of molecules with different diffusion coefficients within the supramolecular system can be obtained. The three peaks corresponding to diffusion coefficients from low to high represent self-diffusion, sliding-diffusion, and cooperative diffusion, respectively. This characteristic has been clearly reported in prior research (Journal of Physics-Condensed Matter, 2005, 17, S2841-S2846). The sliding-diffusion peak specifically represents the supramolecular dynamic properties, with a larger peak area indicating stronger supramolecular dynamics. The method for measuring the diffusion coefficient is based on the principle that Molecular Brownian motion scatters the incident light, causing fluctuations and shifts in both the total intensity and the frequency of the scattered light. By measuring the time-dependent decay of the light intensity function, molecular diffusion state information can be analyzed using the Stokes-Einstein relationship. For nanomaterials, this principle can also be applied to measure the particle size of the nanomaterials, and the corresponding equipment is called a laser particle size analyzer.

R1.4: Manuscript-Line 199-204. The sliding motion of the CDs on PDMS, which represents the dynamic properties of CD/PDMS PPRs, was determined using quasi-elastic light scattering (QELS) (Fig. 4). Molecular Brownian motion scattered the incident light, causing the total intensity and frequency of the scattered light to fluctuate and shift. By measuring the time-dependent decay of the light intensity function, molecular diffusion state information can be analyzed using the Stokes-Einstein relationship.

Q1.5: The authors assert that "the dynamic movement of CDs along PDMS promotes water transport. (Lines 21–22)". However, no experimental or theoretical evidence was provided to support this claim. If it is a hypothesis, please explicitly state so. Otherwise, please supply evidence or revise this statement to align with the presented data.

A1.5: In the original manuscript, we proposed that "the dynamic movement of CDs along PDMS promotes water transport" based on the observed trends in membrane flux under varying CD types, $r_{\text{CD:PDMS}}$, and temperatures. Specifically, membrane flux peaked at $r_{\text{CD:PDMS}}$ of 0.27 and 0.40, with γ -CD/PDMS@M exhibiting higher fluxes, especially at elevated temperatures. These conditions coincide with enhanced dynamic movement of CD/PDMS pseudorotaxanes (PPRs), leading to the initial hypothesis. To support this inference, we ruled out the effects of hydrophilicity and pore size on membrane flux, further corroborating the proposed mechanism. In this revision, we conducted a supplementary counter-evidence experiment to validate the hypothesis. A significant amount of SDS was adsorbed onto the PDMS surface and filled the gaps between CDs and PDMS, effectively hindering CD mobility (*Soft Matter* **3**, 2007, 1456-1473; *Applied Materials Today*, 2017, **9**, 176-183). Under these conditions, membrane flux decreased significantly at $r_{\text{CD:PDMS}}$ of 0.27 and 0.40, while changes at other ratios were negligible. This result provides indirect evidence supporting the claim that the dynamic movement of CDs along PDMS promotes water transport.

R1.5: **Manuscript-Line 308-347.** The initial water fluxes of CD/PDMS@Ms and BM exhibited different trends at 5°C, 20°C, 35°C and 50°C (Fig. 6d-g). At 5°C, the initial water fluxes of CD/PDMS@Ms steadily increased with increasing $r_{\text{CD:PDMS}}$. At 20°C, the fluxes significantly increased as $r_{\text{CD:PDMS}}$ increased up to 0.4, followed by a slight decrease. The trend at 35°C was similar to that observed at 20°C, but the turning point for γ -CD/PDMS@M occurred at $r_{\text{CD:PDMS}} = 0.27$. At a higher temperature of 50°C, the turning point became more pronounced. The basic trends in water flux changes aligned with variations in membrane hydrophilicity. However, at higher temperatures, an additional factor contributed to enhanced flux, particularly at $r_{\text{CD:PDMS}} = 0.27$ and 0.40. This reason was considered as the activity of CD/PDMS PRs. On one hand, the QELS spectra revealed that CD/PDMS PRs exhibited peak activity at $r_{\text{CD:PDMS}} = 0.27$ and 0.40,

corresponding to the flux turning point. Second, grafting CD/PDMS PRs did not significantly alter membrane surface pore size (Supplementary Fig. 11) or pore count (as seen in SEM images), excluding these factors from influencing flux. A further experiment was conducted to confirm the influence of CD/PDMS PR activity on flux. In aqueous environments, sodium dodecyl sulfate (SDS) adsorbs onto the PDMS surface and penetrates CD cavities via hydrophobic interactions, temporarily impeding CD motion along PDMS chains^{33, 34}. The flux variations were monitored with SDS solutions as feed (Fig. 6h). Results showed that only SDS concentrations as high as 1.00 wt% effectively suppressed membrane flux, and the suppression was temporary, for flux gradually returned to initial levels upon switching back to pure water. At 35°C, when fed with 1.00 wt% SDS solution, the temporary steady-state fluxes of CD/PDMS@Ms and BM membranes are presented in Fig. 6i. Compared to Fig. 7f, the BM flux showed minimal change, while the flux of CD/PDMS@Ms decreased significantly at $r_{\text{CD:PDMS}} = 0.13, 0.27, \text{ and } 0.40$, with fluxes for γ -CD/PDMS@M and β -CD/PDMS@M converging. This phenomenon inversely supports the role of CD/PDMS PRs in enhancing flux. This unusual phenomenon was likely due to the abundant hydrogen bonds on the outer surface of the CDs. These hydrogen bonds can bind water molecules and facilitate their rapid transfer through the sliding, rotating, and vibrating movements of CDs along the PDMS chains. At 20°C, γ -CD/PDMS@M showed a maximum flux of approximately $550 \text{ L}\cdot\text{m}^{-2}\cdot\text{h}^{-1}$, surpassing that of most reported heterogeneous membranes (Supplementary Table 10).

Fig. 6 (a) Advancing contact angles, (b) receding contact angles, and (c) surface energies of BM and CD/PDMS@Ms with different $r_{CD:PDMS}$ values. Influence of CD type and $r_{CD:PDMS}$ on initial water flux of CD/PDMS@Ms and initial water flux of BM at (d) 5°C, (e) 20°C, (f) 35°C and (g) 50°C. (h) Effect of SDS addition on flux of γ -CD/PDMS@M-0.27. (i) Flux of BM and CD/PDMS@Ms with 1.00 wt% SDS feed at 35°C.

Q1.6: In crossflow membrane filtration systems, the tangential flow would likely cause unidirectional movement of CD rings towards one end of the PDMS.

R1.6: In microfiltration-ultrafiltration crossflow filtration systems, the membrane surface typically exhibits turbulent flow. From a macroscopic perspective, the water flow is directional. However, at the nanoscale, the direction of water flow changes constantly, making it unlikely to drive the unidirectional movement of CD rings towards one end of the PDMS. To validate this hypothesis, we updated the manuscript to include continuous 72-hour crossflow experiments using composite simulated foulants and real pollutants to evaluate the membrane antifouling performance.

The design of CD/PDMS PRs aims to address the bottleneck issue of flux interference caused by the antifouling mechanism in heterogeneous membranes, making their optimal application as antifouling coatings for ultrafiltration and microfiltration membranes. However, I am particularly intrigued by the phenomenon of laminar flow potentially causing CDs to accumulate at one end of the PDMS. Therefore, as an exploratory study, we attempted to load γ -CD/PDMS PRs onto the surface of nanofiltration membranes and conducted crossflow experiments for dye separation under BSA fouling. Although the membrane surface was in a laminar flow state, the γ -CD/PDMS PRs still exhibited antifouling performance superior to that of conventional heterogeneous membranes. This phenomenon is presumed to result from the elastic spatial hindrance or Brownian motion maintaining a certain degree of dynamic behavior of the CD rings, or the existence of additional antifouling mechanisms in the CD/PDMS PRs system beyond dynamic behavior, which will be further investigated in future studies.

R1.6: Manuscript-Line 423-454. To further evaluate the antifouling performance of the membrane under practical application conditions, a 72-hour cross-flow ultrafiltration experiment was conducted. The feed solutions included one simulated composite foulant solution and two types of real water samples. The former was an equal-volume mixture of the five pollutant solutions/emulsions used in the dead-end five-cycle ultrafiltration experiment³³, while the latter consisted of untreated Songhua River water and municipal wastewater treated by coagulation and sedimentation (Fig. 7g~i). The experiment included periodic cleaning every 4 hours using deionized water with both backwashing and forward flushing, while the first cleaning after the 48th hour was replaced with chemical cleaning. The results showed that γ -CD/PDMS@M maintained remarkable antifouling performance when exposed to both composite and real pollutants. The FDR after 4 hours of continuous operation was only 3~7%, and FRR after deionized water cleaning was nearly 100%. The membrane flux at the 12th cycle (before chemical cleaning) remained at over 97% of its initial value, and chemical cleaning restored the flux to its original level. The antifouling performance of β -CD/PDMS@M was slightly inferior to that of γ -CD/PDMS@M, with a flux decay rate

of 6~12% after 4 hours of continuous operation and a membrane flux of over 95% of its initial value at the 12th cycle. In comparison to previous studies, γ -CD/PDMS@M exhibits remarkable overall antifouling performance, maintaining a high flux of approximately $550 \text{ L}\cdot\text{m}^{-2}\cdot\text{h}^{-1}$. It achieves notable advantages over most hydrophilic-LSE synergistic antifouling membranes (Supplementary Table 11) and high-performing hydrophilic antifouling membranes (Supplementary Table 12).

Fig. 7 (a) Influence of $r_{\text{CD:PDMS}}$, (b) feed pH, (c) stirring rate, and (d) temperature on antifouling performance of CD/PDMS@Ms in dead-end ultrafiltration experiment. 5-cycle antifouling performance of (e) β -CD/PDMS@M and (f) γ -CD/PDMS@M with multiple foulants in dead-end ultrafiltration experiment. 72-hour cross-flow ultrafiltration experiment with feed of (g) composite foulant solution, (h) untreated Songhua River water and (i) municipal wastewater treated by coagulation and sedimentation. Unless otherwise specified, the model foulant was BSA, the $r_{\text{CD:PDMS}}$ of β -CD/PDMS@M was 0.40, the $r_{\text{CD:PDMS}}$ of γ -CD/PDMS@M was 0.27, and experiments were performed at pH 7 a stirring speed of 60 rpm, and temperature of 20°C .

Q1.7: The manuscript does not discuss mechanisms or forces that could counteract this movement. Are there any theoretical or experimental findings to suggest stabilization of CD positioning under tangential flow conditions? Would the aforementioned Brownian motions be sufficient?

A1.7: We understand your question in two ways, and we would like to address both. The first interpretation is about how to prevent the loss of CD during membrane operation. There are two structural forms of CD/PDMS supramolecules: the poly(pseudo)rotaxanes (PPR) and the polyrotaxane (PR). In the former, CDs can freely move in and out of the PDMS at both ends, while in the latter, due to steric hindrance, the CDs cannot move freely at both ends of the PDMS. In this study, we adopted a "clothesline" loading strategy (shown in Fig. 5a). Ideally, both ends of the PDMS in the CD/PDMS supramolecule are grafted onto poly-glycidyl methacrylate, forming a PR structure, which prevents the CD from leaving. For the CD/PDMS supramolecules with only one end grafted to glycidyl methacrylate, we capped the ungrafted end with FITC to similarly prevent CD loss.

The second interpretation is whether the dynamic motion of CDs on CD/PDMS PPRs can be maintained within a specific region during membrane operation. Specifically, whether CDs might be dragged by water flow to accumulate at one end of the CD/PDMS PPR, losing their dynamic properties, or whether their movable regions could be covered by foulants, thereby also losing their dynamic properties. The likelihood of the first scenario is very low, for details, please refer to my response to Question 1.6. The second scenario is possible, and the experiment we mentioned in response to your Question 1.5 provides an example. Notably, experimental results show that this suppression of PR dynamics becomes significant only when the foulant concentration reaches as high as 1.0wt% and is reversible. The primary reason for this phenomenon lies in the intrinsic low-surface-energy characteristics of PDMS, which result in weak interactions with foulants. Additionally, under the influence of Brownian motion and water flow, CDs can act like a "robot vacuum," gradually scraping the PDMS surface and facilitating the removal of foulants covering it.

R1.7: Manuscript-Line 91-95. These supramolecular products are classified as poly(pseudo)rotaxanes (PPRs) or polyrotaxanes (PRs) depending on whether the CDs can freely detach from the PDMS chain (i.e., when CD/PDMS PPRs are grafted onto the membrane surface via the PDMS end groups, they are transformed into CD/PDMS PRs).

Manuscript-Line 231-240. To minimize the interference of the base membrane surface on the activity of the CD/PDMS PPRs, a "clothesline" loading strategy was adopted to load CD/PDMS PPRs onto the membrane (Fig. 5a). First, glycidyl methacrylate (GM) was grafted onto the surface of a polyethersulfone (PES) base membrane via photo-induced radical polymerization to form molecular brushes with epoxy groups as side chains. Next, the amino groups at the ends of the PDMS chains in the CD/PDMS PPRs underwent an addition reaction with the epoxy groups of the molecular brushes, anchoring the CD/PDMS PPRs to the membrane. Finally, to prevent the CD leaching, the large molecule fluorescein-4-isothiocyanate (FITC) was employed to cap any unreacted terminal amino groups that were bonded to the membrane surface. The resulting membranes were denoted CD/PDMS@M- $r_{CD:PDMS}$ (e.g., PDMS@M and γ -CD/PDMS@M-0.67).

Manuscript-Line 286-292.

Fig. 5 (a) Schematic illustration of CD/PDMS PR linkage modes on membrane surface. (b) Relative fluorescence intensities of γ -CD/PDMS@M surfaces after grafting γ -CD/PDMS PPRs for different lengths of time and capping with FITC for 2 h. SEM images showing surfaces of (c) pristine PES membrane, (d) PDMS@M, and (e) β -CD/PDMS@M-0.13. (f) XPS full spectrum of γ -CD/PDMS@M-0.27. Surface molecular coverage of (g) β -CD/PDMS@M and (h) γ -CD/PDMS@M.

Q1.8: Please reword Lines 475–480, particularly the phrase "oversupplying β -CD at approximately 0.15, 0.19, 0.22, and 0.25 times", the meaning of which I can only guess.

A1.8: The sentence has been rewritten in the updated manuscript to achieve clearer expression.

R1.8: Manuscript-Line 523-527.

In the targeted β -CD/PDMS PPRs, the desired $r_{\text{CD:PDMS}}$ values of 0.13, 0.27, 0.40, and 0.53 were achieved by oversupplying β -CD by 0.15, 0.19, 0.22, and 0.25 times the theoretical amounts required for these ratios, respectively. Preliminary experiments

confirmed that this approach ensures accurate control of the final $r_{\text{CD:PDMS}}$ values in the synthesized PPRs.

Manuscript-Line 535-538.

In the targeted γ -CD/PDMS PPRs, the desired $r_{\text{CD:PDMS}}$ values of 0.13, 0.27, 0.40, 0.53, and 0.67 were achieved by oversupplying γ -CD by approximately 0.10, 0.14, 0.17, 0.19, and 0.20 times the theoretical amounts required for these ratios, respectively.

Addressing the above points is essential to enhance the manuscript's clarity, rigor, and credibility. Specifically, discrepancies in methodology, incomplete descriptions of experiments, missing links between claims, data, and experimental methods, as well as unsupported claims need to be resolved to ensure the findings are robust and credible. I look forward to reviewing a updated manuscript that addresses these concerns comprehensively.

A: Thank you for your valuable comments and suggestions, which have significantly helped improve the manuscript. We have carefully revised the manuscript to address all the points raised, ensuring greater clarity, rigor, and credibility. Specifically, we have resolved discrepancies in methodology, provided more detailed descriptions of the experiments, strengthened the connections between claims, data, and methods, and supported all claims with appropriate evidence. We hope the revised manuscript meets your expectations.

Reviewer #2 (Remarks to the Author):

This manuscript describes the coating of membranes with a layer of a PDMS-cyclodextrin pseudo-poly-rotaxane (PPR) to decrease fouling. The authors claim the reported fouling resistance arises from the high mobility of PDMS. The paper reports the formation of PPRs in detail with good data, though these PPRs have been prepared before in other contexts as shown in some of the later citations in the paper. They then graft the membrane surface with epoxide groups and attach the PPR to the membrane surface from a suspension. The paper includes some membrane fouling data as well as protein adsorption and AFM experiments. While the fouling resistance is good, it is not really exceptional compared with other materials and coatings in the literature, many prepared using more scalable methods and tested under more stringent conditions. As such, I do not believe this manuscript offers sufficient novelty and significance for publication in this journal. It may be a good fit in a somewhat more specialty journal. Beyond this, it also has some significant limitations related with the description of methodology and potential issues with membrane testing methodologies in demonstrating their claims (see below). More detailed comments are listed below.

A: The main innovations of this study lie in the optimization of PPR preparation and the synergistic antifouling mechanism. First, the preparation methods for β -CD/PDMS PPRs in prior research were not optimized, resulting in a low maximum $r_{\text{CD:PDMS}}$ ratio and limited yield (*Macromolecules*, 2001, 34, 6338-6343; *Macromolecules*, 2003, 36, 6422-6429; *Silicon*, 2012, 4, 151-156). This study addresses these limitations by using DMF as a solvent instead of water, significantly improving the synthesis process. Second, while the hydrophilic-low surface energy synergistic antifouling mechanism is less conventional, it offers notable advantages over single hydrophilic antifouling mechanisms, including reduced cleaning intensity requirements, better flux recovery post-cleaning, and an extended membrane lifespan by 1–2 times (*Advanced Functional Materials*, 2011, 21, 191-198; *Journal of Membrane Science*, 2014, 450, 111-123). The primary challenges limiting its widespread application—low baseline flux and reliance on tangential flow velocity—are precisely the critical issues this study aims to overcome (*Journal of Membrane Science*, 2013, 446, 456-464; *Journal of Materials Chemistry A*, 2016, 4, 7892-7902; *Journal of Membrane Science*, 2013, 441, 93-101).

Q2.1: A major problem is that the methodology in the main manuscript and SI does not include some critical pieces of information that would be needed for the work to be reproduced, and at times for the quality of the methodology to be evaluated. For

example, there is no information on what the PDMS molecule is (molar mass, end groups, manufacturer), how the concentrations and conditions for PPS formation and coating were selected, or key details of how the PPS was deposited on the membrane (e.g. swatch size used, immersion geometry, etc). The dispersion from which this was done is not characterized. The base membrane preparation is described, but not well-characterized. The description of filtration experiments is somewhat cursory. Foulant compositions are not documented. These are significant issues regarding publication.

A2.3: The experimental details, including the information on the PDMS molecule, the description of filtration experiments, and the foulant compositions, were provided in the Materials and Methods section of the original Supplementary Information. However, they were not referenced in the main manuscript. This oversight has been addressed, and these details are now incorporated into the updated manuscript.

The preparation method for β -CD/PDMS PPRs is one of the innovative aspects of this study. The research concept is detailed in the Preparation and optimization of CD/PDMS PPRs section, and the research methodology is outlined in the Preparation of CD/PDMS PPRs section of the original manuscript.

The loading of CD/PDMS PPRs onto the membrane surface was performed using standard grafting methods, with only minor adjustments based on experimental performance. The details of this process have been clarified in the updated manuscript. Specifically, we have added descriptions of the ratio of solution volume to membrane surface area and the fixation of membrane samples at the bottom of the beaker. The size and geometry of the membranes are not restrictive parameters; depending on characterization requirements and the shape of the membrane chamber, circular membranes with diameters of 35 mm or 63.5 mm can be prepared, as well as square membranes with side lengths of 70 mm or 150 mm.

Regarding the dispersion of CD/PDMS PPRs, it is referred to as a "dispersion" rather than a "solution" because CD/PDMS PPRs may dissociate into CD and PDMS after extended storage (over one week) or vigorous stirring. However, within a short period, the dispersion behaves similarly to a solution, remaining uniform and clear.

The performance parameters of the base membrane have also been added to the updated

manuscript to facilitate comparison with CD/PDMS@Ms.

R2.3: Manuscript-Line 516-567.

Methods

The materials and reagents used in this study, the preparation method of the base membrane, molecular simulation parameters, characterization methods for the membrane materials, and evaluation methods for the overall membrane performance are described in detail in the section Materials and Methods of Supplementary Information.

Preparation of CD/PDMS PPRs

In the targeted β -CD/PDMS PPRs, the desired $r_{\text{CD:PDMS}}$ values of 0.13, 0.27, 0.40, and 0.53 were achieved by oversupplying β -CD by 0.15, 0.19, 0.22, and 0.25 times the theoretical amounts required for these ratios, respectively. Preliminary experiments confirmed that this approach ensures accurate control of the final $r_{\text{CD:PDMS}}$ values in the synthesized PPRs. The specific preparation steps of β -CD/PDMS PPRs involved adding PDMS (263.3, 122.5, 80.6, and 58.4 mg) to a near-saturated β -CD solution in DMF (10mL, 59.0 mg/mL). The mixture was first subjected to supersonic agitation for 15 min, then slow shaken at room temperature for 7 days and last allowed to stand for 2 days. Post standing, the mixture was subjected to rotary evaporation to remove 90% of DMF by weight, followed by the addition of an equal mass of water to induce precipitation of β -CD/PDMS PPRs. The precipitated PPRs were sequentially washed with cyclohexane and water, then dried under vacuum.

In the targeted γ -CD/PDMS PPRs, the desired $r_{\text{CD:PDMS}}$ values of 0.13, 0.27, 0.40, 0.53, and 0.67 were achieved by oversupplying γ -CD by approximately 0.10, 0.14, 0.17, 0.19, and 0.20 times the theoretical amounts required for these ratios, respectively. The specific preparation process for γ -CD/PDMS PPRs involved adding PDMS (751.7, 349.2, 229.7, 170.4, 133.7 mg) to a near-saturated γ -CD solution in water (10mL, 184.2 mg/mL). The mixture was first subjected to ultrasonic agitation for 15 min, then slow shaken at room temperature for one day, and finally allowed to stand overnight. This resulted in a spontaneous gel-like precipitation of the γ -CD/PDMS PPRs. The precipitated PPRs were sequentially washed with cyclohexane and water, and then dried

under vacuum.

Loading of CD/PDMS PPRs on membrane surface

The base membrane was PES ultrafiltration membrane fabricated via customized surface segregation method (see materials and methods in supplementary information). The process of loading CD/PDMS PPRs onto the base membrane surface is depicted in Supplementary Fig. 14. Initially, the base membrane was coated with a 2.00 wt% glycidyl methacrylate (GM) aqueous solution, devoid of oxygen, at a rate of 100 $\mu\text{l}/\text{cm}^2$. It was then subjected to UV light (surface intensity 5.0 mW/cm^2) for 10 min under nitrogen protection to initiate radical polymerization and graft GM onto the membrane. Following this, the membrane was washed with deionized water to remove unreacted GM. Thereafter, the GM-grafted membrane was fixed at the bottom of a beaker, and a clear CD/PDMS PPR dispersion (2 g/L PPRs, approximately 50 wt% DMF and 50 wt% water) was poured in, with 10 mL of dispersion corresponding to each square centimeter of membrane surface area. Notably, the PPR dispersion was prepared by first dissolving PPRs in DMF before diluting with water, ensuring effective dispersion. When the molar fraction of CD in PPR is 0, meaning only PDMS is present, cyclohexane is used as the solvent. The beaker was left standing at room temperature for 12 hours. During this step, the PPRs were chemically linked to the poly GM via an addition reaction between the epoxy group and an amino-reactive hydrogen. Finally, the membrane was immersed in a 2.5 mL/cm^2 fluorescein-4-isothiocyanate (FITC) solution (5 mg/mL) for 3 hours to cap residual aminos. After the capping process, the CD within the PPRs would not leach out. Before use and characterization, the membrane underwent a 24-hour deionized water soaking period, with water changes every 8 hours.

Manuscript-Line 341-347.

Fig. 6 (a) Advancing contact angles, (b) receding contact angles, and (c) surface energies of BM and CD/PDMS@Ms with different $r_{CD:PDMS}$ values. Influence of CD type and $r_{CD:PDMS}$ on initial water flux of CD/PDMS@Ms and initial water flux of BM at (d) 5°C, (e) 20°C, (f) 35°C and (g) 50°C. (h) Effect of SDS addition on flux of γ -CD/PDMS@M-0.27. (i) Flux of BM and CD/PDMS@Ms with 1.00 wt% SDS feed at 35°C.

Manuscript-Line 445-454.

Fig. 7 (a) Influence of $r_{CD:PDMS}$, (b) feed pH, (c) stirring rate, and (d) temperature on antifouling performance of CD/PDMS@Ms in dead-end ultrafiltration experiment. 5-cycle antifouling performance of (e) β -CD/PDMS@M and (f) γ -CD/PDMS@M with multiple foulants in dead-end ultrafiltration experiment. 72-hour cross-flow ultrafiltration experiment with feed of (g) composite foulant solution, (h) untreated Songhua River water and (i) municipal wastewater treated by coagulation and sedimentation. Unless otherwise specified, the model foulant was BSA, the $r_{CD:PDMS}$ of β -CD/PDMS@M was 0.40, the $r_{CD:PDMS}$ of γ -CD/PDMS@M was 0.27, and experiments were performed at pH 7 a stirring speed of 60 rpm, and temperature of 20°C.

Q2.2: Figure 1 is not really informative; it is unclear what anything really represents. The cartoons make mechanistic claims that are not backed up by data.

A2.2: Figure 1 serves as a schematic illustration of the design concept of this study, aimed at helping readers better understand the research framework. However, we must clarify that the version of the figure included in the original manuscript was incorrect.

Specifically, the antifouling mechanism depicted in relation to small and large foulants was an early focus of our study but was later found to be insignificant. This section has been removed in the latest version of the figure, as the uploaded manuscript mistakenly included an outdated version. The corrected figure has been provided in the updated manuscript.

The other claims presented in the schematic are supported by data in this study: The "clothesline" structure of CD/PDMS PPRs with double-ended grafting can be indirectly confirmed by the fluorescence intensity changes on the membrane surface over time. The dynamic properties of CD/PDMS PPRs are demonstrated through molecular simulations and QELS characterization. The dynamic antifouling mechanism of CD/PDMS@Ms is supported by both dead-end and cross-flow ultrafiltration experiments. The contributions of Brownian motion and hydrodynamic assistance are verified through ultrafiltration experiments under varying stirring rates and adsorption experiments. We hope this clarification addresses your concerns.

R2.2: Manuscript-Line 117-119.

Fig. 1 Structure, dynamic source, and antifouling mechanism of novel antifouling modifier CD/PDMS PRs.

Q2.3: The surface SEMs imply rough, uneven coatings. It is unclear if there is full coverage of the PES surface or not, though that seems unlikely.

A2.3: LSE Materials tend to aggregate in aqueous environments, as illustrated in Figure 5d. In this study, the CD/PDMS PPRs exhibit a ring-threaded structure, where the linear PDMS component has low surface energy. However, the hydrophilic outer walls of the nested CD rings can partially reduce and inhibit aggregation, as shown in Figure 5c. During the drying process, as the surrounding water diminishes, the hydrophobic nature of PDMS causes uneven contraction of the CD/PDMS PPRs, a phenomenon commonly observed in hydrophilic-LSE heterogeneous membranes. Upon rehydration, the coating regains its swollen state.

Unlike conventional hydrophilic modifications, this study aims for the CD/PDMS PPRs to avoid overly dense distribution on the membrane surface. Excessive density could hinder the dynamic properties of the CD/PDMS PPRs. To address this, the UV-initiated free-radical polymerization process was intentionally optimized to control the grafting density at an appropriate level. Within the detection depth of XPS, the surface coverage of CD/PDMS PPRs on the membrane is approximately 45% to 55%.

R2.3: Manuscript-Line 286-292.

Fig. 5 (a) Schematic illustration of CD/PDMS PR linkage modes on membrane surface. (b) Relative fluorescence intensities of γ -CD/PDMS@M surfaces after grafting γ -CD/PDMS PPRs for different lengths of time and capping with FITC for 2 h. SEM images showing surfaces of (c) BM, (d) PDMS@M, and (e) β -CD/PDMS@M-0.13. (f) XPS full spectrum of γ -CD/PDMS@M-0.27. Surface molecular coverage of (g) β -CD/PDMS@M and (h) γ -CD/PDMS@M.

Q2.4: Cross-sectional SEM images are required to show the coating thickness and structure, and to determine if there is deposition inside membrane pores.

R2.4: The cross-sectional SEM images of BM, PDMS@M, and CD/PDMS@Ms have been added to the supplementary information. It has been confirmed that the CD/PDMS PR grafted layer does not penetrate deeply into the membrane pores, and its thickness is so minimal that accurate measurement is challenging. Relevant descriptions have been added to the updated manuscript.

R2.4: **Supplementary Information-Line 221-227.** The membrane cross-sectional

morphology shows that the CD/PDMS PR grafted layer does not penetrate deeply into the membrane pores, and its thickness is so minimal that it is difficult to measure accurately (Supplementary Fig. 10).

Supplementary Fig. 10 Cross-sectional SEM images of (a) BM, (b) PDMS@M, (c) β -CD/PDMS@M-0.13, (d) β -CD/PDMS@M-0.27, (e) β -CD/PDMS@M-0.40, (f) β -CD/PDMS@M-0.53, (g) γ -CD/PDMS@M-0.13, (h) γ -CD/PDMS@M-0.27, (i) γ -CD/PDMS@M-0.40, (j) γ -CD/PDMS@M-0.53, (k) γ -CD/PDMS@M-0.67.

Q2.5: How were the compositions in Figure 5 g-h calculated? If XPS survey elemental compositions were used, were they calibrated with correct materials? Or were high-resolution scans (e.g. C1s) used and if so, how was quantification performed?

A2.5: The compositions in Figure 5 g and h were calculated based on the XPS survey spectra of CD/PDMS@Ms, and the calculation process is detailed in Supplementary

Tables 4–7. To address potential measurement errors inherent in XPS, a determinant-based numerical method was employed to quantify and propagate these errors, ensuring the reliability of the data. Each membrane sample was tested twice starting from its preparation phase. Before analyzing the membrane materials, we calibrated the XPS survey spectra using PES, PDMS, and CD reference materials. The calibration process involved adjusting the sensitivity factors for the primary elements of interest (C, O, Si, S) to improve the accuracy of elemental quantification. High-resolution scans (e.g., C1s or O1s spectra) were not used for the final quantification due to inherent limitations: C1s scans were unsuitable because PDMS lacks distinct carbon peaks, which would prevent accurate differentiation between carbon contributions from other components like PES and CD. O1s scans were not employed because the binding energies of oxygen in CD, PES, PDMS, and GM overlap significantly, making peak deconvolution unreliable for precise quantification. While the use of high-resolution scans might have improved precision in ideal cases, the survey spectra calibration with sensitivity factors based on verified materials provided sufficiently accurate and consistent results for this study.

R2.5: Supplementary Information-Line 67-72.

1.4.4 X-ray photoelectron spectroscopy (XPS)

The surface chemical compositions of the membranes were quantitatively analyzed using a Thermo Scientific ESCALAB 250 XPS (USA) with Monochromated Mg Ka 150 W as the radiation source. The photoelectron take-off angle was set at 90°. Calibration was performed using reference materials, including PES, PDMS, and CD, to adjust the sensitivity factors for the detected elements (C, O, Si, and S), ensuring accuracy in the quantitative analysis. Each membrane sample was tested twice starting from its preparation phase.

Q2.6: The data shows very little PDMS deposited on the surfaces versus how much was used for the PPR. It seems the coatings are almost all CD. How is this explained?

A2.6: The data shown in Figures 5g and h represent molecular coverage, defined as the

ratio of the number of atoms from a specific molecule to the total number of atoms from all molecules. Since CD is nested within PDMS, the atom count for CD appears higher. However, the molar ratio of CD to PDMS units ($r_{\text{CD:PDMS}}$) on the membrane surface is comparable to that of the standalone CD/PDMS PPRs. For example, in the case of γ -CD/PDMS@M-0.40, the membrane was tested twice after preparation. The molecular coverage ratio of CD to PDMS was 6.98 in the first test and 6.68 in the second test, with an average of 6.83. This corresponds to an $r_{\text{CD:PDMS}}$ of 0.41. These results indicate that the loss of CD during the loading process of CD/PDMS PPRs onto the membrane surface is minimal.

Q2.7: The permeances of several coated membranes appear to be higher than the value for the uncoated base membrane. How is this possible?

A2.7: In the updated manuscript, we have included additional performance data for a similar base membrane (BM), such as water contact angle, surface energy, and flux at various temperatures. A comparison between BM and CD/PDMS@Ms reveals that at 5°C and 20°C, the membrane flux is primarily correlated with hydrophilicity, indicating that the influence of CD/PDMS PRs on pore size is negligible and mainly alters the surface hydrophilicity. At 35°C and 50°C, the flux of γ -CD/PDMS@Ms-0.27 and γ -CD/PDMS@Ms-0.40 significantly exceeds that of the BM. We attribute this to the enhancement of water permeability by the supramolecular dynamic properties. To further support this hypothesis, we applied a method to inhibit the supramolecular dynamics and successfully observed a corresponding decrease in flux, serving as a reverse validation.

R2.7: Manuscript-Line 308-347. The initial water fluxes of CD/PDMS@Ms and BM exhibited different trends at 5°C, 20°C, 35°C and 50°C (Fig. 6d-g). At 5°C, the initial water fluxes of CD/PDMS@Ms steadily increased with increasing $r_{\text{CD:PDMS}}$. At 20°C, the fluxes significantly increased as $r_{\text{CD:PDMS}}$ increased up to 0.4, followed by a slight decrease. The trend at 35°C was similar to that observed at 20°C, but the turning point for γ -CD/PDMS@M occurred at $r_{\text{CD:PDMS}} = 0.27$. At a higher temperature of 50°C, the turning point became more pronounced. The basic trends in water flux changes aligned

with variations in membrane hydrophilicity. However, at higher temperatures, an additional factor contributed to enhanced flux, particularly at $r_{\text{CD:PDMS}} = 0.27$ and 0.40 . This reason was considered as the activity of CD/PDMS PRs. On one hand, the QELS spectra revealed that CD/PDMS PRs exhibited peak activity at $r_{\text{CD:PDMS}} = 0.27$ and 0.40 , corresponding to the flux turning point. Second, grafting CD/PDMS PRs did not significantly alter membrane surface pore size (Supplementary Fig. 11) or pore count (as seen in SEM images), excluding these factors from influencing flux. A further experiment was conducted to confirm the influence of CD/PDMS PR activity on flux. In aqueous environments, sodium dodecyl sulfate (SDS) adsorbs onto the PDMS surface and penetrates CD cavities via hydrophobic interactions, temporarily impeding CD motion along PDMS chains. The flux variations were monitored with SDS solutions as feed (Fig. 6h). Results showed that only SDS concentrations as high as 1.00 wt% effectively suppressed membrane flux, and the suppression was temporary, for flux gradually returned to initial levels upon switching back to pure water. At 35°C , when fed with 1.00 wt% SDS solution, the temporary steady-state fluxes of CD/PDMS@Ms and BM membranes are presented in Fig. 6i. Compared to Fig. 7f, the BM flux showed minimal change, while the flux of CD/PDMS@Ms decreased significantly at $r_{\text{CD:PDMS}} = 0.13$, 0.27 , and 0.40 , with fluxes for γ -CD/PDMS@M and β -CD/PDMS@M converging. This phenomenon inversely supports the role of CD/PDMS PRs in enhancing flux. This unusual phenomenon was likely due to the abundant hydrogen bonds on the outer surface of the CDs. These hydrogen bonds can bind water molecules and facilitate their rapid transfer through the sliding, rotating, and vibrating movements of CDs along the PDMS chains. At 20°C , γ -CD/PDMS@M showed a maximum flux of approximately $550 \text{ L}\cdot\text{m}^{-2}\cdot\text{h}^{-1}$, surpassing that of most reported heterogeneous membranes (Supplementary Table 10).

Fig. 6 (a) Advancing contact angles, (b) receding contact angles, and (c) surface energies of BM and CD/PDMS@Ms with different $r_{CD:PDMS}$ values. Influence of CD type and $r_{CD:PDMS}$ on initial water flux of CD/PDMS@Ms and initial water flux of BM at (d) 5°C, (e) 20°C, (f) 35°C and (g) 50°C. (h) Effect of SDS addition on flux of γ -CD/PDMS@M-0.27. (i) Flux of BM and CD/PDMS@Ms with 1.00 wt% SDS feed at 35°C.

Q2.8: There is no measurement or report of any membrane selectivity measure (e.g. rejection, MWCO). This is important to contextualize the reported permeances, and also to determine if coating changes this capability.

A2.8: In fact, information on the pore size distribution and BSA rejection rate of CD/PDMS@Ms is included in the original supplementary materials, serving as supporting data for the effects of CD/PDMS PRs on membrane flux and antifouling performance. The overall conclusion is that the loading of CD/PDMS PRs onto the membrane surface has negligible impact on the pore size distribution and BSA rejection rate. After careful consideration, we believe that placing the pore size distribution and

BSA rejection data in the supplementary materials is appropriate since these are not the primary focus of this study. However, to ensure readers do not miss this relevant information while reading the main text, we have adjusted the referencing method to make it more prominent.

R2.8: Manuscript-Line 314-321. The basic trends in water flux changes aligned with variations in membrane hydrophilicity. However, at higher temperatures, an additional factor contributed to enhanced flux, particularly at $r_{\text{CD:PDMS}} = 0.27$ and 0.40 . This reason was considered as the activity of CD/PDMS PRs. On one hand, the QELS spectra revealed that CD/PDMS PRs exhibited peak activity at $r_{\text{CD:PDMS}} = 0.27$ and 0.40 , corresponding to the flux turning point. Second, grafting CD/PDMS PRs did not significantly alter membrane surface pore size (Supplementary Fig. 11) or pore count (as seen in SEM images), excluding these factors from influencing flux.

Manuscript-Line 373-377. Both CD/PDMS@Ms exhibited BSA rejection rates of close to 99.3%, and these rejection rates were unaffected by the $r_{\text{CD:PDMS}}$ value (see Supplementary Fig. 12). This indicates that supramolecular antifouling modification does not compromise the rejection performance of the modified membranes.

Supplementary Information-Line 253-262.

Supplementary Fig. 11 Surface pore size distribution of (a) PDMS@M, (b) β -CD/PDMS@M-0.13, (c) β -CD/PDMS@M-0.27, (d) β -CD/PDMS@M-0.40, (e) β -

CD/PDMS@M-0.53, (f) γ -CD/PDMS@M-0.13, (g) γ -CD/PDMS@M-0.27, (h) γ -CD/PDMS@M-0.40, (i) γ -CD/PDMS@M-0.53, (j) γ -CD/PDMS@M-0.67, measured via liquid-liquid displacement method.

Supplementary Fig. 12 Influence of CD kinds and $r_{\text{CD:PDMS}}$ on BSA rejection rate of CD/PDMS@Ms.

Q2.9: There are no fouling data reported for the base membrane or the epoxy-only coated membrane. This is important to show a baseline.

A2.9: As a baseline, data on the affinity, surface energy, flux, and antifouling performance of the base membrane have been added to the updated manuscript. For the GM grafted membrane, it was not listed separately because its performance differs very little from that of the base membrane due to the low grafting density. This study aims to highlight the potential of the supramolecular dynamics of CD/PDMS PRs in enhancing membrane flux and antifouling capabilities, rather than focusing on the modification effects of hydrophilic-LSE materials—an aspect that has been thoroughly analyzed in our prior research. Therefore, the discussion primarily centers on the differences between β -CD and γ -CD and variations in $r_{\text{CD:PDMS}}$, with less emphasis on the base membrane.

R2.9: Manuscript-Line 341-347.

Fig. 6 (a) Advancing contact angles, (b) receding contact angles, and (c) surface energies of BM and CD/PDMS@Ms with different $r_{CD:PDMS}$ values. Influence of CD type and $r_{CD:PDMS}$ on initial water flux of CD/PDMS@Ms and initial water flux of BM at (d) 5°C, (e) 20°C, (f) 35°C and (g) 50°C. (h) Effect of SDS addition on flux of γ -CD/PDMS@M-0.27. (i) Flux of BM and CD/PDMS@Ms with 1.00 wt% SDS feed at 35°C.

Manuscript-Line 445-454.

Fig. 7 (a) Influence of $r_{CD:PDMS}$, (b) feed pH, (c) stirring rate, and (d) temperature on antifouling performance of CD/PDMS@Ms in dead-end ultrafiltration experiment. 5-cycle antifouling performance of (e) β -CD/PDMS@M and (f) γ -CD/PDMS@M with multiple foulants in dead-end ultrafiltration experiment. 72-hour cross-flow ultrafiltration experiment with feed of (g) composite foulant solution, (h) untreated Songhua River water and (i) municipal wastewater treated by coagulation and sedimentation. Unless otherwise specified, the model foulant was BSA, the $r_{CD:PDMS}$ of β -CD/PDMS@M was 0.40, the $r_{CD:PDMS}$ of γ -CD/PDMS@M was 0.27, and experiments were performed at pH 7 a stirring speed of 60 rpm, and temperature of 20°C.

Q2.10: How stable is the coating? I noted the fouling experiments were done at only 60 rpm stirring, max 240 rpm, which is very slow. Did the coating delaminate at higher stir speeds?

A2.10: The low stirring speeds were employed to investigate whether the dynamics-enhanced synergistic antifouling mechanisms could overcome the limitations of

conventional synergistic antifouling mechanisms under low tangential flow conditions, rather than implying that CD/PDMS@Ms can only tolerate stirring speeds up to 240 rpm. In preliminary experiments, dead-end filtration was conducted at stirring speeds of up to 800 rpm (stirrer diameter ~61.5 mm), and the tangential velocity at the membrane's outer edge was approximately 2.58 m/s. No structural damage to the CD/PDMS@Ms membranes was observed within a minimum duration of 4 hours. In the updated manuscript, cross-flow experimental data demonstrated that, under a tangential flow velocity of 1.5 m/s, the membranes operated continuously for 72 hours and underwent backwashing with negligible changes in performance, further supporting the stability of the coating.

Q2.11: The fouling experiments reported are extremely short, only 10 minutes of exposure to the foulant between washes. This is not realistic. Importantly, the AFM data on Figure 8 indicate longer contact results in stronger adhesion to the foulant - so longer exposures can lead to major changes to the reported results.

A2.11: In fact, during the dead-end filtration experiments, the exposure time of the membrane surface to the foulants was 30 minutes. In the updated manuscript, this data has been clearly marked to avoid misunderstandings. To verify the antifouling performance of CD/PDMS@Ms under prolonged foulant retention conditions, a 72-hour cross-flow ultrafiltration experiment was added. The results demonstrated the membrane's long-lasting antifouling performance.

R2.11: Manuscript-Line 349-352.

The antifouling performance of CD/PDMS@Ms was evaluated using dead-end stirred ultrafiltration experiments with four steps: filtration with pure water (15 minutes), foulant separation (30 minutes), surface tangential flow cleaning (15 minutes), and another filtration step with pure water (15 minutes).

Manuscript-Line 423-454.

To further evaluate the antifouling performance of the membrane under practical application conditions, a 72-hour cross-flow ultrafiltration experiment was conducted. The feed solutions included one simulated composite foulant solution and two types of

real water samples. The former was an equal-volume mixture of the five pollutant solutions/emulsions used in the dead-end five-cycle ultrafiltration experiment³³, while the latter consisted of untreated Songhua River water and municipal wastewater treated by coagulation and sedimentation (Fig. 7g~i). The experiment included periodic cleaning every 4 hours using deionized water with both backwashing and forward flushing, while the first cleaning after the 48th hour was replaced with chemical cleaning. The results showed that γ -CD/PDMS@M maintained remarkable antifouling performance when exposed to both composite and real pollutants. The FDR after 4 hours of continuous operation was only 3~7%, and FRR after deionized water cleaning was nearly 100%. The membrane flux at the 12th cycle (before chemical cleaning) remained at over 97% of its initial value, and chemical cleaning restored the flux to its original level. The antifouling performance of β -CD/PDMS@M was slightly inferior to that of γ -CD/PDMS@M, with a flux decay rate of 6~12% after 4 hours of continuous operation and a membrane flux of over 95% of its initial value at the 12th cycle. In comparison to previous studies, γ -CD/PDMS@M exhibits remarkable overall antifouling performance, maintaining a high flux of approximately 550 L·m⁻²·h⁻¹. It achieves notable advantages over most hydrophilic-LSE synergistic antifouling membranes (Supplementary Table 10) and high-performing hydrophilic antifouling membranes (Supplementary Table 11).

Fig. 7 (a) Influence of $r_{CD:PDMS}$, (b) feed pH, (c) stirring rate, and (d) temperature on antifouling performance of CD/PDMS@Ms in dead-end ultrafiltration experiment. 5-cycle antifouling performance of (e) β -CD/PDMS@M and (f) γ -CD/PDMS@M with multiple foulants in dead-end ultrafiltration experiment. 72-hour cross-flow ultrafiltration experiment with feed of (g) composite foulant solution, (h) untreated Songhua River water and (i) municipal wastewater treated by coagulation and sedimentation. Unless otherwise specified, the model foulant was BSA, the $r_{CD:PDMS}$ of β -CD/PDMS@M was 0.40, the $r_{CD:PDMS}$ of γ -CD/PDMS@M was 0.27, and experiments were performed at pH 7 a stirring speed of 60 rpm, and temperature of 20°C.

Q2.12: As mentioned earlier, several other researchers have reported similar or better fouling resistance performance, often using amphiphilic copolymers. I do not know why these are not included in the comparison table in the SI, though it appears cited manuscripts are mostly from a smaller subset of groups/institutions. Some other papers that have reported comparable or higher FRR values, often under more challenging fouling conditions include:

<https://www.sciencedirect.com/science/article/abs/pii/S0376738807002669>

<https://www.sciencedirect.com/science/article/abs/pii/S0376738818303727?via=ihub>

<https://pubs.acs.org/doi/abs/10.1021/acsami.0c03075>

<https://www.sciencedirect.com/science/article/abs/pii/S0376738817314485>

<https://pubs.acs.org/doi/abs/10.1021/acsami.7b04884>

A2.12: This study primarily aims to explore the influence of molecular dynamics on membrane surfaces in synergistic antifouling mechanisms. Therefore, the comparative analysis focuses specifically on membranes with heterogeneous antifouling mechanisms. The selection criteria prioritize works that represent significant milestones in the evolution of heterogeneous antifouling strategies, with a preference for studies showcasing excellent comprehensive antifouling performance and high citation impact. The inclusion of studies is not influenced by the affiliation of specific groups or institutions; any observed concentration of citations reflects the systematic and in-depth research contributions of those groups in this area.

Following your valuable suggestion, we have conducted a supplementary 72-hour crossflow filtration experiment under challenging fouling conditions. The experimental conditions were based on the article you recommended and have been appropriately cited. Additionally, we expanded our comparative analysis to include 10 high-performing hydrophilic membranes, including those recommended. Results demonstrate that our study's membrane maintains superior antifouling capabilities under real water conditions with mixed foulants, achieving lower flux decline and higher flux recovery under crossflow conditions compared to dead-end filtration. Overall, the results reaffirm that our membrane's performance leads both in hydrophilic-low surface energy heterogeneous membranes and in hydrophilic membranes. This underscores the significance and robustness of our findings in advancing antifouling membrane technology.

R2.12: Manuscript-Line 423-444.

To further evaluate the antifouling performance of the membrane under practical application conditions, a 72-hour cross-flow ultrafiltration experiment was conducted. The feed solutions included one simulated composite foulant solution and two types of

real water samples. The former was an equal-volume mixture of the five pollutant solutions/emulsions used in the dead-end five-cycle ultrafiltration experiment^{33, 34}, while the latter consisted of untreated Songhua River water and municipal wastewater treated by coagulation and sedimentation (Fig. 7g~i). The experiment included periodic cleaning every 4 hours using deionized water with both backwashing and forward flushing, while the first cleaning after the 48th hour was replaced with chemical cleaning. The results showed that γ -CD/PDMS@M maintained remarkable antifouling performance when exposed to both composite and real pollutants. The FDR after 4 hours of continuous operation was only 3~7%, and FRR after deionized water cleaning was nearly 100%. The membrane flux at the 12th cycle (before chemical cleaning) remained at over 97% of its initial value, and chemical cleaning restored the flux to its original level. The antifouling performance of β -CD/PDMS@M was slightly inferior to that of γ -CD/PDMS@M, with a flux decay rate of 6~12% after 4 hours of continuous operation and a membrane flux of over 95% of its initial value at the 12th cycle. In comparison to previous studies, γ -CD/PDMS@M exhibits remarkable overall antifouling performance, maintaining a high flux of approximately $550 \text{ L}\cdot\text{m}^{-2}\cdot\text{h}^{-1}$. It achieves notable advantages over most hydrophilic-LSE synergistic antifouling membranes (Supplementary Table 11) and high-performing hydrophilic antifouling membranes (Supplementary Table 12).

Supplementary Information-Line 270-275.

Supplementary Table 12 Comparison of reported hydrophilic membrane performances

Report date	Hydrophilic substances - zwitterionic copolymer	Membrane permeability ($\text{L}\cdot\text{m}^{-2}\cdot\text{h}^{-1}\cdot\text{bar}^{-1}$)	Model foulants	Fouling duration, stirring speed (rpm), flux decline rate (FDR)	Cleaning methods, flux recovery rate (FRR)
2007 ²⁰	PEO-g- PAN	159	BSA	24 h, 500, 35%	Deionized water

					backwashing, 100%
2017 ²¹	MPC	4.74	Real textile wastewater	6.5 h, cross-flow filtration, 11%	Deionized water surface washing, 100%
2017 ²¹	MPC	4.74	High concentration mixture of EfOM components	6.5 h, cross-flow filtration, 8%	Deionized water surface washing, 100%
2017 ²²	SBMA	5.83	BSA	24 h, /, 13%	Deionized water surface washing, 99%
2018 ²³	PAES-co- SBAES	2.5	BSA	12 h, cross-flow filtration, 10%	Deionized water surface washing, 94%
2020 ²⁴	P(TFEMA- OEGMA- AHPMA)	3.1	BSA	40 h, /, 18%	Deionized water surface washing, 99%
2021 ²⁵	DMAPAPS	364	HA	30 min, /, 46%	Deionized water backwashing, 98.1%

2023 ²⁶	MPC/QDMA	255.4	BSA	30 min, /, 65%	Deionized water surface washing, ~100%
2024 ²⁷	SBMA	1000	BSA	2 h, /, 60%	Deionized water surface washing, 88%
2024 ²⁸	MDSA-PEGDA hydrogel	246	BSA	3 h, /, 33.3%	Deionized water surface washing, 80%
2025 ²⁹	P(4VP-co-MPC)	150	BSA	30 min, /, 23.1%	Deionized water surface washing, 95.3%

SBMA: Sulfobetaine methacrylate; MPC: 2-methacryloyloxyethyl phosphorylcholine; PAES-co-SBAES: Poly(arylene ether sulfone-co-sulfobetaine arylene ether sulfone); P(TFEMA-OEGMA-AHPMA): poly(trifluoroethyl methacrylate-co-oligo-(ethylene glycol) methyl ether methacrylate-co-(3-azide-2-hydroxypropyl methacrylate)) ; DMAPAPS : Sulfonated 3-Dimethylaminopropylamine; MPC/QDMA: Methacryloyloxyethyl phosphorylcholine/ 2-(methacryloyloxyethyl)trimethylammonium iodide; MDSA: [2-(Methacryloyloxy) ethyl] dimethyl-(3-sulfopropyl) ammonium hydroxide; PEGDA: Poly (ethylene glycol) diacrylate P(4VP-co-MPC): Poly(4-vinylpyridine-co-methylacryloyloxyethyl phosphocholine).

Reviewer #3

The “resistance-release synergistic antifouling mechanism” based on hydrophilic–low surface energy heterogeneous structure is an excellent strategy for antifouling applications. However, its implementation on membrane surfaces faces two major bottlenecks: low initial water flux and reliance of antifouling performance on intense tangential flow. This study introduces a heterogeneous modification of the membrane surface using pseudo-polyrotaxane (PPR) composed of hydrophilic cyclodextrin (CD) and low-surface-energy polydimethylsiloxane (PDMS). Leveraging the dynamic nature of PPRs, the study effectively addresses the two key challenges of the synergistic antifouling mechanism, demonstrating significant innovation and practical value. It is recommended for publication in Nature Communications. The following suggestions are provided to refine and optimize the manuscript:

A: Thank you for your strong support and recognition.

A3.1: The figure mentions “against fine particulate foulants” and “against coarse particulate foulants,” but corresponding descriptions are missing in the text. Either revise the figure labels or supplement the text with the relevant descriptions.

Q3.1: Figure 1 in the original manuscript was an outdated version that was used by mistake. The impact of foulant size on the membrane dynamic antifouling effect was originally one of the key focuses of this study, and the old version of Figure 1 reflected that research direction. However, after experimental comparisons, we found that the relationship between foulant size and the membrane antifouling performance was not significant. As a result, this direction was abandoned. In the updated manuscript, the figure has been corrected accordingly to reflect the updated focus of the study.

R3.1: anuscript-Line 117-119.

Fig. 1 Structure, dynamic source, and antifouling mechanism of novel antifouling modifier CD/PDMS PRs.

A3.2: PPRs exhibit three types of dynamics: sliding, rotating, and vibrating. This study focuses solely on sliding dynamics through molecular simulations and quasi-elastic light scattering. While the results show that the sliding dynamics of β -CD/PDMS PPRs are weaker than those of γ -CD/PDMS PPRs, does this necessarily imply that the overall dynamics of β -CD/PDMS PPRs are also weaker? Or is sliding dynamics the only critical factor influencing membrane performance? Please clarify.

Q3.2: We believe that the three types of dynamics—sliding, rotating, and vibrating—of CD/PDMS PRs all contribute to membrane performance. However, based on the significant effects of the $r_{CD:PDMS}$ on membrane flux and antifouling performance, it is reasonable to conclude that the contribution of sliding dynamics is the most pronounced. When the $r_{CD:PDMS} = 0.67$, the sliding dynamics are primarily suppressed, while the rotating and vibrating dynamics are less affected. If the rotating and vibrating dynamics

were the dominant factors influencing membrane performance, membrane flux and antifouling performance should continuously improve as the $r_{\text{CD:PDMS}}$ increases, rather than initially increasing and then decreasing.

In practice, the three dynamic behaviors of CDs coexist within the CD/PDMS PRs, and the main influencing factor is steric hindrance, so the general trend follows a similar pattern. The molecular simulations presented in the study focus primarily on sliding dynamics, but at any given moment, atomic movement in the x and y directions is not restricted, meaning that the effects of rotation and vibration on sliding are inherently included in the simulation. As for simulations focused on rotation and vibration, these are difficult to quantify due to the significant influence of the initial positions. Similarly, due to limitations of characterization techniques, the QELS was used to capture the sliding characteristics of CD/PDMS PRs. However, this characteristic can represent the overall dynamic behavior of the CD/PDMS PRs.

R3.2: Manuscript-Line 188-193. Theoretically, steric hindrance also affects rotational and vibrational behavior. However, the rotational and vibrational energy barriers of CDs are much more influenced by their initial positions than by the CD type, making it difficult to quantify and analyze this aspect. Overall, the γ -CD/PDMS PPRs were expected to exhibit higher overall activity compared to the β -CD/PDMS PPRs.

Manuscript-Line 199-201. The sliding motion of the CDs on PDMS, which represents the dynamic properties of CD/PDMS PPRs, was determined using quasi-elastic light scattering (QELS) (Fig. 4).

Q3.3: How was fluorescence intensity used to determine the degree of single-ended and double-ended grafting of PPRs on the membrane surface? The current description is confusing. Please provide a detailed explanation in the text.

A3.3: We have revised the relevant section to clarify the reasoning behind comparing the number of single-ended and double-ended grafts, as well as the principle behind using fluorescence intensity for semi-quantitative characterization of the grafting degree.

R3.3: Manuscript-Line 241-257.

The grafting of CD/PDMS PPRs on the membrane surface occurs in single-end and double-end grafting, with double-end grafting being preferred due to its ability to reduce the overall mobility of CD/PDMS PPRs. This, in turn, highlights the contribution of CD's supramolecular dynamics on PDMS to the antifouling effect. The amount of single-end grafted CD/PDMS PPRs on the membrane surface is directly proportional to the amount of the cap agent FITC, which emits green fluorescence under UV light. Therefore, fluorescence microscopy was employed for semi-quantitative analysis of the ratio of single-end to double-end grafting at different reaction duration, as shown in Figure 5 (corresponding fluorescence images in Supplementary Fig. 8). The fluorescence intensity on the membrane surface initially increased and then decreased with the extension of reaction duration, indicating that most CD/PDMS PPRs initially underwent single-end grafting, and gradually transitioned to double-end grafting as the reaction progressed. After 4 hours of reaction, the amount of single-end grafted CD/PDMS PPRs was approximately 3.4 times greater than that after 8 hours of reaction, suggesting that the amount of double-end grafted CD/PDMS PPRs was at least 2.4 times greater than that of single-end grafted CD/PDMS PPRs.

Q3.4: In the section Separation performance of CD/PDMS@Ms, systematically analyze the differences in antifouling performance between the two membranes for various pollutants, and discuss the underlying mechanisms.

A3.4: In this study, the slightly atypical antifouling performance compared to other studies is the high flux decline rate caused by SA. Relevant analysis has been added to the updated manuscript.

R3.4: Manuscript-Line 413-422. To investigate the broad applicability and multi-use stability of CD/PDMS@Ms, continuous five-cycle filtration experiments were conducted using BSA, emulsified hexadecane, sodium alginate (SA), yeast, or fulvic acid (FA) as model foulants (Fig. 7h, i). The results showed that γ -CD/PDMS@M consistently exhibited significantly lower FDR compared to β -CD/PDMS@M and nearly 100% FRR, indicating the strong generalizability of the dynamics-enhanced synergistic antifouling mechanisms. Notably, the high FDR caused by SA warrants

attention. However, given the nearly 100% FRR after cleaning, it is speculated that this is due to the formation of a cake layer composed of gel-like structures over the membrane surface, an issue that cannot be resolved solely by enhancing the dynamic properties of the membrane surface.

Q3.5: In the section Loading of CD/PDMS PPRs on membrane surface, it is stated that the solvent for the PPR is approximately 50 wt% DMF and 50 wt% water. However, in an extreme case where the PPR contains no CD (i.e., the modifier consists solely of PDMS), DMF and water are unable to dissolve PDMS. Please address this discrepancy.

A3.5: In the case where the modifier consists solely of PDMS, cyclohexane is used as the solvent, as DMF and water cannot dissolve PDMS. This clarification was missing in the original manuscript but has been added in the revised version.

R3.5: Manuscript-Line 559-560.

Loading of CD/PDMS PPRs on membrane surface

When the molar fraction of CD in PPR is 0, meaning only PDMS is present, cyclohexane is used as the solvent.

Supplementary Information-Line 23-24.

Materials and methods-Materials and reagents

Cyclohexane (AR), hexadecane (AR) and dimethyl formamide (DMF, AR) were purchased from Shanghai Macklin Biochemical Co. Ltd. (China).

Q3.6: In the same section, some experimental steps only provide the concentration and volume of the modification solution but omit the membrane area submerged in the solution. Please specify that the solution was used in significant excess or provide the amount of solution used per unit membrane area.

A3.6: Regarding the experimental procedure, the modification solution is used in excess relative to the membrane area. The ratio of solution to membrane area has now been specified in the updated manuscript.

R3.6: Manuscript-Line 554-561 and 563-565.

Methods

Loading of CD/PDMS PPRs on membrane surface

Thereafter, the GM-grafted membrane was fixed at the bottom of a beaker, and a clear CD/PDMS PPR dispersion (2 g/L PPRs, approximately 50 wt% DMF and 50 wt% water) was poured in, with 10 mL of dispersion corresponding to each square centimeter of membrane surface area. Notably, the PPR dispersion was prepared by first dissolving PPRs in DMF before diluting with water, ensuring effective dispersion. When the molar fraction of CD in PPR is 0, meaning only PDMS is present, cyclohexane is used as the solvent. The beaker was left standing at room temperature for 12 hours.

Finally, the membrane was immersed in a 2.5 mL/cm² fluorescein-4-isothiocyanate (FITC) solution (5 mg/mL) for 3 hours to cap residual aminos.

Q3.7: In this study, the dead-end stirred cell filtration system is used to evaluate membrane performance. A long-term filtration experiment using cross flow equipment should proceed to assess its potential in industry application.

A3.7: A 72-hour cross-flow ultrafiltration experiment has been added to the updated manuscript.

R3.7: Manuscript-Line 423-454.

To further evaluate the antifouling performance of the membrane under practical application conditions, a 72-hour cross-flow ultrafiltration experiment was conducted. The feed solutions included one simulated composite foulant solution and two types of real water samples. The former was an equal-volume mixture of the five pollutant solutions/emulsions used in the dead-end five-cycle ultrafiltration experiment³³, while the latter consisted of untreated Songhua River water and municipal wastewater treated by coagulation and sedimentation (Fig. 7g~i). The experiment included periodic cleaning every 4 hours using deionized water with both backwashing and forward flushing, while the first cleaning after the 48th hour was replaced with chemical cleaning. The results showed that γ -CD/PDMS@M maintained remarkable antifouling performance when exposed to both composite and real pollutants. The FDR after 4 hours of continuous operation was only 3~7%, and FRR after deionized water cleaning was nearly 100%. The membrane flux at the 12th cycle (before chemical cleaning)

remained at over 97% of its initial value, and chemical cleaning restored the flux to its original level. The antifouling performance of β -CD/PDMS@M was slightly inferior to that of γ -CD/PDMS@M, with a flux decay rate of 6~12% after 4 hours of continuous operation and a membrane flux of over 95% of its initial value at the 12th cycle. In comparison to previous studies, γ -CD/PDMS@M exhibits remarkable overall antifouling performance, maintaining a high flux of approximately $550 \text{ L}\cdot\text{m}^{-2}\cdot\text{h}^{-1}$. It achieves notable advantages over most hydrophilic-LSE synergistic antifouling membranes (Supplementary Table 11) and high-performing hydrophilic antifouling membranes (Supplementary Table 12).

Fig. 7 (a) Influence of $r_{\text{CD:PDMS}}$, (b) feed pH, (c) stirring rate, and (d) temperature on antifouling performance of CD/PDMS@Ms in dead-end ultrafiltration experiment. 5-cycle antifouling performance of (e) β -CD/PDMS@M and (f) γ -CD/PDMS@M with multiple foulants in dead-end ultrafiltration experiment. 72-hour cross-flow ultrafiltration experiment with feed of (g) composite foulant solution, (h) untreated Songhua River water and (i) municipal wastewater treated by coagulation and sedimentation. Unless otherwise specified, the model foulant was BSA, the $r_{\text{CD:PDMS}}$ of

β -CD/PDMS@M was 0.40, the $r_{\text{CD:PDMS}}$ γ -CD/PDMS@M was 0.27, and experiments were performed at pH 7 a stirring speed of 60 rpm, and temperature of 20°C.

RESPONSE TO REVIEWERS' COMMENTS

The first-round revisions in the manuscript and supplementary information are highlighted in green, while the second-round revisions are highlighted in blue.

Reviewer #1:

The authors have satisfactorily addressed most of my comments on the original manuscript. The following are some minor questions/suggestions for their reference.

A: Thank you very much for your guidance and support. We appreciate the time and effort you have dedicated to reviewing our manuscript. Below are our responses to the minor questions and suggestions you raised.

Q1.1: I was not able to find the methodology for grafting CD/PDMS to membrane surface, either in the manuscript or in the supplementary materials. Please add it.

A1.1: The methodology for grafting CD/PDMS to the membrane surface is included in the manuscript under the "Methods - Loading of CD/PDMS PPRs on membrane surface" section. In the revised manuscript, we have further clarified and expanded this section to make the explanation more precise.

R1.1: (Manuscript-Line 539)

Loading of CD/PDMS PPRs on membrane surface

The base membrane was PES ultrafiltration membrane fabricated via surface segregation method (see materials and methods in supplementary information). The process of loading CD/PDMS PPRs onto the base membrane surface consisted of three main steps, as depicted in Supplementary Fig. 15.

The first step was the grafting of GMA onto the membrane surface to form linear PGMA. The base membrane was coated with a 2.00wt% GMA aqueous solution, devoid of oxygen, at a rate of 100 $\mu\text{l}/\text{cm}^2$. It was then subjected to UV light (surface intensity 5.0 mW/cm^2) for 10 min under nitrogen protection to initiate radical polymerization. Following this, the membrane was washed with deionized water to remove unreacted GMA.

The second step was the reaction between CD/PDMS PPRs and GMA. The PGMA-grafted membrane was fixed at the bottom of a beaker, and a clear CD/PDMS PPR dispersion (2 g/L PPRs, approximately 50wt% DMF and 50wt% water) was poured in, with 10 mL of dispersion corresponding to each square centimeter of membrane surface area. Notably, the PPR dispersion was prepared by first dissolving PPRs in DMF before diluting with water, ensuring effective dispersion. When the molar fraction of CD in PPR is 0, meaning only PDMS is present, cyclohexane is used as the solvent. The beaker was left standing at room temperature for 12 hours. During this step, the PPRs were chemically linked to PGMA via an addition reaction between the epoxy groups on the side chains of PGMA and the terminal amines of PDMS.

The third step was the capping of CD/PDMS PPRs. The membrane was immersed in a 2.5 mL/cm² fluorescein-4-isothiocyanate (FITC) solution (5 mg/mL) for 3 hours to cap residual amines. After the capping process, the CD within the PPRs would not leach out. Before use and characterization, the membrane underwent a 24-hour deionized water soaking period, with water changes every 8 hours.

Q1.2: I still feel that the description on the methodology of Molecular (Dynamics) Simulation is not sufficient. Please add more details.

A1.2: We have revised and expanded the description of the molecular dynamics simulation methodology in the manuscript. The revised section provides more detailed information on the simulation setup, including the specific ensembles used, system configurations, and the calculation of Gibbs free energy changes (ΔG). We believe these additions address the concern and clarify the methodology. Specific changes can be found in the revised manuscript.

R1.2: (Supplementary Information-Line 34)

1.3 Molecular simulation

The investigation of CD/PDMS PPR formation propensity and CD activity was conducted using the Forcite module in Materials Studio software. The simulation boxes were constructed using the COMPASS force field at 298 K, and the initial structure was energy-optimized before molecular dynamics simulations. The simulations were

carried out in two stages: An NPT ensemble was applied for 1000 ps to achieve density equilibrium, followed by an NVT ensemble for an additional 1000 ps to reach motion equilibrium.

1.3.1 Simulation for CD/PDMS PPR Formation

For the study of CD/PDMS PPR formation, the simulation box included 200 solvent molecules (water, DMF, or ethanol), one CD molecule, and a 14-mer PDMS chain. The Gibbs free energy change (ΔG) was calculated before and after the formation of the CD/PDMS PPR to assess whether the formation process was spontaneous. The solvent molecules were considered in this simulation to represent the realistic system conditions.

1.3.2 Simulation for CD Migration along PDMS Chain

For evaluating the migration behavior of CD along the PDMS chain, a separate simulation was conducted without solvent molecules to improve computational efficiency and repeatability. The simulation focused on a single CD molecule and a 10-mer PDMS chain. The migration process was analyzed by calculating the system's Gibbs free energy (ΔG) at each step, to assess the feasibility and resistance of the migration process. Detailed Simulation Setup for CD Migration: The PDMS chain was configured in a helical extension along the z-axis of the 3D coordinate system and kept stationary during the migration, with its geometric center fixed on the z-axis. The CD molecule was moved along the PDMS chain towards the direction of the larger aperture of the CD. The step size for each migration step was set to $2/21$ of the z-axis length of a single PDMS monomer. The migration process was divided into two stages: the first stage was the movement towards the larger aperture of the CD, and the second stage was the reverse migration towards the smaller aperture. The simulation was carried out with a time step of 1 fs, and the Gibbs free energy was recorded at 21-step intervals during the migration. At step 21, the direction of the CD movement was reversed, and the next 21 steps (steps 22-42) were analyzed for the Gibbs free energy change. In each simulation, structural optimizations were performed after each step to allow for the relaxation of the system. The analysis focused on changes in the system's Gibbs free energy (ΔG) and the behavior of CD molecules as they moved along the PDMS chain.

Q1.3: Please make clear how the system energies reported in Supplementary Table 1 were obtained.

A1.3: The system definition has been clarified in Supplementary Table 1, where the conditions of constant temperature and pressure are specified, and the components of the system include CD, PDMS, and the solvent. Additionally, the system energy reported here refers to the Gibbs free energy (ΔG), rather than the internal energy (E). Accordingly, the symbol has been corrected to "G" to accurately reflect this.

R1.3: (Manuscript-Line 141) The Gibbs free energy changes (ΔG) before and after the formation of CD/PDMS PPR were used to determine the formation tendency of CD/PDMS PPR. A larger ΔG value indicates that the formation of CD/PDMS PPR is thermodynamically favorable.

(Supplementary Information-Line 193)

Supplementary Table 1 System Gibbs free energy changes (ΔG) before and after the formation of CD/PDMS PPR^a.

System energy	System Gibbs free energy in different solvent environment (kcal/mol)		
	Water	Ethanol	DMF
$G_{\text{unassembled } \alpha\text{-CD/PDMS}}$	-1794.348	-1461.613	3041.476
$G_{\text{unassembled } \beta\text{-CD/PDMS}}$	-1740.327	-1412.627	3096.61
$G_{\text{unassembled } \gamma\text{-CD/PDMS}}$	-1677.861	-1368.187	3173.466
$G_{\text{assembled } \alpha\text{-CD/PDMS}}$	-1795.602	-1463.062	3037.176
$G_{\text{assembled } \beta\text{-CD/PDMS}}$	-1751.975	-1438.951	3075.396
$G_{\text{assembled } \gamma\text{-CD/PDMS}}$	-1693.745	-1399.813	3137.223
$\Delta G_{\alpha\text{-CD/PDMS assembly}}$	-1.254	-1.449	-4.3
$\Delta G_{\beta\text{-CD/PDMS assembly}}$	-11.648	-26.324	-21.214
$\Delta G_{\gamma\text{-CD/PDMS assembly}}$	-15.884	-31.626	-36.243

^a The system included three components: CD, PDMS, and solvent (water, DMF, or ethanol).

Reviewer #2:

I would like to thank the authors for their responses. The long-term fouling studies add to the study. The responses regarding some of the details for XPS, experimental details for coating, and other requests, which were echoed by other reviewers, are appreciated. However, I still have some concerns about this paper and its true novelty and impact. Please see below for further comments:

Q2.1: I still do not believe that this manuscript shows any data or justification regarding the mechanisms of fouling resistance they claim quite strongly throughout the manuscript. The image in Figure 1 is still unclear in what it means to me - what is meant by "dynamic source"? There is no clear data showing the tangential flow is really changing the movement of CDs. The anti-fouling mechanism with a running CD (I think) pushing off a foulant is also unreasonable. At a basic level, a macromolecular foulant like BSA is much bigger than CD (by an order of magnitude); there is no pushing off involved in fouling resistance. There are, indeed, some studies on how the high molecular mobility of PDMS can contribute to its self-cleaning capabilities, but it is not this type of mechanism.

A2.1: We believe there has been a significant misunderstanding here. The only mechanism we aimed to explain in this study is the supramolecular dynamics-enhanced synergistic antifouling mechanisms, as mentioned in the title. The synergistic antifouling mechanisms are well-established antifouling mechanisms, supported by previous studies (*Adv. Funct. Mater.*, 2011, 21, 191-198; *Chem. Soc. Rev.*, 2016, 45, 5888-592), which include both fouling resistance mechanisms and fouling release mechanisms. The former focuses on hindering the contact between the foulants and the membrane surface, while the latter focuses on the release of already adsorbed foulants. Previous research on fouling release mechanisms concentrated on promoting the dispersion of low-surface-energy microdomains and regulating the proportion of such microdomains, with some limitations, including suppressed membrane flux and antifouling performance depending on tangential flow assistance. The novel contribution of this study lies in the use of heterogeneous supramolecules to replace traditional heterogeneous macromolecules, establishing a method for regulating

membrane surface dynamics and enhancing the synergistic antifouling mechanism through dynamic modulation, thereby overcoming the aforementioned limitations. Our conclusions are supported by robust evidence: first, molecular simulations and quasi-elastic light scattering characterization demonstrate that CD/PDMS PPRs with different CD types and various $r_{\text{CD:PDMS}}$ leads to different dynamic properties. Additionally, through controlled experiments, we show that membranes with stronger dynamics exhibit superior antifouling properties and higher flux.

Regarding the "dynamic source" in Figure 1, the nested supramolecular structures exhibit dynamic behavior (Brownian motion) in a free state, with even stronger dynamics under flow disturbance conditions. This is not a new finding of our study (*J. Pharm. Sci.*, 2017, 106, 395-401; *J. Chem. Phys.*, 2010, 132, 054901). We are merely stating that CD/PDMS PPRs have this property and can make a unique contribution to surface modification of membranes.

Regarding the "push-off" mechanism of CDs. In fact, we never used the term "push-off." We referred to the process as "release," which is a standard description for fouling release mechanisms. The essence of this mechanism is that foulants interact with the membrane surface in an unstable state, and can subsequently be removed due to flow disturbance. We intended to express that dynamic behavior enhances the "release" process, and our corresponding characterization and performance experiments fully support this view. To avoid any further confusion, we have revised the wording and removed all elements in Figure 1 that may imply "push."

R2.1: (Manuscript-Line 98) The relative motion within the heterogeneous microdomains causes the contact between the foulants and the membrane surface to be in a highly unstable state, making it easier for the foulants to be released.

Fig. 1 Structure, dynamic source, and antifouling mechanism of novel antifouling modifier CD/PDMS PRs.

Q2.2: As the authors note, heterogeneous surfaces have been studied as fouling-resistant materials. At a minimum, CD is indeed quite hydrophilic (more than the base membrane or PDMS). I noticed that the authors created a separate table in the SI with the references I provided and claimed these are "hydrophilic" homogeneous materials. I suggest the authors read these papers more carefully as these papers, from multiple groups, all include copolymers of various architectures that combine hydrophilic and hydrophobic/LS units. For instance, reference 21 combines the hydrophilic MPC and SBMA with highly hydrophobic/LS trifluoroethyl methacrylate (TFEMA) groups. Reference 23 combines zwitterions (hydrophobic) with PES units. Reference 24 combines PEO and other charged groups with TFEMA. As such, these references should not be deemed outside of what should be compared with the manuscript described here. They are clear prior demonstration of the ability of multiple heterogeneous surfaces that combine hydrophilic and hydrophobic groups to resist fouling, with FRR values comparable to or better than reported in this document. It is important to acknowledge this, in my personal opinion.

R2.2: These studies do demonstrate excellent antifouling performance. However, their antifouling mechanism is fundamentally based on "fouling resistance" rather than a "synergistic antifouling mechanism." Synergistic antifouling mechanism requires a combination of hydrophilic and low-surface-energy materials, rather than hydrophilic and hydrophobic materials. The studies in question rely solely on the latter. The poly(trifluoroethyl methacrylate) component may have the lowest surface energy in these study (though it was not characterized in the studies), but it still does not meet the requirements for synergistic antifouling mechanism, as six continuous perfluorocarbons are typically considered necessary for good low surface energy properties. Moreover, the graphical abstract, the results and discussion sections of these studies explicitly state that the main role of the hydrophobic segments is to facilitate microphase separation from the hydrophilic segments, creating uniform membrane channels. The primary antifouling effect comes from the amphoteric or anionic groups in the hydrophilic segments (*Journal of Membrane Science*, 2017, 543, 184-194; *ACS Applied Materials & Interfaces*, 2017, 9, 20859-20872; *ACS Applied Materials & Interfaces*, 2020, 12, 19944-19954).

Furthermore, we have no intention of discrediting others' research, and we are simply presenting our results honestly. The raw data are faithfully provided in the supporting information. The application scenarios for these membranes are different—the studies recommended by the reviewer focus on antifouling nanofiltration membranes, whereas our work investigates antifouling ultrafiltration membranes. Compared to nanofiltration membranes, ultrafiltration membranes have larger pores and higher flux, which makes them more susceptible to pore blockage. Additionally, the higher flux means that foulant accumulation occurs at a faster rate. Balancing high flux with antifouling performance is a more challenging issue, and this is precisely where our study has made a breakthrough.

Q2.3: As I read carefully, the reported CD/PDMS ratios are extremely high, particularly given how much larger a CD molecule is in comparison with a PDMS repeat unit. From figures S6 and S7, you can see that when one CD molecule is placed on a PDMS chain,

it covers 3-4 repeat units within its core. In this light, CD:PDMS molar ratios between 0.27-0.67 do not look reasonable, and definitely do not appear/act like a clothesline as far as I can imagine. Even if the units were narrower, a CD:PDMS ratio of 0.67 would be more than every other repeat unit. This really needs to be addressed in the document, as it is in contrast with the proposed mechanisms.

A2.3: In PDMS, the repeating unit is defined as the distance from one silicon atom to another. Therefore, figures S6 and S7 show that each CD molecule covers approximately 1.5 repeating units of PDMS, not 3-4 units as the reviewer suggest. When the CD fully covers PDMS, the molar ratio is exactly 0.67, a trend not discovered for the first time by our study (*Macromolecules*, 2003, 36, 6422-6429).

The clothesline structure of CD/PDMS PR can be proved based on exclusivity. First, the GMA undergoes UV-induced polymerization on the membrane surface to form linear PGMA (*J. Mem. Sci.*, 2014, 455, 405-414), which we liken to the support of a clothesline. Secondly, at room temperature, only the terminal amines of PDMS in CD/PDMS PPRs can react with the epoxy groups of PGMA, while the hydroxyl groups of CDs are not sufficiently reactive. This selective fixation approach is very common in supramolecular studies (*Soft Matter*, 2007, 3, 1456–1473). At this stage, two possible structures exist: the first is that both ends of CD/PDMS PPR are linked to PGMA, forming the desired clothesline structure, and the second is that only one end of CD/PDMS PPR is linked to PGMA, resulting in a brush-like structure. We then used excess FITC, a fluorescent dye, to cap the unreacted PDMS amine groups and demonstrated through changes in the membrane surface fluorescence intensity that, as the reaction time increased, the majority of the CD/PDMS PPRs were doubly grafted, i.e., they behaved as the clothesline structure. We designed the clothesline structure to suspend the CD/PDMS PPRs, preventing direct contact with the membrane surface in order to preserve the mobility of the CDs.

Regarding the relationship between $r_{\text{CD:PDMS}}$ and the proposed mechanism, they not only do not conflict but actually support each other. Our experimental results show that as $r_{\text{CD:PDMS}}$ increases, the dynamics of CD/PDMS PPRs initially increase and then decrease (as measured by quasi-elastic light scattering), and the antifouling ability of

CD/PDMS@Ms also increases and then decreases, reaching a peak at $r_{\text{CD:PDMS}} = 0.27$ and 0.40. This is because, as $r_{\text{CD:PDMS}}$ increases, the number of mobile CDs increases, but the available space for CD movement becomes more restricted, thus resulting in a maximum dynamic value. We discuss this in the manuscript, stating: "As the density of CDs on the PDMS chain increases, the sliding behavior of individual CDs becomes increasingly hindered by the neighboring CDs."

Q2.4: I am still unclear about the XPS data, and the mismatch between CD:PDMS ratios vs the "molecular coverage" ratios from XPS. The authors claim, in their response, that "For example, in the case of γ -CD/PDMS@M-0.40, the membrane was tested twice after preparation. The molecular coverage ratio of CD to PDMS was 6.98 in the first test and 6.68 in the second test, with an average of 6.83. This corresponds to an $r_{\text{CD:PDMS}}$ of 0.41." I do not understand how this works, honestly.

A2.4: The concept of "membrane surface coverage of certain molecular" has been widely used in membrane research, particularly when focusing on the interface of membranes (*J. Mem. Sci.*, 2002, 207(2), 207-222). It typically appears in the detailed analysis of membrane surface XPS results. The physical meaning of this term refers to the proportion of a certain molecule on the membrane surface relative to the total number of molecules, which can be expressed in either molar ratio or mass ratio. The calculation is based on the element ratios obtained from XPS data. Since XPS results inherently contain errors, the calculated results are approximate values. To reduce errors, we applied two approaches: first, we used a determinant-based numerical method, and second, we calculated the average of two batches of membranes. For γ -CD/PDMS@M-0.40, in the first calculation, the surface coverage of CD was 43.28wt%, and the surface coverage of PDMS was 6.2wt%, with their ratio being 6.98 (misabeled as molar ratio in the last reply). The $r_{\text{CD:PDMS}}$ was defined in this study as the "molar ratio of CDs to PDMS units" to visually represent the space in which CDs can move on PDMS. To convert from surface coverage to $r_{\text{CD:PDMS}}$, the relative molecular weights of γ -CD and PDMS units need to be considered, which gives $6.98 \times (74.1/1297) \approx 0.40$. For the second calculation, $r_{\text{CD:PDMS}}$ was 0.38, and the average value was 0.39. The previously

mentioned value of 0.41 was likely due to approximation errors in the conversion process. All relevant calculations have been double-checked and verified in the revised manuscript.

R2.4: (Manuscript-Line 275) The calculated $r_{\text{CD:PDMS}}$ values of β -CD/PDMS@M surfaces were 0.12, 0.31, 0.42, and 0.49, while those of γ -CD/PDMS@M surfaces were 0.13, 0.27, 0.39, 0.51, and 0.70. Considering measurement uncertainties, it can be inferred that the grafting and capping processes on the membrane surface did not cause significant CD detachment.

Q2.5: About the membrane formation process and flux changes: The authors immerse the membrane into a DMF/water mixture for coating. The base membrane material is soluble in DMF and will likely swell heavily in this mixture. Therefore, it is crucial to include a base membrane immersed in DMF/water with no polymer in it as a control at least for its effect on water flux, as it can cause increased flux. The same is needed for the epoxy-modified membrane, which will make it more hydrophilic. Indeed, the fact that the PDMS-immersed membrane has a higher permeance (with nothing to attach the PDMS to the membrane) than the base implies there is a combination of these effects going on.

A2.5: Based on our experimental experience, a 1:1 DMF/water mixture does not cause significant swelling of the PES membrane. In fact, swelling of the PES membrane occurs only when the DMF/water ratio is at least 5:2. Even if the mixed solvent leads to some swelling of the PES base membrane, it only affects the comparison between the base membrane and the PGMA-grafted membrane, as well as the PGMA-grafted membrane and the CD/PDMS@M membrane. The former reflects the impact of hydrophilic modification on membrane performance, while the latter highlights the differences between “fouling resistance” and the “resistance-release synergistic antifouling mechanism,” which have been thoroughly discussed in previous studies and are outside the scope of this paper. The core goal of this study is to investigate the influence of supramolecular dynamics on fouling mechanisms. Therefore, it is essential to focus on the adjustment of the dynamic properties of the CD/PDMS PPRs. To this

end, we employed two methods: first, we varied the type of CD, with β -CD and γ -CD having similar chemical structures but differing ring sizes. The primary effect of this size difference is the steric hindrance between PDMS and CD, with larger steric hindrance resulting in reduced dynamic properties. Second, we adjusted the $r_{CD:PDMS}$. As the ratio increased, the number of movable CDs increased, but the available space for CD movement decreased, resulting in an overall increase in dynamics followed by a decrease. As for the effect of the mixed solvent, the experimental results show that it does not significantly impact the control of the related variables, so investigating its influence is unnecessary.

Regarding the comment on "the PDMS-immersed membrane having a higher permeance than the base membrane," this is not a finding from our study, and there seems to be a misunderstanding. In fact, there is an interaction between PDMS and the membrane surface, which we describe in the methodology section: "The PPRs were chemically linked to PGMA via an addition reaction between the epoxy groups on the side chains of PGMA and the terminal amines of PDMS."

Q2.6: I think the x axis in Figure 7 g-i is meant to be in hours, not minutes.

A2.6: This error has been corrected in the revised manuscript.

Q2.7: I appreciate the cross-flow data. However, once again, this needs to be benchmarked with a commercial membrane. Given the regular backwashing and chemical cleaning, coupled with the relatively low COD of these feeds, I am not positive about the advantage the coating offers without the benchmarking.

A2.7: In the revised manuscript, the cross-flow experiment section has been updated to include FilmTec™ TW30-2514 membrane from Dow DuPont as a control membrane. Additionally, Lurgi gasification wastewater with high COD was introduced as a test feedwater, while the original Songhua River water data has been moved to Supporting Information S13. Experimental results indicate that the antifouling performance of FilmTec™ TW30-2514 membrane is overall comparable to that of β -CD/PDMS@M-0.40, which exhibits low dynamic behavior, but inferior to γ -CD/PDMS PPRs, which

exhibit high dynamic behavior. Furthermore, the antifouling performance of γ -CD/PDMS PPRs did not significantly deteriorate with increasing COD content in the water. These findings further validate that the dynamics-enhanced synergistic antifouling mechanisms demonstrate significant advantages in maintaining high flux while effectively resisting and releasing composite foulants.

R2.7: (Manuscript-Line 414) To further evaluate the antifouling performance of the membrane under practical application conditions, a 72-hour cross-flow ultrafiltration experiment was conducted. FilmTec™ TW30-2514 membrane from Dow DuPont was used as the control group, cut from an unused spiral-wound membrane module. Its pure water flux at 1 bar and 20°C is approximately $530 \text{ L}\cdot\text{m}^{-2}\cdot\text{h}^{-1}$, which is similar to that of the γ -CD/PDMS@M-0.27, and it is renowned for its excellent antifouling capability. The feed solutions included one simulated composite foulant solution and three types of real water samples. The former was an equal-volume mixture of the five pollutant solutions/emulsions used in the dead-end five-cycle ultrafiltration experiment^{35, 36}, while the latter consisted of coagulation-settled municipal wastewater, Lurgi gasification wastewater and untreated Songhua River water (Fig. 7g~I and Supplementary Fig.13). The experiment included periodic cleaning every 4 hours using deionized water with both backwashing and forward flushing, while the first cleaning after the 48th hour was replaced with chemical cleaning. The results showed that γ -CD/PDMS@M maintained remarkable antifouling performance when exposed to both composite and real foulants. The FDR after 4 hours of continuous operation was only 3~7%, and FRR after deionized water cleaning was nearly 100%. The membrane flux at the 12th cycle (before chemical cleaning) remained at over 97% of its initial value, and chemical cleaning restored the flux to its original level. The antifouling performance of β -CD/PDMS@M was slightly inferior to that of β -CD/PDMS@M, with a flux decay rate of 6~12% after 4 hours of continuous operation and a membrane flux of over 94% of its initial value at the 12th cycle. The antifouling performance of FilmTec™ TW30-2514 is slightly better than that of β -CD/PDMS@M but inferior to γ -CD/PDMS@M. Compared to other studies heterogeneous membranes and hydrophilic membranes, the dynamics-enhanced synergistic antifouling mechanisms

demonstrated significant advantages in maintaining high flux and effectively resisting and releasing composite foulants (Supplementary Table 11 and 12).

Fig. 7 (a) Influence of $r_{CD:PDMS}$, (b) feed pH, (c) stirring rate, and (d) temperature on antifouling performance of CD/PDMS@Ms in dead-end ultrafiltration experiment. 5-cycle antifouling performance of (e) β -CD/PDMS@M and (f) γ -CD/PDMS@M with multiple foulants in dead-end ultrafiltration experiment. 72-hour cross-flow ultrafiltration experiment with feed of (g) composite foulant solution, (h) coagulation-settled municipal wastewater and (i) Lurgi gasification wastewater. Unless otherwise specified, the model foulant was BSA, the $r_{CD:PDMS}$ of β -CD/PDMS@M was 0.40, the $r_{CD:PDMS}$ γ -CD/PDMS@M was 0.27, and experiments were performed at pH 7, a stirring speed of 60 rpm, and temperature of 20°C.

(Supplementary Information-Line 163) The first real water sample was municipal wastewater treated by coagulation and sedimentation, with 39.7 mg/L suspended solids, 6.72 NTU turbidity, 117.5 mg/L COD (dichromate method), 43.9 mg/L TOC, and a pH of 7.57. The second real water sample was Lurgi gasification wastewater treated by coagulation and sedimentation, with 59.9 mg/L suspended solids, 3130 mg/L COD

(dichromate method), 1397 mg/L TOC, and a pH of 7.5–9.0. The third real water sample was untreated Songhua River water, with 55.1 mg/L suspended solids, 5.72 NTU turbidity, 26.2 mg/L COD (determined by the dichromate method), 10.25 mg/L TOC, and a pH of 7.25.

(Supplementary Information-Line 284)

Supplementary Fig. 13 72-hour cross-flow ultrafiltration experiment with feed of untreated Songhua River water at 20°C.

Q2.8: In their responses, the authors state the novelty of their work differently in different sections. The PPRs reported are not novel on their own. The authors state "The main innovations of this study lie in the optimization of PPR preparation and the synergistic antifouling mechanism." The optimization of a material in terms of component ratios is of course valuable, but I am not sure it fits in this journal. In other sections, they also note "This study aims to highlight the potential of the supramolecular dynamics of CD/PDMS PRs in enhancing membrane flux and antifouling capabilities, rather than focusing on the modification effects of hydrophilic-LSE materials—an aspect that has been thoroughly analyzed in our prior research." The data reported, however, does not really show clear insight into these mechanisms as is...

A2.8 Regarding the novelty of our work, we have already clarified the innovative aspects related to the mechanisms and supporting evidence in our response to Question

2.1. Here, we address the innovation concerning the materials. Through molecular simulations and experimental analyses, we have developed a method to synthesize β -CD/PDMS PPRs with a high $r_{\text{CD:PDMS}}$ of 0.59 using high-molecular-weight PDMS (5000 Da), with a yield of 45%. This development allows us to investigate the impact of $r_{\text{CD:PDMS}}$ on the dynamic properties of both β -CD/PDMS and γ -CD/PDMS PPRs. Prior to this study, β -CD/PDMS PPRs could only be synthesized with PDMS of molecular weight below 1000 Da, with the maximum $r_{\text{CD:PDMS}}$ being around 0.3 (*Macromolecules*, 2003, 36, 6422-6429; *Macromolecules* 2001, 34, 6338-6343; *Soft Matter*, 2007, 3, 1456–1473). In the revised manuscript, we have modified the relevant descriptions to better emphasize the innovation in this aspect.

R2.8: (Manuscript-Line 149) Experimental results (see the Preparation of CD/PDMS PPRs section for details) confirmed that using DMF as a solvent instead of water enabled the controlled synthesis of β -CD/PDMS PPRs with 5000 Da PDMS. The maximum $r_{\text{CD:PDMS}}$ achieved was 0.59, with a yield of 45%, which meets the study's requirements for controlling the dynamic properties of β -CD/PDMS PPRs by adjusting the $r_{\text{CD:PDMS}}$.

Q2.9: To be clear, I am not aware of any membrane materials that combine PDMS, a very high mobility chain material, with hydrophobic groups, so there is still some novelty. I do believe this manuscript has a home in the literature, and this is an interesting membrane material. However, given how few membrane papers this journal publishes, I am not sure it hits that bar of novelty specifically based on how the authors state their intended contribution and impact.

A2.9: In this study, the CD/PDMS PPRs are indeed created by combining low-surface-energy PDMS with hydrophilic CDs, rather than hydrophobic ones. However, the true innovation lies in harnessing their dynamic properties to enhance the synergistic antifouling mechanism. The ingenuity of this approach lies in the inherent low-interaction nature of the low-surface-energy component, which not only plays a critical role in the antifouling synergy but also significantly contributes to the dynamics of the CD/PDMS PPRs, resulting in reduced molecular friction. This advancement addresses

two major challenges in the development of heterogeneous antifouling mechanisms for water treatment membranes. The contributions of this study are groundbreaking, setting the stage for further development in the field of supramolecular dynamics enhanced antifouling technologies. Given the novelty and depth of these findings, we are confident that they meet the high standards expected of leading journals such as Nature Communications.

Reviewer #3

The authors have solved all my concerns, and I think it can be considered to be accepted.

A: Thank you very much for your guidance and support. We are grateful for your insightful comments and are pleased that our revisions have addressed your concerns. We greatly appreciate your positive feedback.

RESPONSE TO REVIEWERS' COMMENTS

Reviewer #1:

The authors have addressed all my questions.

A: We appreciate the reviewer's thoughtful feedback and are pleased to know that all questions have been satisfactorily addressed. Thank you again for your time and support.

Reviewer #2:

I appreciate the responses of the authors. I still have a few comments.

A: We sincerely thank the reviewer for the continued careful evaluation of our manuscript and for the constructive feedback. We appreciate your recognition of our previous responses and welcome the additional comments. We have carefully addressed each of them below and revised the manuscript accordingly.

Q2.1: I appreciated the explanations the authors provided in the response document. However, very few of these seem to be reflected to the main text. Some of the phenomena are not described clearly - especially given the interdisciplinary nature of the paper, where some aspects and terminology will be obvious to researchers in one area but not others.

An example is the discussion of the maximum CD:PDMS ratios, which the authors allude to in the text as follows:

"While the 126 synthesis of β -CD/PDMS PPRs was possible, the maximum molar ratio of CDs to 127 PDMS units ($r_{\text{CD:PDMS}}$) and the synthesis yield (calculated based on CD) declined when 128 the molecular weight of PDMS exceeded 1000 Da. In contrast, the synthesis of γ -129 CD/PDMS PPRs showed limited sensitivity to PDMS chain length (maximum $r_{\text{CD:PDMS}}$ 130 = 0.67, maximum yield = 90%)^{28, 31}." This statement does not clarify what limits the CD:PDMS ratio; it sounds a lot more like an experimental limitation than a CD coverage effect. I recommend clarifying this in the text better also.

A2.1: We have carefully reviewed the manuscript and made several revisions to improve clarity, particularly in light of the interdisciplinary nature of the work. These revisions include the addition of explanations for specialized terminology, the correction of potentially misleading statements, and the removal of repetitive content to balance readability with the manuscript's length constraints. The major modifications are as follows:

1) In the last paragraph of the Introduction, we have explicitly outlined the focus of this study: the type of CD and the molar ratio of CD to PDMS in CD/PDMS PPRs, as well as how these factors influence supramolecular dynamics. We have also emphasized how this approach differs from traditional strategies that regulate hydrophilic-LSE microdomain ratios.

2) We clarified in the text that steric hindrance is the primary factor limiting the synthesis of α -CD/PDMS PPRs and the maximum achievable $r_{\text{CD:PDMS}}$ in β -CD/PDMS PPRs. Subsequently, we elaborated on how this study seeks to overcome that

limitation—namely, by altering the solvent system to enhance the thermodynamic driving force for PPR formation to counteract steric effects.

3) We have added justification for using sliding behavior as a representative of supramolecular dynamics, explaining that sliding, rotation, and vibration are all inversely correlated with steric hindrance. Additionally, we clarified that a higher Gibbs free energy barrier indicates greater steric hindrance and therefore a lower potential for dynamic motion.

4) We rewrote the section describing the simulation method for CD migration along the PDMS chain. The revised text now explains why a solvent-free model was used and eliminates ambiguities that may have caused confusion in the original version.

R2.1: (**Manuscript-Line 99**) Specifically, this study investigated the influence of supramolecular dynamics on membrane permeability and synergistic antifouling performance by tuning the type of CD and the molar ratio of CD to PDMS in CD/PDMS PPRs. Different types of CDs possess similar chemical compositions but vary in cavity size, which affects the spatial hindrance during their movement. This allows the elimination of compositional influences related to the hydrophilic–LSE microdomain ratio, enabling an isolated assessment of microdomain dynamics. By adjusting the CD/PDMS molar ratio, both the number of mobile CDs and their available movement space can be simultaneously regulated, facilitating the identification of supramolecular structures with the highest dynamic mobility.

(**Manuscript-Line 118**) According to the results, α -CD/PDMS PPRs could not be synthesized. The synthesis of γ -CD/PDMS PPRs, however, was straightforward, with a yield (calculated based on CD) exceeding 90%. When γ -CD fully covered PDMS, the molar ratio of CDs to PDMS units ($r_{\text{CD:PDMS}}$) was 0.67, which is the theoretical maximum value for $r_{\text{CD:PDMS}}$ ^{28, 31}. For β -CD/PDMS PPRs, synthesis was possible, but the achievable $r_{\text{CD:PDMS}}$ significantly decreased when the molecular weight of PDMS exceeded 1000 Da. This is because the formation of CD/PDMS PPRs essentially involves the displacement of solvent molecules from the CD cavities by PDMS molecules. This process is reversible and influenced by steric hindrance, which acts as a resistance, and solvent polarity, which serves as the driving force. The cavity diameter of α -CD is small, causing excessive steric hindrance for the entry of PDMS, which makes synthesis difficult. γ -CD, with a larger and more hydrophobic cavity, facilitates the entry of PDMS. β -CD has an intermediate cavity size, allowing PDMS to enter, but the steric hindrance is more significant. As the number of β -CD molecules that interact with PDMS increases, the steric hindrance becomes even more pronounced. Steric hindrance is fixed for certain CD, so the solvent conditions were optimized to attempt the synthesis of α -CD/PDMS PPRs and to improve the maximum $r_{\text{CD:PDMS}}$ and yield of β -CD/PDMS PPRs.

(**Manuscript-Line 168**) Molecular simulations were conducted to investigate the supramolecular dynamics of β -CD/PDMS PPRs and γ -CD/PDMS PPRs. Theoretically, the sliding, rotational, and vibrational motions of CDs along the PDMS chains are all influenced by steric hindrance between the CDs and PDMS, exhibiting a negative correlation. As a representative parameter, the Gibbs free energy barrier associated with CD sliding along the PDMS chain was used to evaluate the impact of steric hindrance.

A lower sliding energy barrier reflects lower steric hindrance, thereby enabling a higher potential for supramolecular mobility.

(Supplementary Information-Line 50) To evaluate the migration behavior of CD along the PDMS chain, a separate simulation was conducted without solvent molecules to improve computational efficiency and ensure reproducibility. Since the dominant interactions in this system are van der Waals forces and steric hindrance, the key characteristics of CD migration along the PDMS chain are expected to be retained even in the absence of solvent. The simulation involved a single CD molecule and a 10-mer PDMS chain. The PDMS chain was configured in a helical conformation aligned along the z-axis of a 3D coordinate system, and it was fixed in place with its geometric center constrained on the z-axis. The CD molecule was translated stepwise along the PDMS chain, with each step set to one-tenth of the projected z-axis length of a single PDMS monomer unit. The migration consisted of two stages: in the first stage, the CD moved toward the direction of its larger aperture for a total distance equivalent to two PDMS monomer units along the z-axis projection; in the second stage, the CD reversed direction and moved toward its smaller aperture for the same total distance. A time step of 1 fs was used, and structural optimization was performed after each step to ensure system relaxation. The Gibbs free energy (ΔG) of the system was recorded before and after each step, resulting in 21 data points in each direction. By analyzing the ΔG profile throughout the CD migration process, the resistance encountered by the CD can be inferred.

Q2.2: A similar issue is with the XPS section. It seems the surface coverage is based on a mass fraction; this is not clearly stated anywhere - it must be specified. The word "molecular coverage" with an undefined percentage implies a mole percentage; it needs to be stated either way. The CD:PDMS ratio is a mole ratio.

A2.2: The surface coverage, based on mass fraction, has now been explicitly clarified in the revised manuscript, including both the text description and the units on the y-axis of Fig. 5g, h.

R2.2: (Manuscript-Line 258) The XPS full spectra of the CD/PDMS@M surfaces showed oxygen, carbon, sulfur, and silicon peaks, although the nitrogen content was below the detection limit. The full spectrum of γ -CD/PDMS@M-0.27 is shown in Fig. 5d as a representative example. Disregarding the nitrogen elements from the PDMS terminal amino group and isothiocyanate fluorophore, the surface coverage, **in terms of mass percentage**, of CD, PDMS, PGMA, and PES molecules on the membrane surfaces was approximately calculated based on the proportions of oxygen, carbon, sulfur, and silicon elements (Fig. 5g, h, the calculation process is detailed in Supplementary Table 4–7). The extent of membrane surface modification was approximately **40wt%-55wt%**, with a slight increase with increasing $r_{\text{CD:PDMS}}$ value of CD/PDMS PRRs. This was attributed to the higher CD content on each PDMS chain. The calculated $r_{\text{CD:PDMS}}$ values of β -CD/PDMS@M surfaces were 0.12, 0.31, 0.42, and 0.49, while those of γ -CD/PDMS@M surfaces were 0.13, 0.27, 0.39, 0.51, and 0.70. Considering measurement error, it can be inferred that the grafting and capping processes on the membrane surface did not cause significant CD detachment from

PDMS chains.

Fig. 5 (a) Schematic illustration of CD/PDMS PR linkage modes on membrane surface. (b) Relative fluorescence intensities of γ -CD/PDMS@M surfaces after grafting PPRs for different durations and capping with FITC for 2 h. SEM images showing surfaces of (c) BM, (d) PDMS@M and (e) β -CD/PDMS@M-0.13. (f) XPS full spectrum of γ -CD/PDMS@M-0.27. Surface molecular coverage of (g) β -CD/PDMS@Ms and (h) γ -CD/PDMS@Ms.

Q2.3: In every figure where flux is reported, the pressure difference used in the experiment must be noted in the caption. I think 1 bar, was used, but not this is not clear anywhere in the main text as the experimental section in the main text does not discuss filtration experiments; it is only in the SI section.

A2.3: It has been explicitly stated that the membrane flux and antifouling performance experiments were conducted under a transmembrane pressure difference of 1.0 bar at the captions of Fig. 6 and 7 in the revised manuscript. In addition, details regarding the stirring speed in the dead-end filtration experiments and the tangential flow velocity on the membrane surface in the cross-flow filtration experiments have also been supplemented.

R2.3: (Manuscript-Line 327) Fig. 6 (a) Advancing contact angles, (b) receding contact angles, and (c) surface energies of CD/PDMS@Ms. Initial water flux of CD/PDMS@Ms at (d) 5°C, (e) 20°C, (f) 35°C and (g) 50°C. (h) Flux variation of γ -CD/PDMS@M-0.27 with the feed of SDS solution. (i) Flux of CD/PDMS@Ms with

1.00 wt% SDS feed at 35°C. **All experiments were conducted under a transmembrane pressure of 1.0 bar.**

(Manuscript-Line 457) Fig. 7 (a) Influence of $r_{\text{CD:PDMS}}$, (b) feed pH, (c) stirring rate, and (d) temperature on antifouling performance of CD/PDMS@Ms in dead-end ultrafiltration experiment. Variations in membrane flux of (e) β -CD/PDMS@M and (f) γ -CD/PDMS@M over five filtration cycles with multiple foulants in dead-end ultrafiltration, the baseline flux corresponds to the pure water flux. Variations in membrane flux of CD/PDMS@Ms and Synder Filtration MQ membrane with feeds of (g) composite foulant solution, (h) coagulation-settled municipal wastewater and (i) Lurgi gasification wastewater in 72-hour cross-flow ultrafiltration, the baseline flux refers to the stabilized membrane flux after 30 minutes of operation. **Unless otherwise stated, all experiments were conducted using BSA as the model foulant, with $r_{\text{CD:PDMS}}$ of 0.40 (β -CD/PDMS@M) and 0.27 (γ -CD/PDMS@M), at approximately pH 7, 20 °C and 1.0 bar. Dead-end tests used 60 rpm stirring while cross-flow tests employed a tangential velocity of approximately 0.12 m/s when the membrane flux was 500 L·m⁻²·h⁻¹.**

(Supplementary Information-Line 50) Supplementary Fig. 13 Variations in membrane flux of CD/PDMS@Ms and Synder Filtration MQ membrane with feeds of untreated Songhua River in 72-hour cross-flow ultrafiltration, the baseline flux refers to the stabilized membrane flux after 30 minutes of operation. The $r_{\text{CD:PDMS}}$ of β -CD/PDMS@M and γ -CD/PDMS@M were 0.40 and 0.27, respectively. **Experiments were conducted at 20 °C with a transmembrane pressure of 1.0 bar and a tangential flow velocity of approximately 0.12 m/s when the membrane flux was 500 L·m⁻²·h⁻¹.**

Q2.4: I appreciate the addition of the base membrane, but there is some issue going on there. The FilmTec membrane reported is a brackish water RO membrane, not a UF membrane. A quick search led me to:

https://www.home-water-purifiers-and-filters.com/filmtec-TW30-2514.php?srsId=AfmBOop-aNvB7S5YZt0Df24tm2eLjI1JVwI28SaErRIens_mdIK3uEOd

I back-calculated its permeance to be 3.2 LMH/bar (so its flux at 1 bar would be 3.2 LMH) based on these data sheets. Granted testing conditions will affect this value - but still, the reported flux seems quite unreasonable. Maybe the membrane acquired defects during the construction and deconstruction of the module; maybe something else happened - but the data seems problematic. The authors also note this is a UF membrane, so they should select a UF membrane. Perhaps the base membrane could be an option.

A2.4: Thank you very much for your careful reading and insightful feedback. We sincerely apologize for the mistake in the membrane identification. Upon review, we found that the commercial control membrane used in our experiments was most likely the MQ ultrafiltration membrane from Synder Filtration (PES 50,000Da), rather than the FilmTec™ TW30-2514 membrane. To ensure the accuracy and reproducibility of the data, we re-conducted the antifouling experiments using a new Synder MQ membrane over a 72-hour period. The results showed similar fouling behavior to our

previous data, although the flux was slightly higher, likely due to batch-to-batch variability.

We aimed to demonstrate the practical performance of our membrane, including its advantages and limitations, by comparing it with a representative commercial ultrafiltration membrane currently used in real applications. Since our lab-fabricated base membrane—designed specifically as a support for supramolecular modification—is relatively hydrophobic and lacks low-surface-energy groups, it is not suitable for long-term operation (as shown in revised Figure R1). Therefore, we chose to use a commercial UF membrane as the benchmark for comparison.

Fig. R1 Variations in membrane flux of CD/PDMS@Ms and BM with feeds of untreated composite foulant in 72-hour cross-flow ultrafiltration, the baseline flux refers to the stabilized membrane flux after 30 minutes of operation. The $r_{CD:PDMS}$ of β -CD/PDMS@M and γ -CD/PDMS@M were 0.40 and 0.27, respectively. Experiments were conducted at 20 °C with a transmembrane pressure of 1.0 bar and a tangential flow velocity of approximately 1.12 m/s when the membrane flux was 500 L·m⁻²·h⁻¹.

R2.4: **(Manuscript-Line 405)** To further evaluate the antifouling performance of the membrane under practical application conditions, a 72-hour cross-flow ultrafiltration experiment was conducted. **MQ (PES 50,000Da) membrane from Synder Filtration, renowned for its good antifouling capability, was used as the control group, cut from an unused spiral-wound ultrafiltration membrane module.** The feed solutions included one simulated composite foulant solution and three types of real water samples. The former was an equal-volume mixture of the five pollutant solutions/emulsions used in the dead-end five-cycle ultrafiltration experiment^{35, 36}, while the latter consisted of coagulation-settled municipal wastewater, Lurgi gasification wastewater and untreated Songhua River water (Fig. 7g~i and Supplementary Fig.13). The experiment included periodic cleaning every 4 hours using deionized water with both backwashing and forward flushing, while the first cleaning after the 48th hour was replaced with chemical cleaning. The results showed that γ -CD/PDMS@M maintained remarkable antifouling performance when exposed to both composite and real foulants. The FDR after 4 hours of continuous operation was only 3~7%, and FRR after deionized water cleaning was

nearly 100%. The membrane flux at the 12th cycle (before chemical cleaning) remained at over 97% of its initial value, and chemical cleaning restored the flux to its original level. The antifouling performance of β -CD/PDMS@M was slightly inferior to that of γ -CD/PDMS@M, with a flux decay rate of 6~12% after 4 hours of continuous operation and a membrane flux of over 94% of its initial value at the 12th cycle. **The antifouling performance of the Synder Filtration MQ membrane was comparable to that of β -CD/PDMS@M, with the latter showing a somewhat higher flux recovery rate under hydraulic cleaning conditions.** Compared to membranes in other studies, including heterogeneous and hydrophilic membranes, the dynamics-enhanced synergistic antifouling mechanisms demonstrated significant advantages in maintaining high flux and effectively resisting and releasing composite foulants (Supplementary Table 11 and 12).

Fig. 7 (a) Influence of $r_{CD:PDMS}$, (b) feed pH, (c) stirring rate, and (d) temperature on antifouling performance of CD/PDMS@Ms in dead-end ultrafiltration experiment. Variations in membrane flux of (e) β -CD/PDMS@M and (f) γ -CD/PDMS@M over five filtration cycles with multiple foulants in dead-end ultrafiltration, the baseline flux corresponds to the pure water flux. **Variations in membrane flux of CD/PDMS@Ms and Synder Filtration MQ membrane with feeds of (g) composite foulant solution, (h) coagulation-settled municipal wastewater and (i) Lurgi gasification wastewater in 72-hour cross-flow ultrafiltration, the baseline flux refers to the stabilized membrane flux after 30 minutes of operation.** Unless otherwise stated, all experiments were conducted using BSA as the model foulant, with $r_{CD:PDMS}$ of 0.40 (β -

CD/PDMS@M) and 0.27 (γ -CD/PDMS@M), at approximately pH 7, 20 °C and 1.0 bar. Dead-end tests used 60 rpm stirring while cross-flow tests employed a tangential velocity of approximately 0.12 m/s when the membrane flux was 500 L·m⁻²·h⁻¹.

(Supplementary Information-Line 290)

Supplementary Fig. 13 Variations in membrane flux of CD/PDMS@Ms and Synder Filtration MQ membrane with feeds of untreated Songhua River in 72-hour cross-flow ultrafiltration, the baseline flux refers to the stabilized membrane flux after 30 minutes of operation. The $r_{CD:PDMS}$ of β -CD/PDMS@M and γ -CD/PDMS@M were 0.40 and 0.27, respectively. Experiments were conducted at 20 °C with a transmembrane pressure of 1.0 bar and a tangential flow velocity of approximately 0.12 m/s when the membrane flux was 500 L·m⁻²·h⁻¹.

Q2.5: Every figure with Normalized flux should include the base flux all values were normalized with for each membrane clearly in the caption.

A2.5: In the dead-end ultrafiltration experiments, pure water flux was measured and used as the baseline for normalization. In the cross-flow ultrafiltration experiments, pure water flux was not measured; therefore, the baseline flux was defined as the stabilized flux after 30 minutes of operation. This information has been clarified in the caption and legend of Figure 7.

R2.5: As shown in R2.4.

Title: *Supramolecular Dynamics-Enhanced Synergistic Antifouling Mechanisms for Superior Membrane Antifouling and Permeability*

Authors: Mingrui He, Yunlun He, Mengfei Wang, Junjie Yang, Tong Wu, Dongwei Lu*, and Jun Ma*

The manuscript reports a novel approach utilizing a surface modifier composed of hydrophilic cyclodextrin (CD) rings threaded on hydrophobic polydimethylsiloxane (PDMS) to enhance the antifouling properties of polyethersulfone (PES) ultrafiltration membranes. While the concept appears innovative and the reported membrane performance improvements are notable, I have major concerns in the ambiguity of the presentations of the manuscript, which often make it almost impossible to understand. It is more about the logic flow of ideas and cohesiveness of the presentations rather than a language issue. The following are some examples.

1. The authors state that simulation on the CD/PDMS ratios were "partially confirmed by verification experiments". Questions: 1) What was the simulation method? And 2) what was the methodology of the verification experiments?
2. It seems that the molecular simulation (Section 1.3, Supplementary Information) was used to determine the CD/PDMS ratios, a critical parameter in the study, in some places, e.g., Lines 168-170. However, this seems to be unlikely since the simulation was about calculating the energy changes in association with the "Brownian motions" of CDs along their PDMS thread. Whereas the experimental procedures of CD/PDMS threading (Lines 474-495) suggest that the probability of CDs to align satisfactorily well with one end of a PDMS molecule in the homogenized mixture should dictate the CD/PDMS ratio.
3. If you used molecular simulation for the Brownian motions, please make it clear in relevant parts in both the manuscript and the supplementary information.
4. How to interpret the significance of the diffusion coefficients (Figure 4) and how exactly were they determined?
5. The authors assert that "the dynamic movement of CDs along PDMS promotes water transport. (Lines 21-22)". However, no experimental or theoretical evidence was provided to support this claim. If it is a hypothesis, please explicitly state so. Otherwise, please supply evidence or revise this statement to align with the presented data.
6. In crossflow membrane filtration systems, the tangential flow would likely cause unidirectional movement of CD rings towards one end of the PDMS.
 - o The manuscript does not discuss mechanisms or forces that could counteract this movement. Are there any theoretical or experimental findings to suggest stabilization of CD positioning under tangential flow conditions? Would the aforementioned Brownian motions be sufficient?

7. Please reword Lines 475–480, particularly the phrase "oversupplying β -CD at approximately 0.15, 0.19, 0.22, and 0.25 times", the meaning of which I can only guess.

Addressing the above points is essential to enhance the manuscript's clarity, rigor, and credibility. Specifically, discrepancies in methodology, incomplete descriptions of experiments, missing links between claims, data, and experimental methods, as well as unsupported claims need to be resolved to ensure the findings are robust and creditable. I look forward to reviewing a revised manuscript that addresses these concerns comprehensively.